# Loss of *Kmt2c* or *Kmt2d* primes urothelium for tumorigenesis and redistributes KMT2A–menin to bivalent promoters

Naitao Wang[1,10], Mohini R. Pachai[1,10], Dan Li [1,10], Cindy J. Lee[1], Sarah Warda [1], Makhzuna N. Khudoynazarova[1], Woo Hyun Cho [1], Guojia Xie[2], Sagar R. Shah[3,4], Li Yao [4,5], Cheng Qian[1], Elissa W. P. Wong[1], Juan Yan[1], Fanny V. Tomas[1], Wenhuo Hu[1], Fengshen Kuo [6], Sizhi P. Gao[1], Jiaqian Luo [1], Alison E. Smith [1], Ming Han[7], Dong Gao [7], Kai Ge[2], Haiyuan Yu [4,5], Sarat Chandarlapaty [1,8,9], Gopakumar V. Iyer [8,9], Jonathan E. Rosenberg[8,9], David B. Solit [1,8,9], Hikmat A. Al-Ahmadie [1,6], Ping Chi [1,8,9] ✉ & Yu Chen [1,8,9] ✉

Members of the KMT2C/D–KDM6A complex are recurrently mutated in urothelial carcinoma and in histologically normal urothelium. Here, using genetically engineered mouse models, we demonstrate that *Kmt2c/d* knockout in the urothelium led to impaired differentiation, augmented responses to growth and inflammatory stimuli and sensitization to oncogenic transformation by carcinogen and oncogenes. Mechanistically, KMT2D localized to active enhancers and CpG-poor promoters that preferentially regulate the urothelial lineage program and *Kmt2c/d* knockout led to diminished H3K4me1, H3K27ac and nascent RNA transcription at these sites, which leads to impaired differentiation. *Kmt2c/d* knockout further led to KMT2A–menin redistribution from KMT2D localized enhancers to CpG-high and bivalent promoters, resulting in derepression of signal-induced immediate early genes. Therapeutically, *Kmt2c/d* knockout upregulated epidermal growth factor receptor signaling and conferred vulnerability to epidermal growth factor receptor inhibitors. Together, our data posit that functional loss of *Kmt2c/d* licenses a molecular 'field effect' priming histologically normal urothelium for oncogenic transformation and presents therapeutic vulnerabilities.

The urothelium lines the renal pelvis, ureters, bladder and proximal urethra. Urothelial carcinoma is classified as non-muscle invasive, which can be treated with local excisions and intravesical therapy, or muscle-invasive bladder cancer (MIBC), which requires cystectomy and chemotherapy. Urothelial carcinoma is highly relapsing, often at distinct sites[1], and sequencing studies have shown that spatially separated urothelial carcinoma of bladder and renal pelvis are frequently clonally related[2]. Thus, urothelial carcinoma has been hypothesized

to arise from a field of precancerous, molecularly perturbed but histologically normal urothelium[3,4].

Both non-muscle-invasive urothelial carcinoma and MIBC frequently harbor mutations in epigenetic modifiers (for example, loss-of-function mutations in *KDM6A*, *KMT2C*, *KMT2D*, *STAG2* and *ARID1A*)[5,6]. Two recent studies of histologically normal urothelium identified frequent loss-of-function mutations in *KDM6A*, *KMT2C*, *KMT2D*, *STAG2* and *ARID1A*, whereas mutations in *FGFR3*, *KRAS*, *PIK3CA*,

*TP53* and *RB1* were uncommon[7,8], suggesting that mutations of epigenetic modifiers are early tumorigenic events that may underlie field cancerization. The lysine methyltransferases KMT2C or KMT2D together with the lysine demethylase KDM6A are core components of the KMT2C/D–KDM6A complex that mediate monomethylation of histone H3K4 at enhancers[9–12]. Prior studies have shown that loss of KDM6A leads to disruption of urothelial differentiation[13,14]. Mouse bladder cancers generated by *N*-butyl-*N*-(4-hydroxybutyl)-nitrosamine (BBN), a carcinogen found in tobacco[15], also harbor prevalent mutations in the KMT2C/D–KDM6A complexes[16].

To characterize the role of *Kmt2c* and *Kmt2d* loss in the urothelium in vivo, we generated a genetically engineered mouse model (GEMM) of urothelial loss of *Kmt2c* and/or *Kmt2d* and studied the alterations in transcriptional regulation, epigenetic reprogramming and tumorigenesis.

## Results

### Kmt2c/d loss is insufficient to induce urothelial carcinoma

To investigate the functional consequences of depletion of *Kmt2c* or *Kmt2d* in urothelium, we utilized *Tmprss2-CreER^T2-IRES-nlsEGFP* (referred to as *Tmprss2-CreER^T2* onward) that mediates tamoxifen-induced LoxP recombination in epithelial cells of the bladder, prostate and gastrointestinal tract[17]. In the bladder, EGFP fused to nuclear localization sequence (nlsEGFP) was specifically detected by immunohistochemistry (IHC) and fluorescence-activated cell sorting (FACS) in EpCAM-positive urothelial cells and not stromal cells in the microenvironment (Extended Data Fig. 1a,b). We crossed *Tmprss2-CreER^T2* with *Kmt2c^f/f* and/or *Kmt2d^f/f* to induce conditional knockout (KO) of *Kmt2c* and/or *Kmt2d* in the urothelium (Fig. 1a)[18,19]. In *Tmprss2-CreER^T2; LSL-EYFP* control mice, we observed enhanced yellow fluorescent protein (EYFP) expression in almost all urothelial cells 1 week after tamoxifen administration (Fig. 1b). Over the 8 months after tamoxifen administration, most mice remained healthy and viable, though 1/11 males in the *Kmt2c^f/f* (*Kmt2c* KO) and 2/11 males in the *Kmt2c^f/f;Kmt2d^f/f* (*Kmt2c/d* double knockout (dKO)) groups died with hydronephrosis (Fig. 1c,d). Histological examination of the ureter and bladder 6 months post tamoxifen administration showed no abnormalities in most mice, with ureteral hyperplasia in a few male mice (Fig. 1e,f), consistent with previous observations with germline KO of *Kmt2c*[20].

We confirmed deletion of the floxed alleles by in situ hybridization (BaseScope) using probes targeting the floxed exons (Extended Data Fig. 1c,d). In addition, IHC of H3K4me1, a direct substrate for the catalytic function of KMT2C/D–KDM6A complexes, showed a progressive decrease in nuclear staining in the urothelium but not in stroma of *Kmt2c* KO, *Kmt2d* KO and dKO mice (Fig. 1g). These data indicate that *Kmt2c/d* KO and the downstream effect on chromatin modification can be maintained over time without histological changes.

### Kmt2c/d loss induces a oncogenically primed molecular state

We performed single-cell RNA sequencing (scRNA-seq) on FACS-sorted urothelial cells from *Tmprss2-CreER^T2;Kmt2c^f/f;Kmt2d^f/f* mice either 3 months post tamoxifen administration (dKO), or matched non-tamoxifen treated mice (wild-type, WT) (Extended Data Fig. 1b). We analyzed transcriptomes from 16,818 single cells in the WT (*n* = 4) and 9,040 single cells in the dKO mice (*n* = 3). Using Uniform Manifold Approximation and Projection (UMAP) and Leiden clustering, we identified seven clusters (Fig. 2a and Supplementary Fig. 1a–c). Cells from the WT and dKO groups were well separated, reflecting global transcriptional alterations (Fig. 2a and Supplementary Fig. 1a). Only 96 cells (1.06%) in the dKO group clustered with WT cells, probably due to incomplete recombination (Fig. 2b).

The urothelium comprises a basal cell layer where stem cells reside, intermediate cells and luminal cells with barrier functions. WT urothelial cells clustered into four groups (Fig. 2a) that were not well separated but represented a continuum, consistent with prior reports[21]. We classified these clusters as WT-basal, WT-intermediate 1, WT-intermediate 2 and WT-luminal cell clusters. *Kmt2c/d* dKO urothelial cells clustered into three groups, dKO clusters 1–3, also with a gradient of basal and luminal markers (Fig. 2a–e). dKO cells expressed higher basal markers (for example, *Krt5*, *Krt14* and *Col17a1*) and lower luminal markers (for example, *Uroplakins*, *Krt8* and *Krt20*) (Fig. 2c–e and Supplementary Fig. 1d). dKO cluster 1 exhibited expression of *Krt14* (Fig. 2d) that is found in rare basal cells with regenerative and tumorigenic properties[22]. Immunofluorescence staining confirmed an increased number of KRT14-positive cells and decreased UPK2 and KRT20 expression in dKO urothelium (Fig. 2f,g). Bladder cancer data from the The Cancer Genome Atlas (TCGA) showed an increased basal signature in human bladder cancers with *KMT2C* and/or *KMT2D* mutations (Fig. 2h). In addition, dKO cluster 1 harbored cells that express the epithelial–mesenchymal transition (EMT) marker *Zeb2*, as well as cells with decreased expression of epithelial marker *Cdh1* (Fig. 2c). We observed an approximately twofold increase of *Mki67*-positive cells in dKO mice (Extended Data Fig. 2a), which corresponded to a trend of increased Ki-67-positive nuclei by IHC in dKO mice (Extended Data Fig. 2b).

To characterize the clonal fitness of dKO urothelial cells in situ, we followed *Kmt2c and Kmt2d* BaseScope signal over time. We observed a gradual decrease (Extended Data Fig. 2c), suggesting a selection advantage of dKO urothelial cells. To assay stemness, we performed organoid formation assays using freshly FACS-sorted urothelial cells[23–25]. We observed significantly increased organoid formation efficiency in *Kmt2c/d* dKO urothelial cells (Fig. 2i,j). WT organoids formed small hollow spheres and dKO organoids exhibited larger size, irregular-shaped borders with filled lumen, a decreased number of KRT8-positive cells and an increased number of Ki-67-positive cells (Fig. 2i and Extended Data Fig. 3a,b). Under the 'differentiation' condition of growth factor withdrawal[25], WT organoids differentiated into luminal cells with high levels of KRT8 expression, whereas dKO organoids remained KRT8 negative to low (Extended Data Fig. 3c,d). Further, quantitative PCR with reverse transcription showed increased basal cell markers, decreased luminal cell markers and elevated EMT markers under both full medium and 'differentiation' conditions (Extended Data Fig. 3e). Compared with WT, dKO urothelial cells exhibited significantly increased transwell Matrigel invasion (Fig. 2k). Collectively, these findings suggest that *Kmt2c/d* dKO inhibits differentiation of urothelial cells and augments stem/progenitor potential and induces EMT.

We next investigated transcriptional perturbations induced by *Kmt2c/d* KO, using pooled single-cell transcriptomes (Supplementary Fig. 2a). Gene set enrichment analysis (GSEA) identified immediate early genes (IEGs)[26], inflammation gene sets and antigen presentation gene sets that are upregulated in dKO cells (Fig. 3a,b and Supplementary Fig. 3a). Analyses of 21 representative IEGs in scRNA-seq demonstrated their upregulation in all three dKO cell clusters (Supplementary Fig. 3b,c). IEGs are rapidly transcribed and regulate responses to a wide variety of cellular and extracellular stimuli[27]. Both baseline and epithelial growth factor (EGF)-stimulated expression of three representative IEGs, *Egr3*, *Fosl1* and *Nr4a1* were substantially higher in dKO than WT cells (Fig. 3c,d). We further confirmed upregulation of major histocompatibility complex (MHC) class I surface expression in freshly dissociated nlsEGFP-positive urothelial cells (Fig. 3e and Supplementary Fig. 3d). When propagated in vitro, urothelial cells from dKO mice exhibited higher baseline MHC-I and programmed death ligand 1 (PD-L1, *Cd274*) and further augmented responses to interferon (IFN)-γ stimulation (Fig. 3f). These data suggest that the *Kmt2c/d* dKO context is primed for responses to EGF and IFN-γ stimulation.

Gene sets associated with cellular differentiation such as LUMINAL_MARKERS, HSIAO_LIVER_SPECIFIC_GENES and HALLMARK_ADIPOGENESIS were enriched in WT cells (Fig. 3a and Supplementary Fig. 3a). This pattern of gene expression in *Kmt2c/d* dKO cells is reminiscent of normal adjacent tissues (NAT) to cancers[28]. NAT, unlike cancerous tissue, did not overexpress growth-associated gene sets such

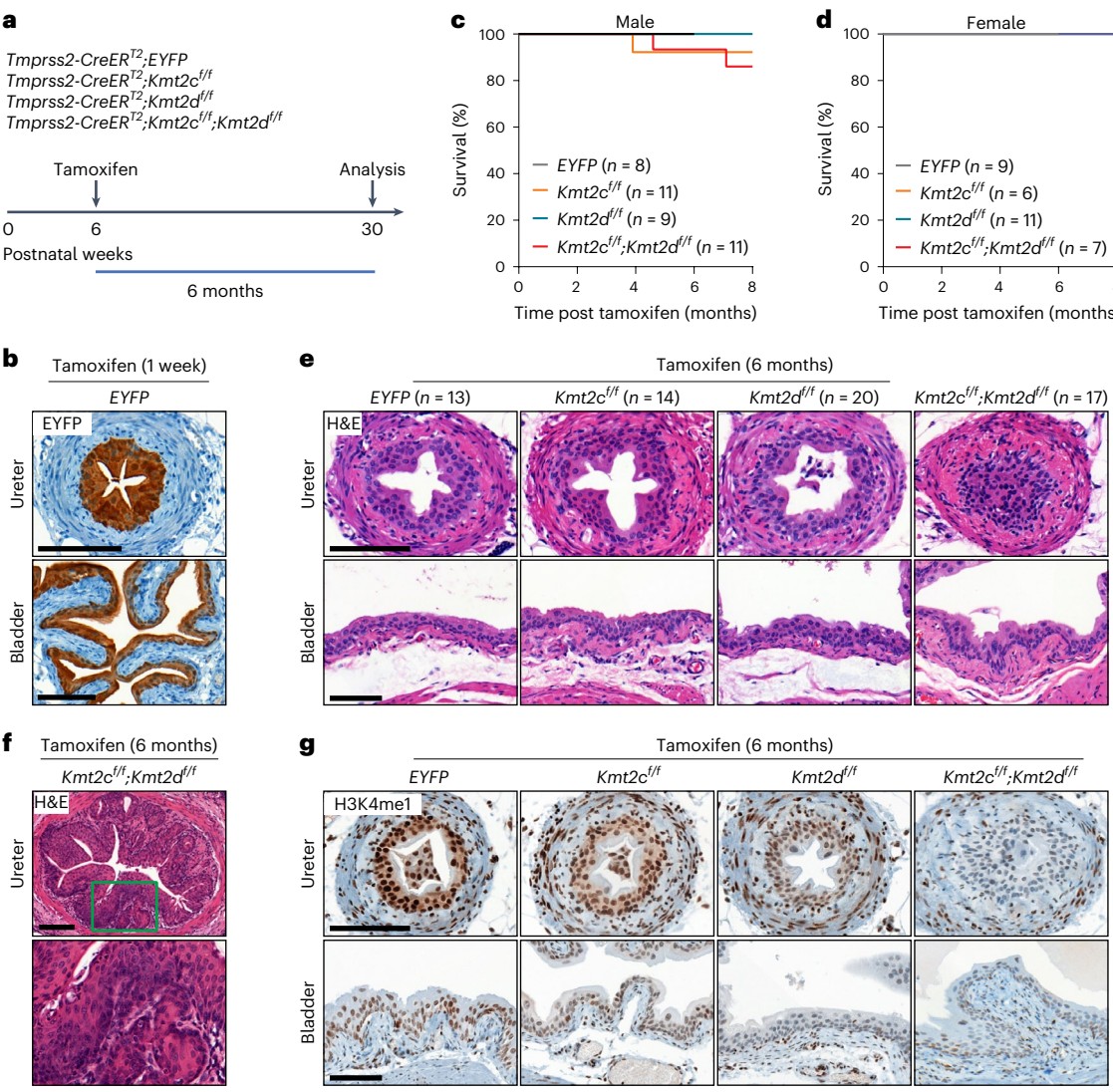

**Fig. 1 | *Kmt2c/d* KO is insufficient to induce robust histological changes in adult mouse urothelium. a**, A schematic of mouse models: two doses of tamoxifen (3 mg ×2) were injected intraperitoneally with a 48 h interval. **b**, IHC using an anti-GFP antibody that recognizes both EGFP and EYFP in the ureter and bladder sections from *Tmprss2-CreER^T2;Rosa26-CAG-LSL-EYFP* mice (*n* = 3 mice). Tissues were collected 1 week after tamoxifen administration. Scale bar, 100 μm. **c,d**, Kaplan–Meier plots showing the survival of male (**c**) and female (**d**) mice after *Kmt2c/d* KO. Dead mice and severely morbid mice requiring immediate euthanasia were both counted as dead cases in this study. **e**, Representative hematoxylin and eosin (H&E) staining of ureter and bladder sections after tamoxifen administration. Scale bar, 100 μm. **f**, Representative H&E staining of ureteral hyperplasia (2 in 17 mice) in *Kmt2c/d* dKO mice. Scale bar, 100 μm. **g**, Representative IHC of H3K4me1 in ureter and bladder tissue sections (*n* = 3 mice in each genotype), validating the successful deletions of *Kmt2c* and/or *Kmt2d*. Scale bar, 100 μm.

as MYC, E2F or G2M. Instead, NAT upregulated gene sets associated with inflammation, antigen presentation and IEGs and downregulated gene sets associated with differentiation. Indeed, GSEA showed that BLCA_NAT_versus_HEALTHY_UP and BLCA_NAT_versus_HEALTHY_DN, two custom gene sets upregulated or downregulated between bladder NAT and healthy bladder[28], were the most positively (1/2,626) and negatively (9/1,730) enriched gene sets, respectively (Fig. 3a,g). These results suggest that *Kmt2c/d* dKO defines a histologically normal but molecularly distinct urothelium that is primed for oncogenic transformation by additional oncogenic mutations.

### *Kmt2c/d* deletion leads to changes in chromatin states

In addition to the KMT2C/D–KDM6A complex that mediates enhancer H3K4 monomethylation, KMT2A/B–menin and SET1A/B–CXXC1 complexes have been characterized to mediate di- and trimethylation of H3K4 primarily at promoters, though their functions at enhancers are increasingly appreciated[10–12,29–31]. We quantified global changes

in H3K4 modifications using mass spectrometry in cultured WT and dKO urothelial cells. We observed the expected reduction of H3K4me1, but surprisingly a significant increase of H3K4me2/3 in *Kmt2c/d* dKO cells (Fig. 4a). The remaining H3K4me1/2/3 were probably catalyzed by KMT2A/B–menin or SET1A/B–CXXC1 complexes.

To investigate the impact of *Kmt2c/d* KO on chromatin states, we performed chromatin immunoprecipitation with sequencing (ChIP-seq) and/or cleavage under targets and release using nuclease (Cut&Run) of H3K4me1, H3K4me2, H3K4me3, H3K27ac, H3K27me3, H3K9me3 and H3K36me3 in WT and dKO urothelial cells (Extended Data Fig. 4a–c). ChromHMM annotated 12 chromatin states (Fig. 4b,c and Extended Data Fig. 4d)[32]. Compared with WT, dKO cells exhibited loss of poised enhancer (state 11, H3K4me1 only) and active enhancer (state 12, H3K4me1/H3K27ac) states, consistent with a loss of H3K4 monomethylation activity. Interestingly, *Kmt2c/d* loss dramatically expanded state 3, characterized by enrichment of both H3K4me1 and H3K4me3 at transcription start site (TSS) flanking regions (Fig. 4c and Extended

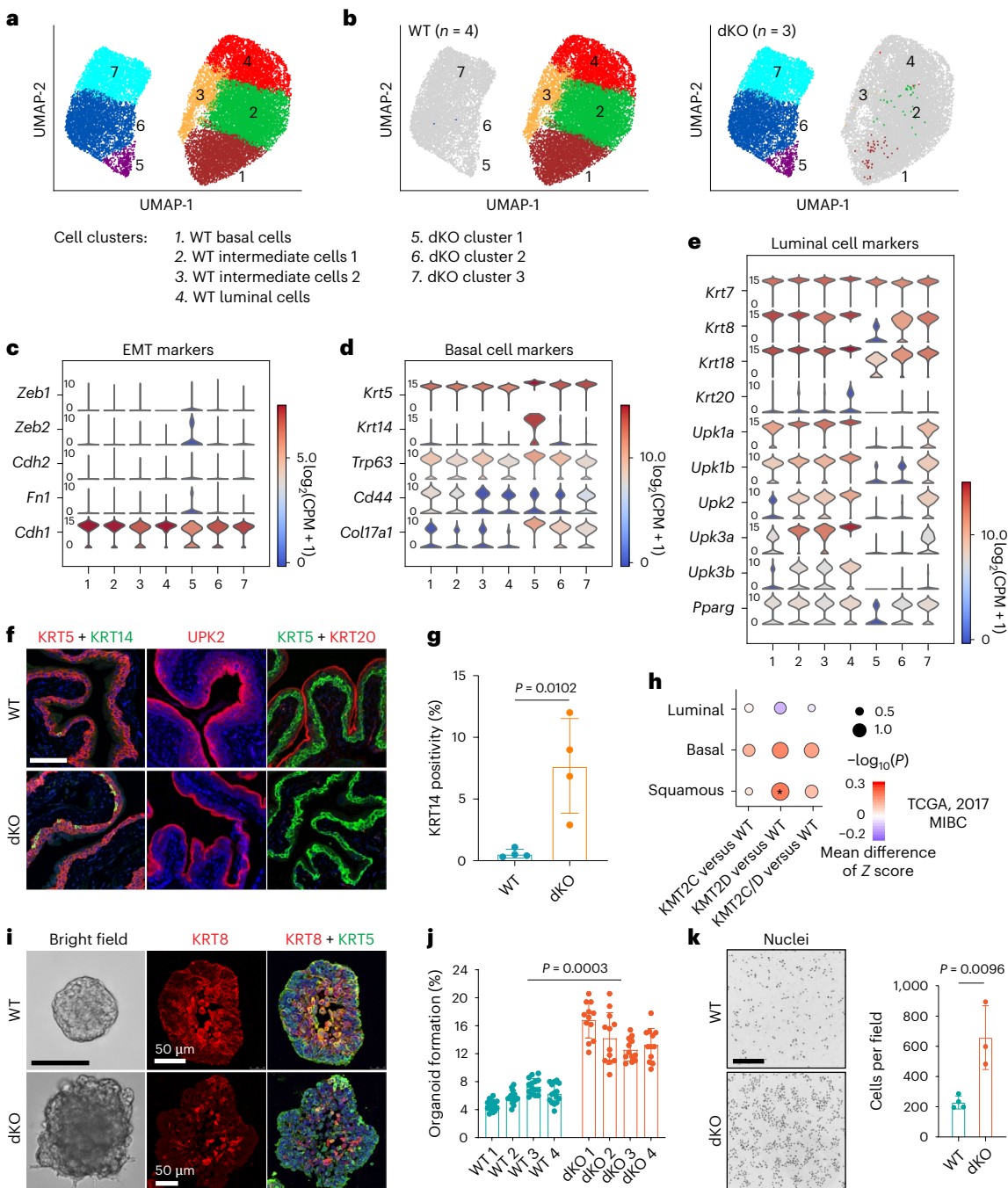

**Fig. 2 | *Kmt2c/d* loss alters stem cell potential, basal differentiation and EMT in adult mouse urothelium. a**, A UMAP plot showing clusters of WT (*n* = 4 mice) and *Kmt2c/d* dKO (*n* = 3 mice) urothelial cells collected 3 months post tamoxifen administration. **b**, UMAP plots showing that 96 in 9,040 cells (1.06%) from the dKO group clustered with WT cells, whereas only 4 in 16,818 cells (0.02%) from the WT group clustered with dKO cells. **c–e**, Violin plots of EMT (**c**), basal (**d**) and luminal (**e**) cell markers. The color in the violin plots indicates the median normalized expression level of genes. **f**, Representative immunofluorescence staining of KRT5, KRT14, UPK2 and KRT20 in WT and *Kmt2c/d* dKO bladder sections. Cell nuclei were counterstained with DAPI (blue). Tissues were collected 6 months after tamoxifen administration. Scale bar, 100 μm. **g**, Quantification of KRT14 positivity in WT (*n* = 4 mice) and dKO (*n* = 4 mice) bladder urothelium. Data are presented as mean ± s.d. and were analyzed with a two-tailed *t*-test. **h**, Enrichment of luminal, basal and squamous markers in the transcriptome of the TCGA MIBC dataset (2017). The non-parametric Wilcoxon rank-sum test was used if one of the sample group was significantly different than the other

sample group. The *P* value is a nominal two-sided *P* value, with \**P* < 0.05. **i**, Representative bright-field image and immunofluorescence staining of KRT5 and KRT8 in organoids from WT (*n* = 4 mice) and dKO (*n* = 4 mice) groups. Cell nuclei were counterstained with DAPI (blue). Scale bar, 100 μm for bright-field images and 50 μm for immunofluorescent images. **j**, Organoid formation efficiency of freshly FACS-sorted urothelial cells from *Tmprss2-CreER^T2^;Kmt2c^f/f^; Kmt2d^f/f^* mice treated with or without tamoxifen (*n* = 4 mice per group). Tissues were collected 3 months post tamoxifen administration. Each point represents one Matrigel blob seeded with 500 cells. Data are presented as mean ± s.d. and were analyzed with a two-tailed *t*-test. **k**, Left: Matrigel invasion assay with fluorescence blocking transwell insert (pore size, 8 μm). Cells were stained with DAPI. Scale bar, 200 μm. Right: quantification of cells invading to the bottom of transwell inserts. Data are shown as mean ± s.d. (*n* = 4 independent assays in the WT group and *n* = 3 independent assays in the dKO group) and were analyzed with a two-tailed *t*-test.

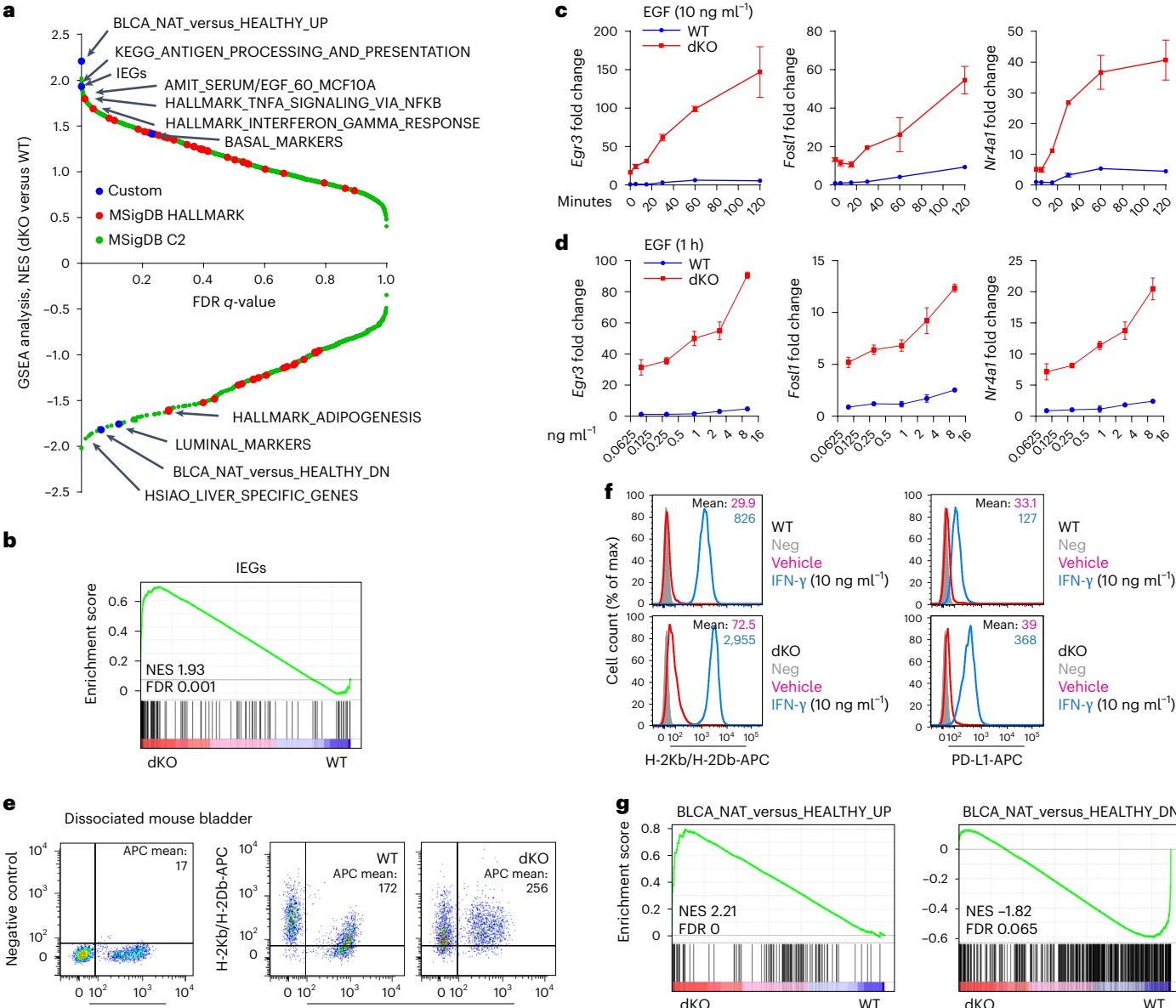

**Fig. 3 | *Kmt2c/d* KO induces an oncogenically primed molecular state characterized by augmented responses to stimuli. a**, A Plot of the normalized enrichment score (NES) versus the FDR *q*-value of GSEA analyses with MSigDB Hallmark v7.4, MSigDB C2 v7.4 and five custom gene sets. BASAL_MARKERS and LUMINAL_MARKERS consist of differentiation markers in Fig. 2h. **b**, GSEA analyses showing enrichment of a previously defined gene set consisting of 139 IEGs in dKO cells. **c,d**, QPCR analysis of *Egr3*, *Fosl1* and *Nr4a1* expression in WT and dKO cells: cells were starved with basic DMEM/F12 medium for 24 h and murine EGF was used to stimulate gene expression in either a time-dependent (**c**) or a dose-dependent (**d**) manner. Data are shown as the mean ± s.d. (representative of *n* = 3 independent experiments). **e**, Flow cytometry of MHC class I molecules

H-2Kb and H-2Db in freshly dissociated urothelial cells from WT and dKO mice (3 months post tamoxifen administration, *n* = 4 mice in each group). Urothelial cells are nlsEGFP positive. **f**, Flow cytometry of MHC class I molecules H-2Kb and H-2Db in cultured urothelial cells (*n* = 4 independent experiments). To induce the expression of H-2Kb/Db, cells were treated with vehicle or mouse IFN-γ (10 ng ml⁻¹) for 24 h. Neg, APC-conjugated isotype antibodies used as negative control. **g**, GSEA comparison of single-cell transcriptomes and human bladder gene sets. Gene sets consisted of genes differentially expressed between NAT (*n* = 19) and healthy human bladder tissues (*n* = 11), NAT versus healthy, absolute fold change >5, *P* < 0.05.

Data Fig. 4d). The mass spectrometry and ChromHMM results suggest that *Kmt2c/d* KO leads to loss of H3K4me1 at enhancers but a paradoxical gain of H3K4 methylation at promoters, which we speculated to be catalyzed by KMT2A/B–menin and/or SET1A/B–CXXC1 complexes.

### *Kmt2c/d* loss decreases deposition of KMT2 complexes at enhancers

To determine the genomic distribution of KMT2C/D–KDM6A, KMT2A/B–menin and SET1A/B–CXXC1 complexes, we performed ChIP-seq

or Cut&Run against representative KMT2 components: KMT2D, KMT2A, menin, SET1A and CXXC1 in WT and dKO urothelial cells. We further quantified chromatin accessibility using the assay for transposase-accessible chromatin sequencing (ATAC-seq) and nascent transcriptional activity at promoter and enhancer regions using precision nuclear run-on followed by cap-selection sequencing (PRO-cap)[33]. As expected, KMT2D preferentially bound to promoter-distal regions (12,399 peaks, 93.1%)[29] and to a small number of promoters (920 peaks, 6.9%) (Fig. 4d). KMT2A and menin bound to promoters and many

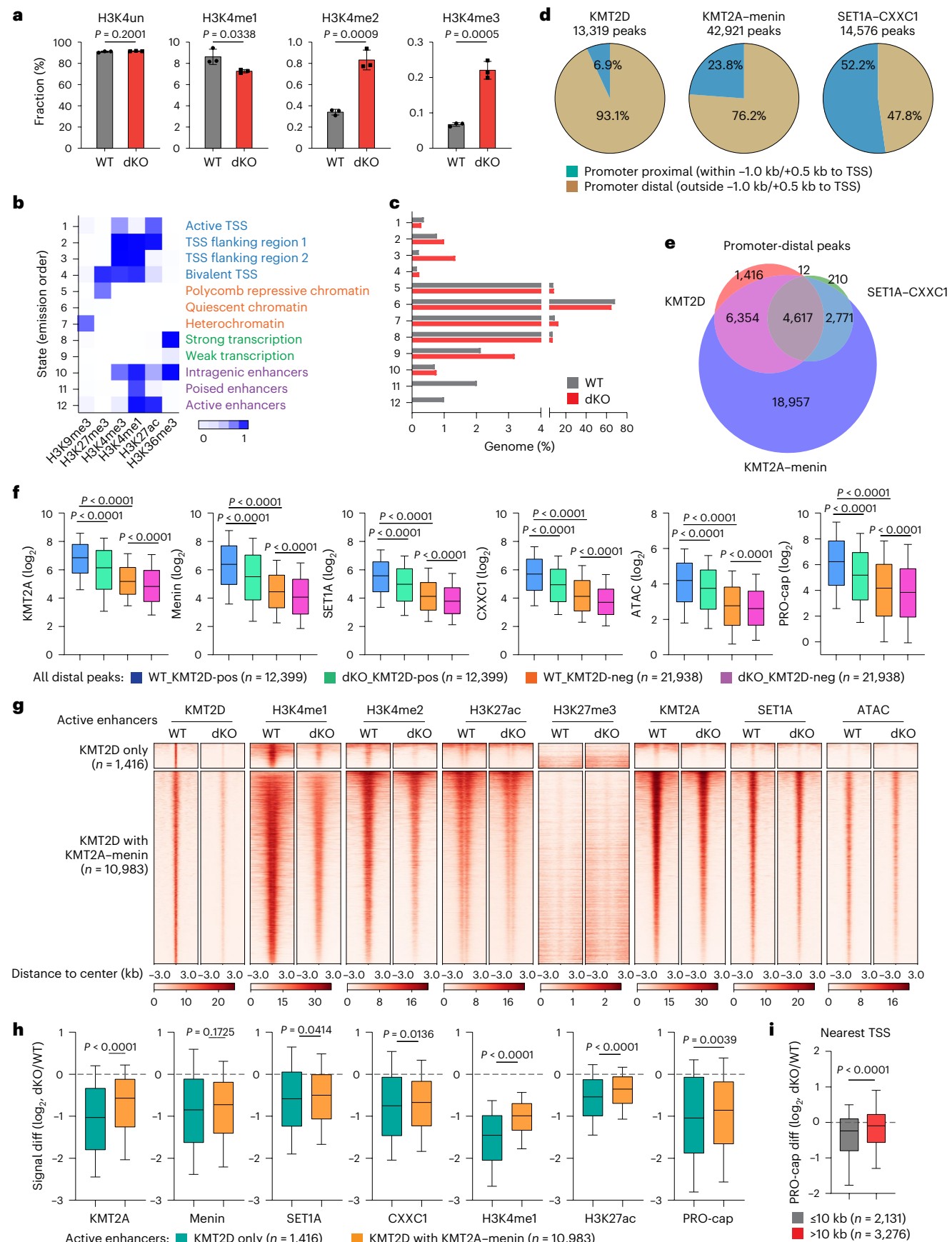

**Fig. 4 | *Kmt2c/d* deletion suppresses enhancer activity and decreases KMT2A/ SET1A deposition at active enhancers. a**, Mass spectrometry comparing the fractions of H3K4 modifications in WT and dKO cells (*n* = 3 independent experiments). Data are shown as mean ± s.d. and were analyzed with a two-tailed *t*-test. **b**, Comparison of 12 chromatin states in WT and dKO groups using ChromHMM annotation. A darker blue color corresponds to a higher probability of observing specific modifications in each state. Annotations of each state are shown on the right side of the heat map. **c**, The genomic fraction of each chromatin state in WT and dKO urothelial cells. **d**, The genomic distributions of KMT2D, KMT2A−menin and SET1A−CXXC1 peaks in WT or dKO urothelial cells. Note that peaks within −1.0 kb/+0.5 kb to the TSS were annotated as promoter proximal and the remaining peaks were annotated as promoter distal. **e**, The overlap of KMT2D, KMT2A−menin and SET1A−CXXC1 peaks on promoter-distal regions. **f**, KMT2A, menin, SET1A, CXXC1, ATAC-seq and PRO-cap signal at KMT2D-positive (pos) and KMT2D-negative (neg) enhancers in WT and dKO urothelial cells. The center line represents the median, the box limits represent the upper and lower quartiles and the minimum and maximum whiskers represent the 10th and 90th percentiles, respectively. Data were analyzed with a two-tailed *t*-test. **g**, A heat map showing the enrichment of KMT2D, H3K4me1, H3K4me2, H3K27ac, H3K27me3, KMT2A, SET1A and ATAC signal at two subgroups of KMT2D-bound enhancers. Data are shown as the average of replicate samples. **h**, The log$_2$ fold change of KMT2A, menin, SET1A, CXXC1, H3K4me1, H3K27ac and PRO-cap signal (signal diff) at two subgroups of KMT2D-bound enhancers. The center line represents the median, the box limits represent the upper and lower quartiles and the minimum and maximum whiskers represent the 10th and 90th percentiles, respectively. Data were analyzed with a two-tailed *t*-test. **i**, The log$_2$ fold change of PRO-cap signal (PRO-cap diff) at active TSS containing a nearest enhancer within or outside ±10 kb. For our analysis, duplicated enhancers were removed as we kept only the closest enhancer to each TSS. The center line represents the median, the box limits represent the upper and lower quartiles and the minimum and maximum whiskers represent the 10th and 90th percentiles, respectively. Data were analyzed with a two-tailed *t*-test.

promoter-distal regions, and their binding sites and peak intensities were concordant. SET1A and CXXC1 preferentially bound to promoters and similarly exhibited high concordance (Fig. 4d and Extended Data Fig. 4e–g).

We defined 'enhancers' as promoter-distal regions that were enriched for KMT2A, menin, SET1A, CXXC1 or KMT2D signal in either the WT or dKO groups (Fig. 4e). We asked whether enhancers bound by KMT2D were functionally distinct. We found that KMT2D-positive enhancers (*n* = 12,399) exhibited significantly higher baseline (WT cells) H3K4me1, H3K4me2, H3K27ac, ATAC and PRO-cap signals compared with KMT2D-negative enhancers (*n* = 21,938), suggesting that KMT2D marked more active enhancers (Fig. 4f, Supplementary Fig. 4a–c and Supplementary Table 1). All these signals were significantly decreased at KMT2D-positive enhancers of dKO cells (Fig. 4f), indicating that KMT2C/D is involved in activation of these enhancers.

Interestingly, we also observed extensive localization of the KMT2A/B−menin and SET1A/B−CXXC1 complexes at KMT2D-positive enhancers. Thus, we further categorized KMT2D-bound enhancers into two classes: class 1 with KMT2D only (*n* = 1,416) and class 2 with KMT2A−menin (*n* = 10,983). Some class 2 enhancers were also bound by SET1A−CXXC1 (Fig. 4e). While both classes of enhancers exhibited similar levels of KMT2D and H3K4me1 at baseline, the baseline levels of H3K4me2, H3K27Ac and ATAC were higher in class 2 enhancers (Fig. 4g and Supplementary Fig. 4d). Class 2 enhancers exhibited a smaller reduction of H3K4me1, H3K4me2, H3K27Ac, ATAC and PRO-cap signals with *Kmt2c/d* loss (Fig. 4g,h and Supplementary Figs. 4d and 5a). These data suggest that while the KMT2C/D−KDM6A complex regulates many enhancers, only a small subset (class 1) is hyperdependent on this complex. The SET1A/B−CXXC1 and KMT2A/B−menin complexes may partially compensate at class 2 enhancers, consistent with a recent observation in embryonic stem cells[34]. Notably, we observed decreased signal of SET1A−CXXC1 and KMT2A−menin at class 2 enhancers after

*Kmt2c/d* KO (Supplementary Fig. 4d and 5a), suggesting that KMT2C/D may help recruit the other H3K4 methyltransferase complexes.

To assess the effect of *Kmt2c/d* loss on enhancer function, we analyzed the change of expression on the nearest gene promoter to KMT2D-positive enhancers. We found significant downregulation of PRO-cap signal at the nearest TSS to the KMT2D-positive enhancer within ±10 kb (Fig. 4i). The loss of enhancer and adjacent promoter activities were demonstrated by two representative genes (*Fgfr3* and *Upk3bl*) of urothelium differentiation (Supplementary Fig. 6a–c).

## *Kmt2c/d* directly regulates activity of CpG-poor promoters

Active promoters and enhancers share many similarities including a central nucleosome depleted region and bidirectional transcription. High CpG content is one distinguishing feature of many promoters[35]. The SET1A/B−CXXC1 complexes and the KMT2A/B−menin complexes are both recruited to CpG islands via the CXXC1 subunit and via the CXXC domain of KMT2A/B, respectively[36]. We next characterized promoters binding by KMT2D, KMT2A−menin and SET1A−CXXC1. We found a substantial overlap between KMT2A−menin and SET1A−CXXC1 binding at the majority of active promoters (Fig. 5a), consistent with prior observations[37]. KMT2D bound to a subset of promoters (*n* = 920), with a small number bound only by KMT2D (*n* = 38) (Fig. 5a). Promoters bound by KMT2A−menin or SET1A−CXXC1 had higher CpG content whereas promoters bound by KMT2D had lower CpG content (Fig. 5b). After *Kmt2c/d* KO, KMT2D-bound promoters exhibited significantly decreased expression by PRO-cap, as well as decreased H3K4me1 and H3K4me3 and increased H3K27me3 enrichment (Fig. 5c).

We next examined the ChromHMM chromatin state of the top 500 up- and downregulated promoters after *Kmt2c/d* loss. Compared with all TSS, promoters of genes downregulated after *Kmt2c/d* deletion were less enriched for state 1 (H3K4me3 and H3K27ac) and more enriched for states 2 and 3 that contained both H3K4me3 and H3K4me1, and

**Fig. 5 | *Kmt2c/d* loss suppresses activities of KMT2D-bound TSS and redistributes KMT2A−menin to CpG-high promoters. a**, The overlap of KMT2D, KMT2A−menin and SET1A−CXXC1 peaks on promoter-proximal regions. **b**,**c**, The fraction of CpG dinucleotide (**b**) and log$_2$ fold changes (**c**) of PRO-cap, H3K4me1, H3K4me3 and H3K27me3 signal in three subgroups of active TSS. The center line represents the median, the box limits represent the upper and lower quartiles and the minimum and maximum whiskers represent the 10th and 90th percentiles, respectively. Data were analyzed with a two-tailed *t*-test. **d**, Fold enrichment over the genome of chromatin states at all TSS and the top 500 PRO-cap up- (Up) or downregulated (Dn) TSS. **e**, A dot plot showing H3K4me3 and H3K27me3 modifications at all TSS in WT urothelial cells. The signal of H3K4me3 and H3K27me3 were normalized with RPGC. The red points indicate TSS with upregulated PRO-cap signal with *Kmt2c/d* KO (top 500). The gray points indicate all remaining TSS. Here we defined TSS with H3K4me3 ≥4 and H3K27me3 <3 as H3K4me3 only, while TSS with H3K4me3 ≥4 and H3K27me3 ≥3 as bivalent.

**f**, Left: the log$_2$ fold change of PRO-cap signal at H3K4me3 only (*n* = 8,152) and bivalent (*n* = 1,756) TSS. Middle: the fraction of CpG. Right: the log$_2$ fold change of KMT2A enrichment. The center line represents the median, the box limits represent the upper and lower quartiles and the minimum and maximum whiskers represent the 10th and 90th percentiles, respectively. Data were analyzed with a two-tailed *t*-test. **g**, A heat map showing the changes of KMT2A, SET1A, H3K4me1, H3K4me2, H3K4me3 and H3K27ac between the WT and dKO groups. Data are shown as the average of replicate samples. The purple color indicates signal upregulated in dKO cells, while cyan indicates signal downregulated in dKO cells. **h**, The log$_2$ fold change of KMT2A, menin, SET1A, CXXC1 and PRO-cap with *Kmt2c/d* KO. The center line represents the median, the box limits represent the upper and lower quartiles and the minimum and maximum whiskers represent the 10th and 90th percentiles, respectively. Data were analyzed with a two-tailed *t*-test. **i**, A dot plot of each TSS with linear correlation of KMT2A and PRO-cap alterations (log$_2$ fold change, dKO/WT) at active TSS. Linear regression coefficient, $R^2$ = 0.4807.

especially state 12 with H3K4me1, H3K27ac and devoid of H3K4me3, typical of active enhancers (Figs. 4b and 5d and Extended Data Fig. 4d). Consistently, genes whose promoters overlapped with H3K4me1 but not H3K4me3 peaks were highly enriched for downregulated genes (Extended Data Fig. 5a,b). This correlated with a significantly

lower CpG content at the promoters of the top 500 downregulated genes (Extended Data Fig. 5c). Examination of a running average of PRO-cap change at promoters and CpG content showed that genes bound by KMT2D and with low CpG content exhibited decreased levels upon *Kmt2c/d* KO (Extended Data Fig. 5d). GSEA of PRO-cap and

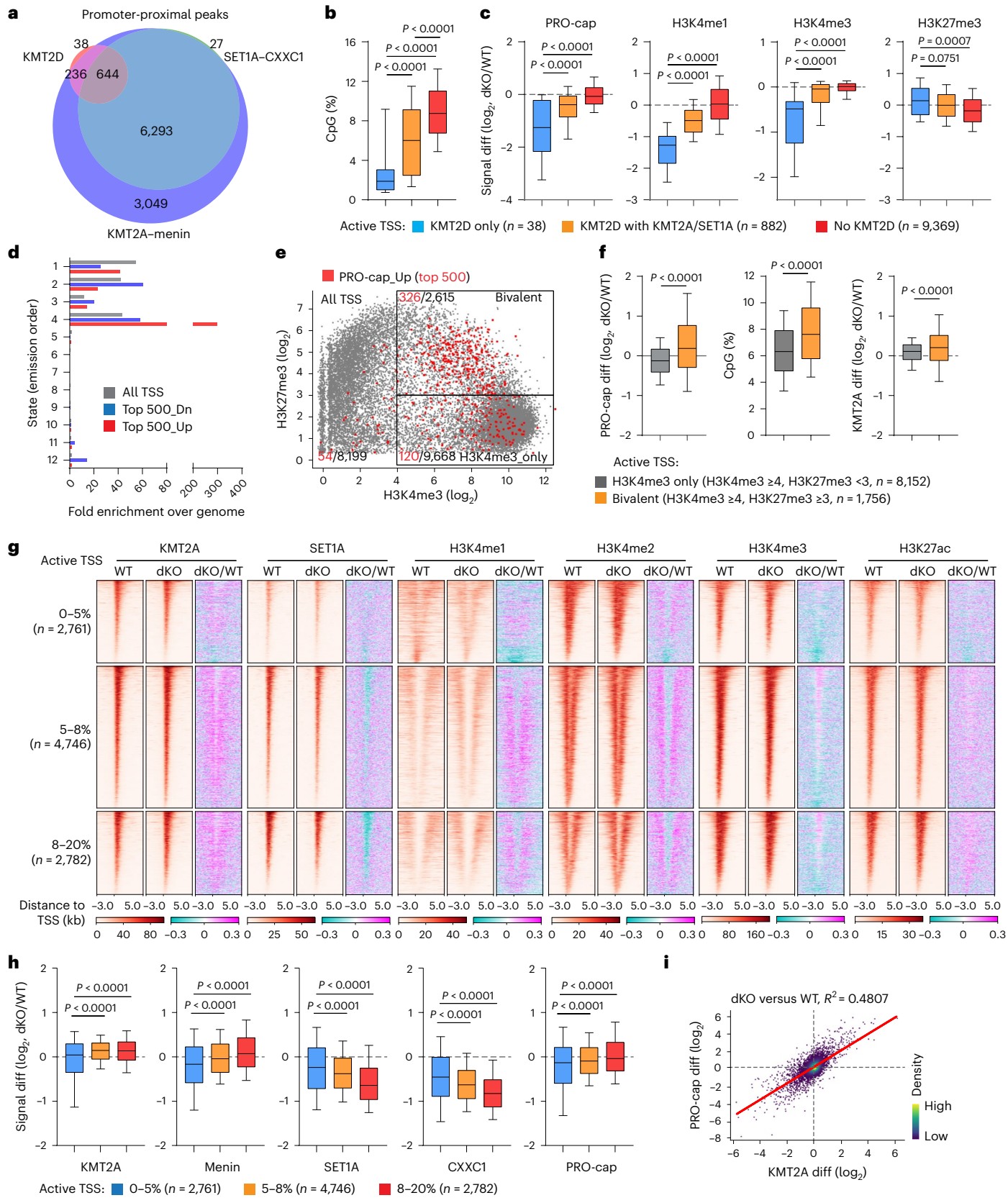

RNA-seq showed that the gene set MIKKELSEN_ES_LCP(low CpG)_WITH_H3K4ME3 was among the most negatively enriched gene sets for both (Extended Data Fig. 5e,f). Gene Ontology analyses of the top 500 downregulated TSS showed significant enrichment of epithelial development and differentiation (Extended Data Fig. 5g), consistent with the observation that CpG-low promoters are preferentially associated with tissue-specific genes[38,39].

Next, we compared the gene expression of *KMT2C/D* mutant (*n* = 160) versus WT (*n* = 248) bladder cancers from the TCGA RNA-seq dataset[6]. We observed that genes with low-CpG promoters were downregulated in *KMT2C/D* mutant cancers (Extended Data Fig. 5h). In summary, these data identify a group of low-CpG genes bound by KMT2C/D whose expression is directly regulated by KMT2C/D–KDM6A complexes.

### *Kmt2c/d* loss leads to KMT2A redistribution to bivalent promoters

Our data suggest that one mechanism by which *Kmt2c/d* loss primes urothelium is through upregulation of IEGs and inflammatory genes. These genes exhibit high CpG content and bivalent chromatin markers (both H3K4me3 and H3K27me3) at their promoters[27,40–42]. We examined the H3K4me3 and H3K27me3 signal at the TSS of the top 500 upregulated genes. We found significant enrichment of bivalent TSS (326/2,615) in the upregulated genes compared with the remaining active genes (120/9,668) (two-tailed Chi-squared test, *P* < 0.0001) (Fig. 5e,f). Using the traditional definition of bivalent TSS as overlap with both H3K4me3 and H3K27me3 peaks, we found upregulation of bivalent genes and no change in H3K4me3-only genes in dKO cells (Extended Data Fig. 5a,b). ChromHMM showed that the promoters of the top 500 upregulated genes were highly enriched for state 4 (bivalent) (Fig. 5d), consistent with high CpG content, a known feature of bivalent promoters (Extended Data Fig. 5c). GSEA showed that gene sets associated with H3K27me3 were among the most positively enriched gene sets in dKO cells (Extended Data Fig. 5i,j). In human bladder cancer, gene sets associated with H3K27me3 and PRC2 (SUZ12/EED) targets were also positively enriched in *KMT2C/D* mutant samples (Extended Data Fig. 5k).

To understand the basis of transcriptional upregulation at high-CpG genes, we categorized active TSS into three subgroups based on CpG content. Compared with CpG-low TSS (0–5%, *n* = 2,761), we observed increased H3K4me1, H3K4me2 and H3K4me3 at CpG-high TSS with *Kmt2c/d* KO (8–20%, *n* = 2,782) (Fig. 5g and Supplementary Fig. 7a). This increase is consistent with the overall increase of H3K4me2 and H3K4me3 marks from mass spectrometry data (Fig. 4a). We observed increased KMT2A–menin but not SET1A–CXXC1 at active high-CpG promoters (Fig. 5f–h and Supplementary Fig. 7b,c). There was a strong correlation between changes in KMT2A promoter binding and changes in gene expression, suggesting that KMT2A–menin redistribution may underlie gene upregulation after *Kmt2c/d* loss (Fig. 5i). IEGs and MHC-I components were more likely to be bivalent, exhibited high CpG content and increased KMT2A–menin binding after *Kmt2c/d* KO (Supplementary Fig. 7d). This observation is consistent with the

reported role of the KMT2A/B–menin complex in regulation of bivalent promoters[43,44]. We confirmed the epigenetic alterations at TSS regions in selective genes, including basal markers (for example, *Krt16*), EMT markers (for example, *Zeb2*), IEGs (for example, *Nr4a1*) and MHC-I components (for example, *B2m* and *H2-K1*) (Supplementary Fig. 8a–f).

Our data suggest a model where *Kmt2c/d* loss leads to redistribution of KMT2A–menin from class 2 enhancers to CpG-high promoters, leading to transcriptional upregulation. We tested whether blockade of the KMT2A/B–menin complex may reverse gene upregulation in dKO urothelial cells. We treated *Kmt2c/d* dKO urothelial cells with the menin inhibitor MI-503 (1 μM) or dimethylsulfoxide (DMSO) for 4 days. MI-503 decreased menin deposition at all active TSS (Extended Data Fig. 6a). GSEA of RNA-seq showed that MI-503 treatment reversed the expression changes induced by *Kmt2c/d* deletion, where *dKO_versus_WT_*DN was positively enriched and *dKO_versus_WT_*UP was negatively enriched (Extended Data Fig. 6b–d). MI-503 downregulated gene sets associated with growth factor signaling and with basal differentiation that were upregulated in dKO cells (Fig. 3 and Extended Data Fig. 6b–d). Functionally, we found that MI-503 treatment partially rescued the phenotypes of basal differentiation and transwell invasion in dKO urothelial cells (Extended Data Fig. 6e,f). These data suggest that loss of *Kmt2c/d* leads to activation of a transcriptional program through KMT2A–menin redistribution.

### *Kmt2c/d* deletion primes tumorigenesis to carcinogen and oncogenes

We assessed whether *Kmt2c/d* KO in the urothelium may confer sensitivity to tumorigenic transformation. *PTEN* is mutated or homozygously deleted in ~6% and heterozygously lost in another ~37% of MIBCs (Extended Data Fig. 7a)[6]. Hence, we crossed a conditional *Pten* mice to the *Kmt2c/d* conditional mice (Fig. 6a). We confirmed deletions of *Kmt2c/d* by BaseScope and loss of H3K4me1 and PTEN by IHC (Supplementary Fig. 9a–c). Mice with *Pten* deletion alone were viable for the 8 month duration of the experiment. In contrast, mice with *Pten* deletion combined with *Kmt2c* and/or *Kmt2d* deletion showed decreased survival (Fig. 6b and Supplementary Table 2). In *Kmt2c^{f/f};Pten^{f/f}* and *Kmt2d^{f/f};Pten^{f/f}* male mice, the mortality was caused by renal failure due to obstruction from urothelial tumors (Fig. 6b and Supplementary Tables 2 and 3). Male mice of these two genotypes exhibited increased mortality (Fig. 6b), reminiscent of the sex dichotomy in human urothelial carcinoma[45]. *Kmt2c^{f/f};Kmt2d^{f/f};Pten^{f/f}* mice suffered early mortality due to stomach cancer, and exhibited dehydration, diarrhea, hematochezia and hunching (Supplementary Table 2).

We performed histologic evaluation of bladders and ureters 6 months after tamoxifen administration, except for *Kmt2c^{f/f};Kmt2d^{f/f};Pten^{f/f}* mice that were analyzed at the 6 week time point (Fig. 6c–f and Extended Data Fig. 8a,b). *Pten* deletion alone caused thickening of the bladder and ureteral urothelium without overt atypia. When combined with deletion of *Kmt2c* and/or *Kmt2d*, the bladder urothelium exhibited dysplasia that progressed to nuclear pleomorphism and abnormal mitoses fulfilling the criteria of carcinoma in situ (CIS) in some mice (Fig. 6d,f). The ureters exhibited a more severe phenotype,

**Fig. 6 | *Kmt2c/d* KO cooperates with *Pten* loss to induce invasive urothelial carcinoma in GEMMs. a**, A schematic of mouse models: two doses of tamoxifen (3 mg ×2) were injected intraperitoneally with a 48 h interval. **b**, Kaplan–Meier plots showing the survival of male and female mice after gene knock out. Dead mice and severely morbid mice requiring immediate euthanasia were both counted as dead cases in this study. **c**, Representative histological staining of H&E (*Pten^{f/f}*, *n* = 16 mice; *Kmt2c^{f/f};Pten^{f/f}*, *n* = 23 mice; *Kmt2d^{f/f};Pten^{f/f}*, *n* = 26 mice; *Kmt2c^{f/f};Kmt2d^{f/f};Pten^{f/f}*, *n* = 17 mice) and Ki-67 IHC in mouse ureter sections. Scale bar, 500 μm in the low-power images and 100 μm in the zoomed-in images. **d**, Representative histological staining of H&E (*Pten^{f/f}*, *n* = 16 mice; *Kmt2c^{f/f};Pten^{f/f}*, *n* = 23 mice; *Kmt2d^{f/f};Pten^{f/f}*, *n* = 26 mice; *Kmt2c^{f/f};Kmt2d^{f/f};Pten^{f/f}*, *n* = 17 mice) and Ki-67 IHC in mouse bladder sections. Scale bar, 5 mm in the low-power

images and 200 μm in the zoomed-in images. **e**, Quantification of Ki-67-positive cells in mouse bladder and ureter tissues collected 6 months after tamoxifen administration. Note that the ureter tissues from both male and female *Kmt2c^{f/f};Kmt2d^{f/f};Pten^{f/f}* mice were collected 6 weeks post tamoxifen administration. Each dot indicates the Ki-67 positivity from multiple sections in one mouse. Data are presented as mean ± s.d. and were analyzed with a two-tailed *t*-test between *EYFP* and each indicated group. NA, not analyzed. **f**, Histological subtypes of bladder urothelium in male (M) and female (F) GEMMs. **g**, A schematic illustration of urothelial carcinoma models induced by BBN, (*n* = 20 mice in the WT group and *n* = 18 mice in the dKO group). **h**, Histological subtypes of BBN-induced bladder urothelial carcinoma (UC) in male and female mice.

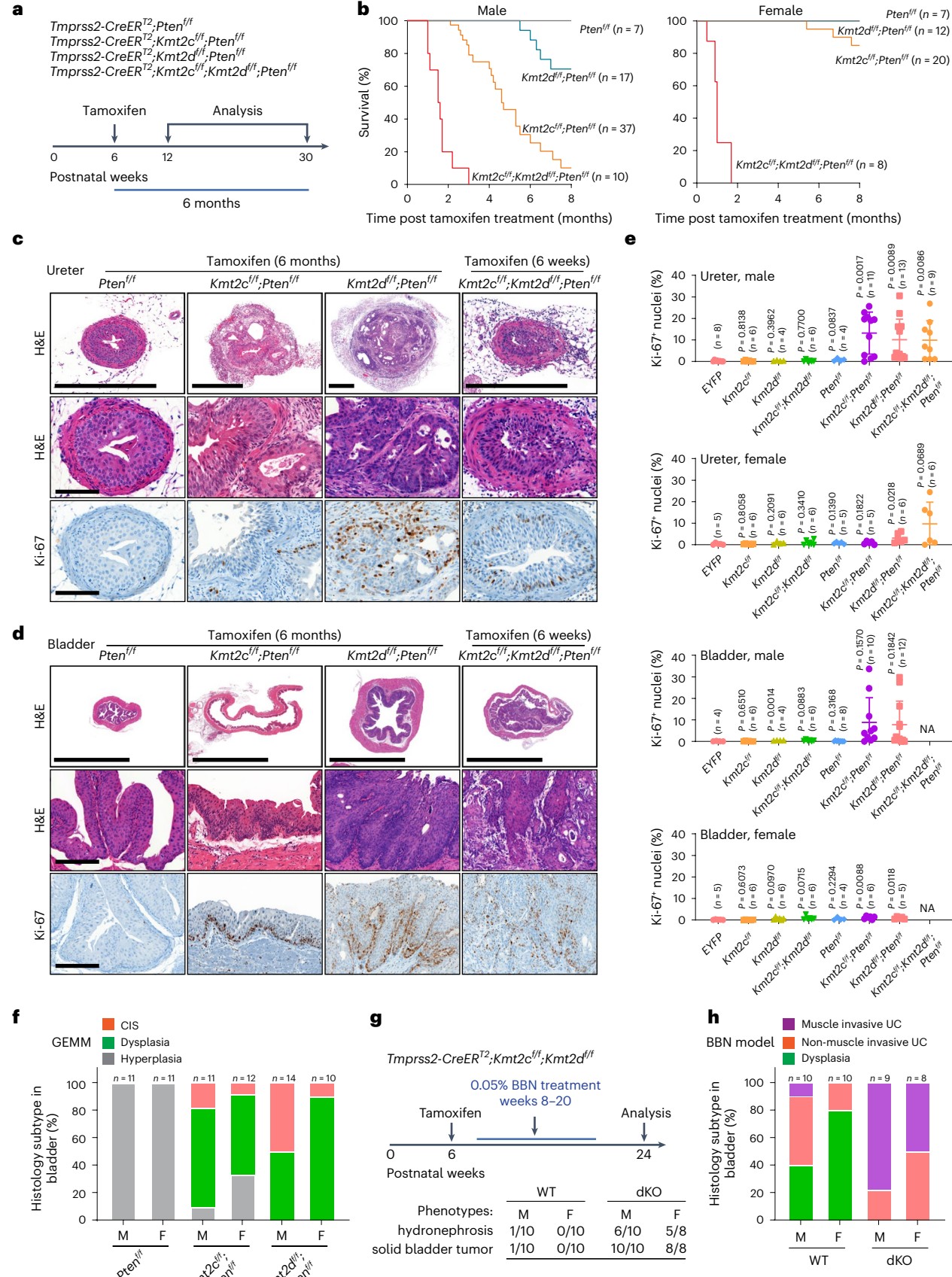

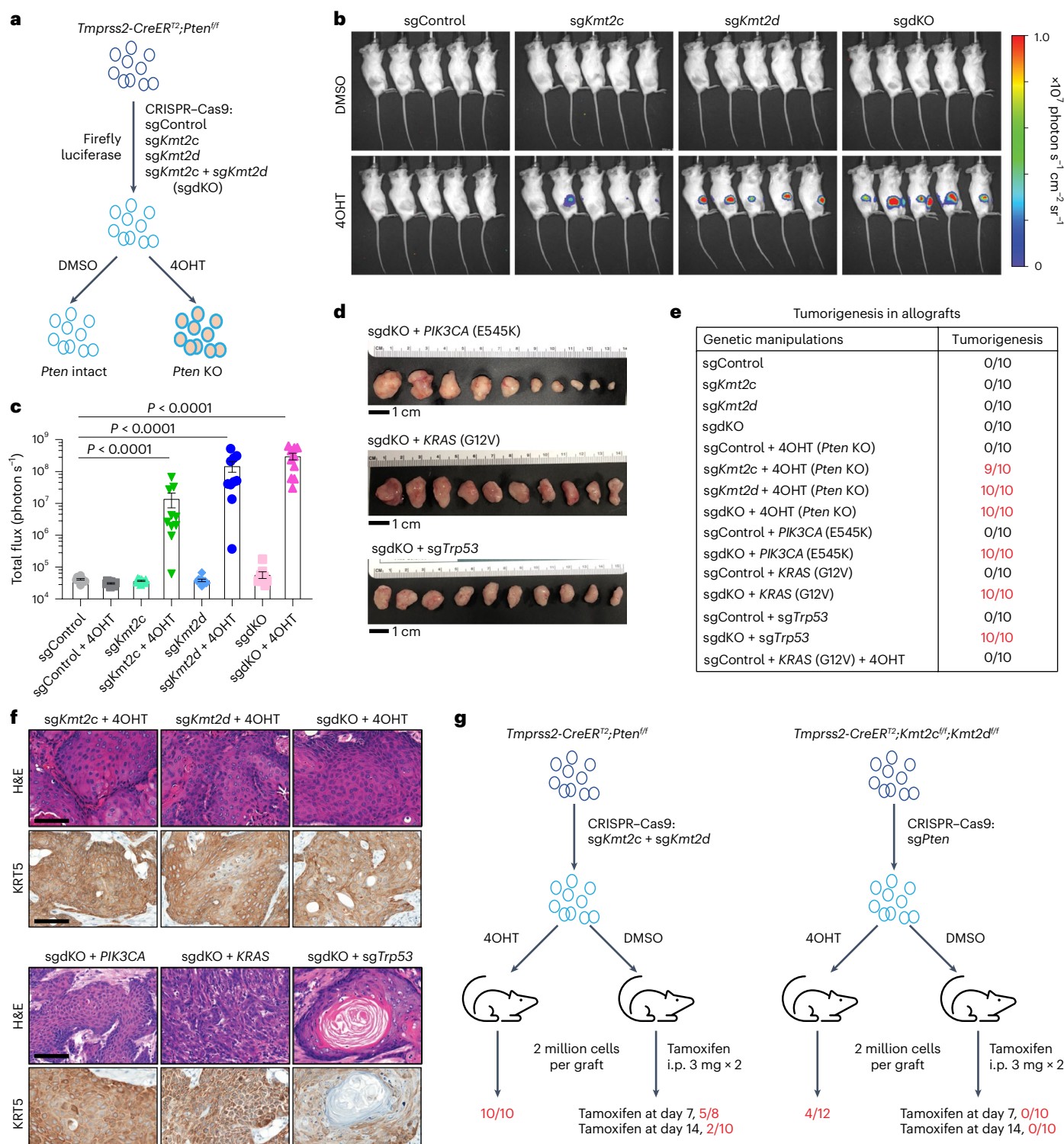

**Fig. 7 | *Kmt2c/d* deletion primes tumorigenic susceptibility to prevalent oncogenic drivers. a**, A schematic illustration of CRISPR–Cas9 KO of *Kmt2c*, *Kmt2d* or *Kmt2c* + *Kmt2d* (sgdKO) in urothelial organoid derived from *Tmprss2-CreER^{T2};Pten^{f/f}* mouse. To knock out *Pten*, cells were treated with 4OHT (0.2 μM) for 24 h. **b**, Bioluminescent imaging of mammary fat pad allografts in SCID mice (*n* = 10 grafts in each group, 5 weeks post grafts). Images are shown at the same range of scale bar. **c**, Mammary fat pad allografts of urothelial organoids with CRISPR–Cas9 KO of *Kmt2c* and/or *Kmt2d* (*n* = 10 grafts per group). Urothelial cells (2 million) were grafted bilaterally into mammary fat pad of NOD-SCID mice. Luminescent signals were examined 5 weeks after grafting. Data are presented as

mean ± s.e.m. Statistical comparisons were performed with a two-tailed *t*-test on log₁₀ normalized data. **d**, Mammary fat pad allografts of dKO urothelial organoids with perturbations of *PIK3CA*^{E545K}, *KRAS*^{G12V} and *Trp53*. Scale bar, 1 cm. **e**, Summary of tumorigenesis in mammary fat pad allografts. In non-tumorigenic groups, no palpable tumor was observed at least 2 months after the grafting. **f**, Representative H&E and KRT5 IHC staining of tumor sections from mammary fat pad allografts (*n* = 4 tumors per group). Scale bar, 100 μm. **g**, Tumorigenesis in allografts by ordering the time sequence of *Kmt2c/d* and *Pten* deletions in urothelial cells. i.p., intraperitoneal.

with evidence of muscle invasion in more than half the mice (Fig. 6c and Extended Data Fig. 8a,b). Male mice developed carcinoma of the prostatic urethra (Extended Data Fig. 8c). The urethral carcinoma was positive for KRT5 and KRT7, consistent with urothelial origin. For both *Kmt2c*$^{f/f}$;*Pten*$^{f/f}$ and *Kmt2d*$^{f/f}$;*Pten*$^{f/f}$, male mice exhibited more advanced pathology and higher Ki-67 positivity (Fig. 6e,f and Extended Data Fig. 8b). We had a very limited number of *Kmt2c*$^{f/f}$;*Kmt2d*$^{f/f}$;*Pten*$^{f/f}$ mice but at 6 weeks, they already exhibited muscle-invasive urothelial carcinoma of the ureter (Fig. 6c–e). Consistent with the basal differentiation in *Kmt2c/d* KO urothelium, the urothelial carcinoma models exhibited basal phenotype, corroborated by KRT5 positivity and loss of UPK2 by IHC (Extended Data Fig. 8d).

Since *Tmprss2-CreER*$^{T2}$ activity is not restricted to the urothelial epithelium, we restricted the deletion of *Kmt2c/d* and *Pten* to the bladder epithelium via intravesical delivery of 4-hydroxytamoxifen (4OHT) or adenoviral delivery of Cre recombinase driven by cytomegalovirus (CMV) (adeno-CMV-Cre), *Krt5* (adeno-K5-Cre) or *Krt8* (adeno-K8-Cre) in *Tmprss2-CreER*$^{T2}$;*Kmt2c*$^{f/f}$;*Kmt2d*$^{f/f}$;*Pten*$^{f/f}$ mice[46]. Despite the variable tumorigenic efficiency, we observed induction of muscle-invasive urothelial cancer at 3 months post injection (Extended Data Fig. 9a–d). We observed similar phenotypes as systemic tamoxifen administration in these tumors, for example, increased basal differentiation and decreased luminal differentiation (Extended Data Fig. 9a,e). We found no tumors in the adeno-K8-Cre mice, which could be due to the low infection rate or that basal cells are the cell of origin for bladder urothelial cancers[22,46].

BBN, a tobacco-associated carcinogen, is widely used to generate carcinogen-induced bladder cancer in preclinical mouse models[15]. Two weeks following tamoxifen-mediated deletion of *Kmt2c/d* (dKO) and WT controls (no tamoxifen) (Fig. 6g), we treated the mice with 0.05% BBN drinking water for 12 weeks and then analyzed the mice 4 weeks later. In the BBN-treated WT group, 12/20 mice developed dysplasia, 7/20 developed non-invasive urothelial carcinoma and one male mouse developed invasive urothelial carcinoma with unilateral hydronephrosis (Fig. 6g,h and Supplementary Fig. 10a,b). In contrast, 7/9 male and 4/8 female mice in the dKO group developed invasive bladder tumors, with >50% of both male and female mice exhibiting bilateral hydronephrosis. Notably, invasive ureteral urothelial carcinoma was observed in 1/9 male and 4/8 female mice in the dKO group, whereas the ureters of the WT mice were histologically normal (Fig. 6g and Supplementary Fig. 10b,c). We observed high KRT5 and loss of UPK2 expression in the dKO groups (Supplementary Fig. 10d).

To investigate whether *Kmt2c/d* KO increases tumorigenic susceptibility to other common recurrent oncogenic mutations in human urothelial carcinoma, we used the organoid culture system to engineer KOs or transgenes (Fig. 7a). We confirmed the cooperativity of *Pten* deletion with *Kmt2c* and/or *Kmt2d* deletion by isolating organoids from *Tmprss2-CreER*$^{T2}$;*Pten*$^{f/f}$ mice, using clustered regularly interspaced short palindromic repeats–associated protein 9 (CRISPR–Cas9)-mediated deletion of *Kmt2c* and/or *Kmt2d* followed by treating with 4OHT to knock out *Pten* (Fig. 7a–c and Extended Data Fig. 10a,b).

We next used this system to evaluate the tumorigenic susceptibility of urothelial cells to recurrent MIBC mutations *PIK3CA*$^{E545K}$, *KRAS*$^{G12V}$ and *Trp53* loss. We observed tumors only when *Kmt2c* and/or *Kmt2d* were deleted together with another oncogene (Fig. 7d–f and Extended Data Fig. 10c–e). Intriguingly, without *Kmt2c/d* loss, the combination of *Pten* KO and *KRAS*$^{G12V}$ overexpression was still insufficient to generate tumors, suggesting that *Kmt2c/d* loss was required for urothelial carcinoma initiation.

The field cancerization hypothesis implicates the importance of the order of molecular events. To model the temporal loss of *Kmt2c/d* first and then *Pten*, we employed CRISPR–Cas9 to delete *Kmt2c/d* in *Tmprss2-CreER*$^{T2}$;*Pten*$^{f/f}$ cells and then deleted *Pten* in allografts by tamoxifen treatment, and vice versa (Fig. 7g and Extended Data Fig. 10f). We observed more robust tumorigenesis when *Kmt2c/d* was deleted first (Fig. 7g), suggesting that *Kmt2c/d* loss primed the urothelial cells for enhanced tumorigenic susceptibility. Overall, these data suggest that *Kmt2c/d* provides a tumor suppressive effect in the urothelium, and *Kmt2c/d* loss license a molecular 'field effect' that primes the urothelium for oncogenic transformations to a broad spectrum of oncogenic signal and environmental stimuli.

### *Kmt2c/d* loss confers sensitivity to EGFR inhibitors

We next sought to identify the therapeutic vulnerabilities unique to *Kmt2c/d* deficiency. Given the augmented cellular response to EGF in *Kmt2c/d*-deficient urothelial cells (Fig. 3c,d), we speculated that *Kmt2c/d* loss may lead to dependence on epidermal growth factor receptor (EGFR) signaling. We observed that dKO cells were more sensitive to EGFR inhibitors afatinib and gefitinib as well as EGF withdrawal (Fig. 8a–c). In vivo, allografts of sgdKO (sg*Kmt2c* + sg*Kmt2d*) + *Pten* KO (sg*Kmt2c* + sg*Kmt2d* + 4OHT) urothelial tumors exhibited significant growth inhibition by afatinib treatment (Fig. 8d). This was accompanied by decreased intensity of EGFR phosphorylation (pY845) and a decreased number of Ki-67-positive proliferating cells with afatinib treatment (Fig. 8e,f).

## Discussion

The concept of a 'field cancerization' was first introduced to describe the predisposition to tumorigenesis and recurrence in precancerous tissues[47]. The role of somatic mutations and clonal expansion in the field effect may be best exemplified in clonal hematopoiesis, where mutations in epigenetic genes such as *DNMT3A, ASXL1* and *TET2* increases the risks of subsequent myelodysplasia and leukemia[48]. Recent DNA sequencing of normal tissues have suggested a mutational basis for field cancerization in skin, esophagus, endometrium, liver and colon[49–56]. Members of the KMT2C/D–KDM6A complex are frequently mutated in normal urothelium[7,8]. In this study, we observed that *Kmt2c/d* KO mouse urothelial cells were histologically normal but were sensitized to subsequent tumorigenesis by oncogenes or a carcinogen, consistent with a field cancerization effect. The major transcriptional effects of *Kmt2c/d* deletion were (1) downregulation of the differentiation program leading to defective luminal differentiation

---

**Fig. 8 | *Kmt2c/d* loss confers therapeutic vulnerability to EGFR inhibitors. a,b**, Cell viability following treatment of WT and dKO urothelial cells with the EGFR inhibitors afatinib (**a**) and gefitinib (**b**). Cells were treated with serially diluted inhibitors for 4 days followed by cell viability measured using the CellTiter-Glo luminescent viability assay. IC$_{50}$ values are shown as mean ± s.d. and were analyzed with a two-tailed *t*-test (*n* = 3 independent experiments). **c**, Cell growth of WT and dKO cells (*n* = 3 independent experiments). A total of 0.1 million cells were initially plated at time day 0 (d0) followed by EGF withdrawal or treatment with gefitinib (100 nM) or afatinib (100 nM) beginning the second day after cell seeding. Cell number counting was performed 4 days after the start of drug treatment or EGF withdrawal. Data are shown as mean ± s.d. and were analyzed with two-tailed *t*-tests for each condition between WT and dKO groups. **d**, Growth curves of syngeneic tumors from sgdKO + *Pten* KO cells treated

with afatinib (10 mg kg$^{-1}$ day$^{-1}$) or vehicle control in NOD-SCID mice. Treatment was started 2 weeks after injection of cells into the mammary fat pad. The tumor volume was measured twice a week (*n* = 10 in each replicate, two independent replicates in total). Data are shown as mean ± s.e.m. and were analyzed with a two-tailed *t*-test at the end time point. **e,f**, Representative histological staining and statistics for p-EGFR Y845 (*n* = 3 tumors per group) (**e**) and Ki-67 (*n* = 5 tumors per group) (**f**) in tumor sections from vehicle and afatinib-treated mice. Tumors were collected 2 h after the last treatment. Scale bar, 500 μm. Data are shown as mean ± s.d. and were analyzed with a two-tailed *t*-test. **g**, A schematic illustration showing that *Kmt2c/d* deletion induces global redistribution of KMT2A from active enhancers to CpG-high promoters that primes the urothelium for transformation and elicits sensitivity to EGFR inhibitors.

and (2) upregulation of signal-dependent transcription such as IEGs and inflammatory genes (Fig. 8g).

Out data suggest that downregulation of a differentiation program is a direct effect of *Kmt2c/d* loss, consistent with its known role as a transcriptional activator. KMT2C/D are known to bind to active enhancers[11,29,57]. Genes regulated through enhancers are enriched

for lineage-specific programs[58]. Accordingly, we observed that *Kmt2c/d* KO led to impaired enhancer function. In addition, we found that KMT2D directly bound and regulated transcription at a set of CpG-poor promoters, which share many features with enhancers, including lineage-specific expression and binding by lineage-specific transcription factors[38,39,58]. Consistent with prior work, complete loss

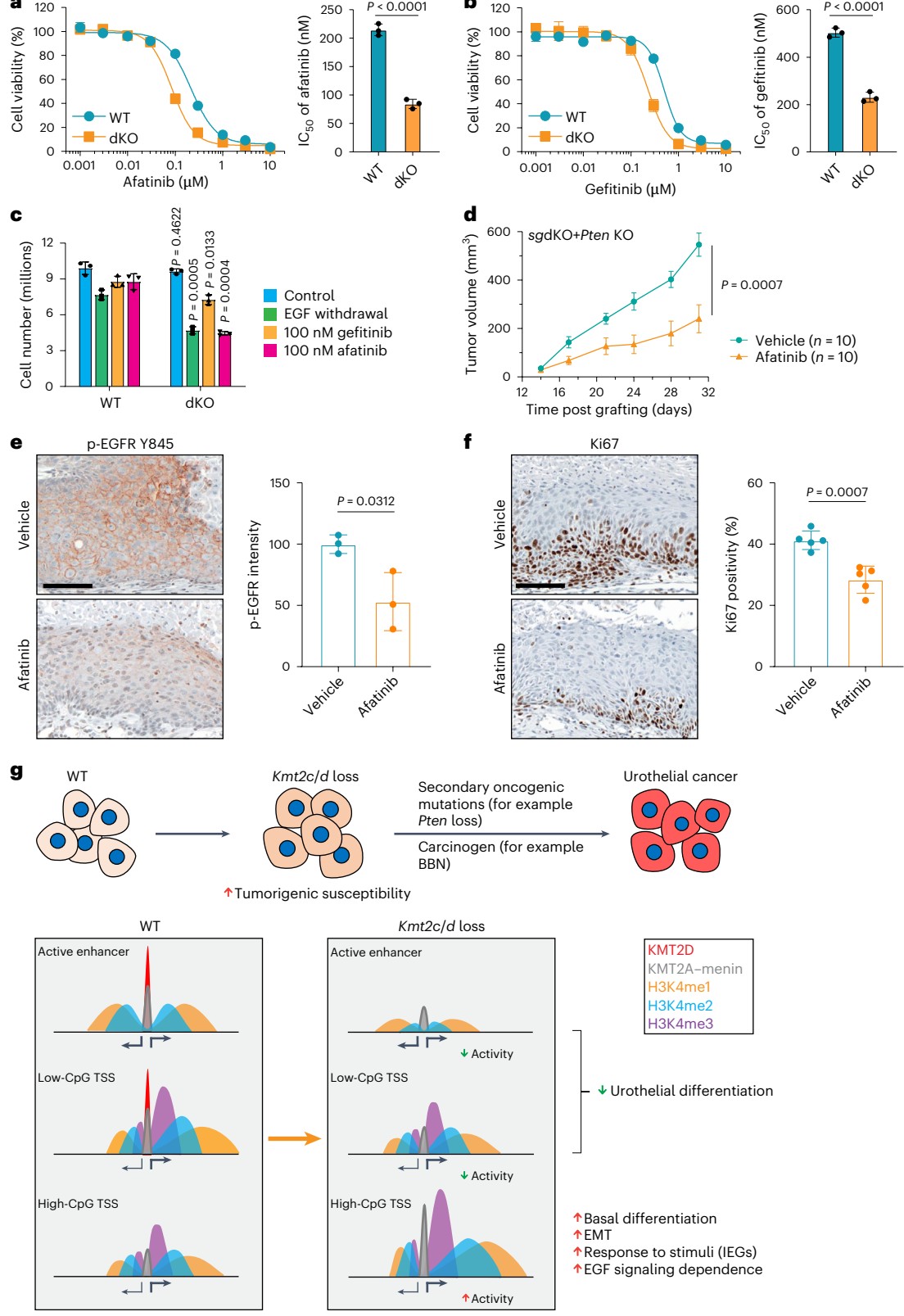

of both *Kmt2c/d* decreased but did not completely abolish H3K4me1 or enhancer RNA at enhancers[10–12,29–31]. We found that the KMT2A/B–menin complex was present at most KMT2D-positive enhancers, partially compensating for the loss of *Kmt2c/d* at these enhancers, consistent with recent work[34].

We unexpectedly observed that *Kmt2c/d* KO cells had a global increase in H3K4me2 and H3K4me3 by mass spectrometry. This correlated with their increased deposition at bivalent and high-CpG promoters, suggesting enhanced activity of other H3K4 methyltransferases in the setting of *Kmt2c/d* deficiency. Our data suggest that the KMT2A/B–menin complex is redistributed to these promoters from KMT2C/D-positive enhancers after *Kmt2c/d* loss. Bivalent promoters were initially characterized in embryonic stem cells and mark genes that can be rapidly silenced or induced during differentiation. Recent studies show bivalency also marks immediate response genes to poise them for rapid induction[27,40–42]. These data unveil an unexpected functional convergence of bivalent gene upregulation between loss of the transcriptional activating KMT2C/D–KDM6A complex and loss of the transcriptional suppressive PRC2 complex found in distinct cancers[59–61].

One limitation of our GEMM is the lack of luminal cancers. This may be due to species differences, as the BBN model also gives rise to basal cancer. We used *Tmprss2-CreER^{T2}* to induce *Kmt2c/d* KO in all urothelial cells, which may preclude the observation of biological consequence of deleting these genes in more differentiated cell populations. Use of luminal cell-specific Cre such as *Krt20-Cre* and *Upk-Cre* might be helpful in further studying the role of these genes in pathogenesis of urothelial tumors with a luminal phenotype.

## Online content

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

[1]Human Oncology and Pathogenesis Program, Memorial Sloan Kettering Cancer Center, New York, NY, USA. [2]National Institute of Diabetes and Digestive and Kidney Diseases, National Institutes of Health, Bethesda, MD, USA. [3]Department of Molecular Biology and Genetics, Cornell University, Ithaca, NY, USA. [4]Weill Institute for Cell and Molecular Biology, Cornell University, Ithaca, NY, USA. [5]Department of Computational Biology, Cornell University, Ithaca, NY, USA. [6]Urology Service, Department of Surgery, Memorial Sloan Kettering Cancer Center, New York, NY, USA. [7]State Key Laboratory of Cell Biology, Shanghai Key Laboratory of Molecular Andrology, Shanghai Institute of Biochemistry and Cell Biology, Center for Excellence in Molecular Cell Science, Chinese Academy of Sciences, Shanghai, China. [8]Department of Medicine, Memorial Sloan Kettering Cancer Center, New York, NY, USA. [9]Department of Medicine, Weill Cornell Medical College, New York, NY, USA. [10]These authors contributed equally: Naitao Wang, Mohini R. Pachai, Dan Li. ✉e-mail: chip@mskcc.org; cheny1@mskcc.org

## Methods

### Ethical regulations

Mouse experiments were conducted under protocol 11-12-027 approved by the Institutional Animal Care and Use Committee of Memorial Sloan Kettering Cancer Center (MSKCC), New York.

### Mouse studies

*Tmprss2-CreER^T2-IRES-nlsEGFP* (*Tmprss2^tm1.1(cre/ERT2)Ychen*, MGI:5911389) and *Pten^flox* (*Pten^tm2.1Ppp*, MGI:2679886) strains were obtained as previously described[17,62]. *Kmt2c^flox* strain (gene ID: 231051, homolog of *KMT2C*) with exon 3 flanked by *LoxP* sites was obtained from the Sarat Chandarlapaty's laboratory[18]. *Kmt2d^flox* strain (gene ID: 381022, homolog of *KMT2D*) with exon 50–51 flanked by *LoxP* sites was obtained from the Kai Ge's laboratory[19]. *Rosa26-CAG-LSL-EYFP* (*LSL-EYFP*) strain was purchased from the Jackson Laboratory (*B6.Cg-Gt(ROSA)26Sor^tm3(CAG-EYFP)Hze*, stock no. 007903)[63]. Primers for genotyping are listed in Supplementary Table 4.

To induce the Cre recombinase activity of the *Tmprss2-CreER^T2-IRES-nlsEGFP* allele, we administered two doses of tamoxifen (Toronto Research Chemicals, T006000, 3 mg per dose in corn oil) intraperitoneally to 6–12-week-old mice with an interval of 48 h.

To assay the effect of *Kmt2c/d* loss on BBN-induced urothelial tumor formation, we administered corn oil (WT group) or tamoxifen (dKO group) to 6–12-week-old *Tmprss2-CreER^T2;Kmt2c^f/f;Kmt2d^f/f* mice. Two weeks later, the mice were subsequently treated with 0.05% BBN in drinking water for 12 weeks. Bladder and ureter tissues were collected 4 weeks after the termination of BBN treatment.

Intravesical delivery of 4OHT and Cre-expressing adenovirus was performed on 6–12-week-old *Tmprss2-CreER^T2;Kmt2c^f/f;Kmt2d^f/f;Pten^f/f* mice[46]. The 4OHT (H7904, Millipore Sigma) was dissolved in ethanol at 1 mg ml$^{-1}$ as stock concentration. For each injection, 50 μl of 4OHT working solution (10% 4OHT stock solution + 5% Tween-80 + 85% Dulbecco's modified Eagle medium (DMEM)) was injected into the bladder cavity. Adenovirus adeno-mCherry (1767, Vector Biolabs), Adeno-mCherry-IRES-Cre (1771, Vector Biolabs), Ad5-bk5-Cre (VVC-Berns-1547-HT, UI Viral Vector Core Web) and Ad5mK8-nlsCre (VVC-Li-535-100ul, UI Viral Vector Core Web) was mixed 1:1 with DMEM with polybrene at 10 μg ml$^{-1}$. For each injection, 20 μl of virus were injected with a 31G insulin syringe. Bladder tissues were collected at 3 months post injection.

To test allograft tumor formation efficacy in urothelial cells, we isolated urothelial cells from *Tmprss2-CreER^T2;Pten^f/f* mice. Cells were infected with retrovirus (MSCV-Firefly Luciferase-PGK-NeoR-IRES-GFP) to express firefly luciferase for in vivo imaging. We further engineered them with oncogenic mutations (for example, CRISPR–Cas9 KO of *Kmt2c*, *Kmt2d*, *Trp53* or expression of *KRAS*^G12V and *PIK3CA*^E545K). Cells were treated with DMSO or 4OHT (0.2 μM for 24 h) to activate Cre-mediated deletion of *Pten*. We grafted 2 million urothelial organoid cells (cultured and virally infected as described above) into the mammary fat pad of 6–8-week-old female non-obese diabetic severe combined immunodeficiency disease (NOD-SCID) mice obtained from the Jackson Laboratory (strain: NOD.CB17-Prkdc<scid>, stock no. 001303). For luminescent imaging, we used the IVIS Spectrum imaging system.

To assay the in vivo response to EGFR blockade, we grafted 2 million sgdKO (sg*Kmt2c* + sg*Kmt2d*) + *Pten* KO (recombination of floxed allele) urothelial cells into the mammary fat pad of 6–8-week-old female NOD-SCID mice (strain: NOD.CB17-Prkdc<scid>, stock no. 001303). We treated the mice with vehicle (0.1% Tween-80 and 0.5% carboxymethylcellulose in sterile water) or afatinib (MedChemExpress, HY-10261, 10 mg kg$^{-1}$ day$^{-1}$, daily, for 5 days per week by oral gavage). Tumor size was measured with a digital caliber and calculated by $v = \frac{4\pi}{3} * \frac{a}{2} * \frac{b}{2} * \frac{c}{2}$, where $a$, $b$ and $c$ represent the length, width and thickness of the tumor, respectively. The maximum tumor burden allowed in our protocol is 2,000 mm$^3$. We performed bilateral mammary fat pad grafts, and no single tumor size exceeded 1,000 mm$^3$ in this study.

### ScRNA-seq and analysis

Bladders from *Tmprss2-CreER^T2-IRES-nlsEGFP;Kmt2c^f/f;Kmt2d^f/f* mice were collected 3 months post tamoxifen administration (dKO, $n = 3$ mice). Bladders from mice at the same age without tamoxifen treatment were collected as WT ($n = 4$ mice). After euthanasia, the bladders were dissected out and minced with a scalpel and then processed by 1 h digestion with collagenase/hyaluronidase (07912, Stemcell Technologies) and 15 min digestion with TrypLE (12605010, Gibco). Live single urothelial cells were sorted out by a BD FACSymphony S6 Cell Sorter as DAPI$^-$/EpCAM$^+$/nlsEGFP$^+$ (17579180, eBioscience). For each mouse, 5,000 cells were directly processed with the 10X genomics Chromium Single Cell 3′ GEM, Library and Gel Bead Kit v3 (10X Genomics), according to the manufacturer's specifications. For each sample, 200 million reads were acquired on a NovaSeq platform S4 flow cell.

Reads obtained from the 10X Genomics scRNA-seq platform were mapped to mouse genome (GRCm38) including the transgenes, using the Cell Ranger (7.0.0) software (10X Genomics). True cells were distinguished from empty droplets using the scCB2 (1.14.0) package[64]. Downstream analysis and figure plotting were processed using Scanpy (1.6.1)[65]. The levels of mitochondrial reads and numbers of unique molecular identifiers were similar among the samples, which indicates that there were no systematic biases in the libraries from mice with different genotypes. Cells were removed if they expressed fewer than 600 unique genes, less than 1,500 total counts, more than 50,000 total counts or greater than 20% mitochondrial reads. Genes detected in less than ten cells and all mitochondrial genes were removed for subsequent analyses. Putative doublets were detected and filtered out using the doublet detection package (4.2)[66,67]. The average gene detection in each cell type was similar among the samples. Combining samples in the entire cohort of WT and dKO groups yielded a filtered count matrix of 25,858 cells by 16,705 genes, with a median of 12,984 counts and a median of 3,135 genes per cell and a median of 3,638 cells per sample. The count matrix was normalized by log$_2$(counts per million (CPM) + 1) for the violin plots in Fig. 2c–e and Supplementary Fig. 3d and the correlation analysis in Supplementary Fig. 1b. The count matrix was then normalized by median library size (12,984) and log$_e$($X$ + 1) transformed for calculating the top 1,000 most highly variable genes using Scanpy. The count matrix was further scaled to mean as 0 and s.d. as 1 for principal component analysis and UMAP dimension reduction (https://arxiv.org/abs/1802.03426), and Leiden clustering[68]. Principal component analysis was performed on the 1,000 most variable genes and the top 50 principal components explained 43% of the variance. Marker genes for each cluster were found with Scanpy[65]. Cell types were determined using a combination of marker genes identified from the literature and gene ontology for cell types using the web-based tool PanglaoDB[69].

Differentially expressed genes between WT and dKO mice were compared with the MAST (1.30.0) package[70]. The log$_e$ fold change of MAST output was used for the ranked gene list in GSEA analysis. GSEA analyses were performed using the JAVA GSEA 4.1.0 program, using curated gene sets (C2) and Hallmark gene sets (H) from the Molecular Signatures Database v7.4. We further added custom gene sets of IEGs[26] and BLCA_NAT_versus_HEALTHY_UP and BLCA_NAT_versus_HEALTHY_DN defined by differentially expressed genes (absolute fold change >5, $P < 0.05$) between NAT ($n = 19$) and healthy tissue ($n = 11$)[28]. Imputation was performed using the Markov affinity-based graph imputation of cells package (3.0.0)[71]. Imputed data were used in the heat map images (Supplementary Figs. 1c, 2a and 3b).

### Urothelial cell culture and characterization

Mouse bladder urothelial cells were dissociated and FACS sorted as described above. For organoid culture, urothelial cells from *Tmprss2-CreER^T2-IRES-nlsEGFP;Kmt2c^f/f;Kmt2d^f/f* mice with or without tamoxifen treatment (two doses, 3 mg per dose, 3 months) were sorted out as dKO and WT groups, respectively. Urothelial organoids were cultured in advanced DMEM/F12 (12634010, Gibco) (supplemented with B27

(2% v/v, 17504044, Gibco), Noggin conditioned medium (10% v/v)[72], R-spondin conditioned medium (10% v/v)[72], EGF (50 ng ml⁻¹, AF-100-15, PeproTech), Y-27632 (10 μM, S1049, Selleckchem), A83-01 (0.5 μM, S7692, Selleckchem), N-acetyl-L-cysteine (1.25 mM, A9165, Millipore Sigma), Primocin (1% v/v, ant-pm-2, InvivoGen), penicillin–streptomycin (1% v/v, 15140122, Gibco), L-glutamine (1% v/v, 25030081, Gibco) and GlutaMAX (1% v/v, 35050061, Gibco)). Urothelial cells were mixed 1:2 with growth factor depleted Matrigel (356231, BD) at final concentration of 10,000 cells per millilitre. We further developed a method of culturing urothelial cells on a two-dimensional (2D) condition, in which collagen I was used to coat the plate (A1048301, diluted with cold-sterilized $H_2O$, 1 h at room temperature). The same organoid culture medium was used for both three-dimensional and 2D culture. Except for organoid culture with freshly dissociated urothelial cells specified in the paper, all other experiments were performed with 2D urothelial cells.

To assay organoid formation efficiency, we seeded 500 freshly FACS-sorted urothelial cells in each 50 μl Matrigel blob. We cultured the organoids for 9 days before quantification under light microscopy. Organoid formation efficiency is the number of organoids divided by 500 per Matrigel blob.

In the transwell invasion assay, FluoroBlock transwell inserts (351152, Corning) were precoated with Matrigel overnight in the incubator (1:30 dilution in sterilized $H_2O$, 356231, BD). Urothelial cells were first starved for 48 h in basic medium (DMEM/F12, primocin, penicillin–streptomycin, L-glutamine and GlutaMAX), then 100,000 cells in 100 μl basic medium were seeded on the top chamber of transwell inserts. After 30 min incubation, 500 μl full medium was added to the bottom chamber. Twenty-four hours later, the transwell inserts were fixed, permeabilized and stained with 4,6-diamidino-2-phenylindole (DAPI) (0.1 μg ml⁻¹). Images were taken using a Nikon ECLIPSE Ti2 inverted microscope.

In differentiation assay, cells were first seeded in full medium overnight and then cultured in differentiation medium (DMEM/F12, B27, Y-27632, N-acetyl-L-cysteine, primocin, penicillin–streptomycin, L-glutamine and GlutaMAX) for 9 days[25]. Organoids were characterized by immunofluorescence staining and intracellular flow cytometry.

For immunofluorescence staining, organoids were digested with Dispase (1 mg ml⁻¹, 17105041, Gibco) for 30 min at 37 °C. Collected organoids were fixed in 4% paraformaldehyde (PFA) for 1 h and dehydrated in 30% sucrose overnight at 4 °C. On the second day, urothelial organoids were embedded in optimal cutting temperature compound and prepared for sectioning. Primary antibodies against KRT5 (905501 and 905901, 1:400, Biolegend), KRT8 (904801, 1:400, Biolegend) and Ki-67 (ab16667, 1:100, Abcam) were used. Secondary antibodies with Alexa fluor 488 (A11039 and A11001, 1:500), Alexa fluor 555 (A21428, 1:500) conjugation were purchased from Thermo Fisher Scientific. Images were taken with a Leica TCS SP5 upright confocal microscope.

For intracellular flow cytometry analysis, organoids were digested with Dispase for 30 min and then further dissociated into single cells with TrypLE for 15 min on a shaker in a cell culture incubator. Single urothelial cells were then fixed with 4% PFA for 10 min and permeabilized with 0.5% Triton-X 100 for 10 min. Primary antibodies against KRT5 (905501, 1:400, Biolegend) and KRT8 (904801, 1:400, Biolegend), secondary antibodies Alexa fluor 633 conjugated goat anti-rabbit (A21071, 1:500, Thermo Fisher Scientific) and Alexa fluor 555 conjugated goat anti-mouse (A21422, 1:500, Thermo Fisher Scientific) were then applied in order for 30 min on ice. For cell surface flow cytometry analysis, fluorescence-conjugated antibodies against H-2Kb/H-2Db-APC (114614, 1:200, Biolegend) and PD-L1-APC (124311, 1:200, Biolegend) were directly stained with viable cells for 30 min on ice. All samples were analyzed with a BD LSRFortessa instrument. Data were further processed using BD FACSDiva software v6.2 and FlowJo 10.7.1.

To determine whether KMT2A/B–menin inhibition with MI-503 can reverse the impaired organoid formation and invasion after *Kmt2c/d*

loss, WT and dKO urothelial cells were pretreated with DMSO or MI-503 (S7817, Selleckchem, 1 μM) for 3 days, and 200 urothelial cells were then cultured in Matrigel with the presence of DMSO or MI-503 (1 μM) for 8 days. In the Matrigel invasion experiment, WT and dKO urothelial cells were pretreated with DMSO or MI-503 (1 μM) in basic medium for 2 days. DMSO or MI-503 were added to both the top and bottom of the transwell chamber. The transwell inserts were then fixed and stained with DAPI (0.1 μg ml⁻¹) after 24 h incubation. Images were taken using a Nikon ECLIPSE Ti2 inverted microscope.

## Mass spectrometry of histone PTMs

Quantification of histone post translational modifications (PTMs) was performed by Active Motif Mod Spec service[73]. Histones from WT and dKO urothelial cells were extracted, processed and measured using the Thermo Scientific TSQ Quantum Ultra mass spectrometer coupled with an UltiMate 3000 Dionex nano-liquid chromatography system. All samples were run in triplicate. Data were quantified using Skyline and represent the percent of each modification within the total pool of that tryptic peptide[74].

## RNA-seq and PRO-cap

Urothelial organoids with the *Tmprss2-CreER^T2^;Kmt2c^f/f^;Kmt2d^f/f^* genotype were infected with Adeno-mCherry (1767, Vector Biolabs) or Adeno-mCherry-IRES-Cre (1771, Vector Biolabs) to generate *Kmt2c/d* WT and dKO cells. Successful *Kmt2c* and *Kmt2d* depletions were determined by qPCR with genotyping primers.

In WT and dKO cells, we performed poly-A RNA-seq in triplicates. Total RNA was extracted from WT and dKO urothelial cells with TRIzol reagent, following the manufacturer's instructions (15596026, Invitrogen). RNA-seq libraries were prepared using the standard Illumina Poly-A library preparation protocol. Next-generation sequencing was performed by the MSKCC Integrated Genomics Operation (IGO) on an Illumina NovaSeq6000 with paired-end 100 bp for 30–40 million reads. The sequencing data were mapped to the mouse genome (GRCm38) with spiked in genes *CreER^T2^* and *EGFP* using STAR (2.7.10b)[75]. Raw counts were quantified using STAR option (–quantMode GeneCounts). In the rescue experiment, urothelial cells were treated with DMSO or MI-503 (1 μM for 4 days).

In WT and dKO cells, we performed PRO-cap[33,76]. Ten million WT or dKO urothelial cells were used for each reaction. Libraries were sequenced by Novogene Corporation for 60–70 million reads (paired end, 150 bp). Peaks were called with algorithms and parameters suggested on PINTS (1.1.8) (https://pints.yulab.org/tre_calling)[33]. Briefly, sequencing data were first processed for adapter trimming (fastp -i) and then aligned to mouse genome (GRCm38) using STAR (2.7.10b). Bidirectional enriched peaks were identified using pints_caller. Bidirectionally enriched peaks that did not overlap with promoters of coding or long-non-coding RNAs were regarded as candidate enhancer RNAs. Bigwig files of plus and minus strand alignments were generated using pints_visualizer. Enrichment analyses of genes of the top 500 PRO-cap down genes among C5 curated gene sets were performed on the GSEA website (https://www.gsea-msigdb.org/gsea/index.jsp).

## Epigenetic sequencing

In WT and dKO organoids, we performed ATAC-seq; ChIP-seq of KMT2D and H3K27ac; and Cut&Run of KMT2A, menin, SET1A, CXXC1, H3K4me1, H3K4me2, H3K4me3, H3K27me3, H3K9me3 and H3K36me3.

ATAC-seq was performed using the standard protocol[77]. For each WT and dKO sample, 50,000 viable cells were processed for nuclei isolation and transposase treatment. The digestions were carried out for 30 min at 37 °C. The library preparation and next-generation sequencing were performed by the MSKCC IGO core facility. For each sample, 40–50 million paired-end reads were sequenced on an Illumina platform HiSeq4000 for a paired-end 50 bp run or a NovaSeq6000 for a paired-end 100 bp run. The sequence data were processed for adapter trimming using

trim_galore and aligned to the mouse genome (GRCm38) with bowtie2 (2.4.5)[78]. The average of the replicated BigWig files was generated with the bigwigCompare function of deepTools (2.0)[79].

The Cut&Run protocol and the reagent pA/Mnase was obtained from Steven Henikoff's laboratory[80]. In each experiment, 250,000 viable WT or dKO cells were processed. Antibodies (1 μg per sample) against H3K4me1 (710795, Thermo Fisher Scientific), H3K4me2 (710796, Thermo Fisher Scientific), H3K4me3 (PA57-27029, Thermo Fisher Scientific), H3K27me3 (9733, Cell Signaling Technology), H3K9me3 (ab176916, Abcam), H3K36me3 (61021, Active Motif), SET1A (ab70378, Abcam), CXXC1 (ab198977, Abcam), KMT2A (A300-086A, Bethyl Laboratories) and menin (A300-105A, Bethyl Laboratories) were applied. Next-generation sequencing was performed by the MSKCC IGO core facility on an Illunima platform with paired-end 50 bp (HiSeq4000) or paired-end 100 bp (NovaSeq6000) to obtain 10–20 million reads. Sequencing data were processed for adapter trimming and aligned to the mouse genome (GRCm38).

ChIP-seq was conducted following the standard protocol[81]. Antibodies (2 μg per 10 million cells) against H3K27ac (ab4729, Abcam) and KMT2D (a kind gift from Dr. Kai Ge's laboratory) were applied. Next-generation sequencing was performed by the MSKCC IGO core facility on an Illunima platform with paired-end 50 bp (HiSeq4000) or paired-end 100 bp (NovaSeq6000) to obtain 30–40 million reads. KMT2D ChIP-seq was performed in duplicate with the Drosophila genome as a spike-in for normalization. Sequencing data were processed for adapter trimming and aligned to the mouse genome (GRCm38) and the drosophila genome (dm6).

For epigenetic sequencing data, duplicates were marked with samblaster[82] (0.1.26). Mapping quality was analyzed with qualimap[83] (2.2.2-dev).

### Integrative analysis of ChIP-seq, Cut&Run, ATAC-seq, RNA-seq and PRO-cap data

To identify peaks for KMT2A, KMT2D, menin, SET1A, CXXC1, H3K4me1 and H3K4me3, we used MACS3 (3.0.0) (https://github.com/macs3-project/MACS)[84] to call both replicates individually with a false discovery rate (FDR) <0.01, and intersected peaks were called in both replicates (bedops intersect)[85]. To identify peaks for H3K27me3, we used Sicer2 (1.0.3) (https://zanglab.github.io/SICER2/)[86] with FDR <0.01, gap size of 600. To generate a peak set for each mark, we merged peaks from WT and dKO cells (bedops merge).

We generated a total merged peak set by merging (using Homer mergePeaks) KMT2D, KMT2A, menin, SET1A and CXXC1 peaks together with all promoter coordinates from the Eukaryotic Promoter Database (https://epd.expasy.org/epd/EPDnew_database.php)[87]. Two peaks were considered overlapping if their peak had 1 bp overlap (Homer mergePeaks -d given). We excluded any peaks that overlapped with ENCODE blacklisted genomic regions[88]. We annotated each region to the nearest TSS, obtained gene expression RNA-seq of the associated gene and calculated the CpG content of the region using Homer 'annotatepeaks.pl mm10 -CpG -gene'. We annotated peaks as a promoter if the peak centers were within −1,000 to +500 bp from the nearest TSS, and other peaks were annotated as 'enhancer'.

We quantified the read counts at the merged enhancer and promoter peak sets on KMT2D, KMT2A, menin, SET1A, CXXC1, H3K4me1, H3K4me2, H3K4me3, H3K27me3, H3K27ac, ATAC-seq and PRO-cap using featureCounts[89] (v2.0.1) of the respective duplicate Bam files (Supplementary Table 1). We used the average of duplicates for downstream analysis. RNA-seq and PRO-cap counts were normalized to the total mapped reads. ATAC-seq, ChIP-seq and Cut&Run counts were normalized with reads per genome content (RPGC), except for Cut&Run against H3K4me1, which was difficult to normalize due to global loss of this mark. We ran ChIP-qPCR against a number of sites and normalized based on this signal. KMT2D ChIP-seq was normalized with a spike-in drosophila genome. Heat maps and aggregation plots were generated

using deepTools (3.5.1)[79]. To obtain RNA-seq gene expression associated with peaks, we used the Homer annotatePeaks -gene function.

To identify whether a promoter overlaps with H3K4me3 (which always have H3K4me1), H3K4me1 only, H3K27me3 only or both H3K4me3 and H3K27me3, we determined whether the TSS ±2,000 kb overlapped with the relevant peak set using bedtools interest -C.

ChromHMM (v1.25) LearnModel was performed in WT and dKO groups with H3K4me1, H3K4me3, H3K27ac, H3K27me3, H3K9me3 and H3K36me3 (ref. 32). We tested the number of states from 8 to 14 and found that the 12-state model was optimal. To calculate the enrichment of chromatin states at the TSS of up- and downregulated genes, we used ChromHMM OverlapEnrichment. For the representative epigenetic alterations on enhancer and promoter, we performed ChIP-qPCR for validation. The primers used for ChIP-qPCR are listed in Supplementary Table 4.

### Analysis of TCGA human MIBC dataset

To analyze the enrichment of luminal, basal and squamous signatures in human bladder samples, we employed the ssGSEA (v4.0) method for bulk RNA-seq deconvolution analysis[90]. Briefly, ssGSEA takes the sample's fragments per kilobase of transcript per million mapped reads expression values as the input and computes an enrichment score for a given gene list as compared with all the other genes in the sample transcriptome. MIBC subtype signatures were obtained according to a prior report[6]. The raw gene expression signature score was defined as the mean of the z-score for the basal markers (*CD44, CDH3, KRT1, KRT14, KRT16, KRT5, KRT6A, KRT6B and KRT6C*), luminal markers (*CYP2J2, ERBB2, ERBB3, FGFR3, FOXA1, GATA3, GPX2, KRT18, KRT19, KRT20, KRT7, KRT8, PPARG, XBP1, UPK1A* and *UPK2*) and squamous markers (*DSC1, DSC2, DSC3, DSG1, DSG2, DSG3, S100A7* and *S100A8*). To determine gene expression differences between *KMT2C/D* mutant and intact groups, we used the cBioportal annotation of the TCGA MIBC dataset (2017) (https://www.cbioportal.org/study/summary?id=blca_tcga_pub_2017). We divided samples into two groups of (1) either *KMT2C* or *KMT2D* mutant and (2) no mutations in *KMT2C* or *KMT2D*. We annotated the promoters of these genes by CpG content using Homer annotatePeaks.pl to determine the expression difference by CpG content. Enrichment analysis of the top 500 differentially expressed genes between *KMT2C/D* mutant and intact groups (by fold change) among the C2 curated gene sets was performed on the GSEA website (https://www.gsea-msigdb.org/gsea/index.jsp).

### Histology and IHC

Mouse bladder, ureter and urethra tissues were collected, fixed in 4% PFA, dehydrated and embedded in paraffin. Paraffin embedding and sectioning was performed by Histioserv Inc. After sectioning, IHC staining was performed on a Ventana automatic stainer. Primary antibodies used in this study were KRT5 (905501, 1:500, Biolegend), UPK2 (ab213655, 1:500, Abcam), GFP (2956, 1:200, Cell Signaling Technology), H3K4me1 (5326, 1:100, Cell Signaling Technology), Ki-67 (ab16667, 1:100, Abcam), PTEN (9188, 1:200, Cell Signaling Technology), SMA (ab5694, 1:1,000, Abcam) and p-EGFR Tyrosine 845 (2231, 1:50, Cell Signaling Technology). Both H&E and IHC sections were scanned using a Mirax Digital Slide scanner. Immunofluorescence staining of KRT14 (906004, 1:400, Biolegend), KRT5 (905501, 1:400, Biolegend), KRT20 (M7019, 1:500, Dako Omnis) and UPK2 (ab213655, 1:500, Abcam) was performed on paraffin-embedded sections. After dewaxing, sections were permeabilized with 0.5% Triton-X 100 for 10 min and blocked with 10% normal goat serum for 30 min at room temperature. Primary antibodies anti-KRT5 and anti-KRT14 were incubated overnight. Secondary antibodies with Alexa fluor 488 or Alexa fluor 555 (A11001, A11039 and A21428, 1:500, Thermo Fisher Scientific) conjugation were applied on the second day. Images were taken with a Leica TCS SP5 upright confocal microscope. Ki-67 positivity and p-EGFR Tyrosine 845 intensity were quantified using QuPath-0.4.4 (ref. 91).

## BaseScope

Paraffin-embedded tissues were sectioned and preserved at 4 °C to prevent RNA degradation. Freshly sectioned slides were stained using Leica Bond RX following the standard protocol. Probes of *Kmt2c* (1285828-C1, ACD Bio) and *Kmt2d* (1285848-C1, ACD Bio) were incubated at 42 °C for 2 h. BaseScope LS Reagent kit-RED was used to visualize the signal (323600, ACD Bio). Sections were scanned using a Mirax Digital Slide scanner.

## QPCR

Total RNA was extracted with either TRIzol reagent or a total RNA extraction Kit (R1034, Omega Bio-Tek). First-strand complementary DNA was synthesized with the High-Capacity cDNA Reverse Transcription kit (4368814, Applied Biosystems). Real-time PCR was performed with PowerUp SYBR Green Master Mix (A25741, Applied Biosystems) on a QuantStudio 7 Flex Real-Time PCR machine. The primers for qPCR are listed in Supplementary Table 4.

## Western blot

Cells were lysed in 1% SDS or RIPA buffer and quantified using the BCA method. Primary antibodies against GAPDH (2118, 1:5,000, Cell Signaling Technology), PTEN (9188, 1:2,000, Cell Signaling Technology) PIK3CA (4249, 1:2,000, Cell Signaling Technology), p-AKT Serine 473 (4060, 1:2,000, Cell Signaling Technology), p-AKT Threonine 308 (13038, 1:2,000, Cell Signaling Technology), p-ERK Threonine202/Tyrosine204 (4370, 1:2,000, Cell Signaling Technology), AKT (4691, 1:5,000, Cell Signaling Technology), RAS (8832, 1:500, Cell Signaling Technology), p53 (2524, 1:1,000, Cell Signaling Technology), KMT2A (14689, 1:1,000, Cell Signaling Technology), KMT2B (47097, 1:1,000, Cell Signaling Technology), menin (A300-115A, 1:1,000, Bethyl Laboratories), SET1A (50805, 1:1,000, Cell Signaling Technology), SET1B (44922, 1:1,000, Cell Signaling Technology), CXXC1 (ab198977, 1:1,000, Abcam), KMT2D (1:1,000, a kind gift from Kai Ge's laboratory) and Vinculin (13901, 1:5,000, Cell Signaling Technology) were applied. Membranes were developed with enhanced chemiluminescence western blotting substrate (32106, Thermo Fisher Scientific) and imaged with an Amersham ImageQuant 800 biomolecular imager.

## CRISPR–Cas9

Plasmids lentiCRISPRv2 with puromycin, hygromycin or blasticidin resistance were obtained from Addgene (98293, 98290 and 98291). Successful CRISPR–Cas9 editing was validated with western blotting or a surveyor assay (M0302, New England Biolabs). Sequences of guide RNA and primers for the surveyor assay are listed in Supplementary Table 4.

## Plasmid

The pMSCV-KRAS$^{G12V}$-IRES-EGFP plasmid was constructed in our laboratory[92]. The pHAGE-PIK3CA$^{E545K}$-IRES-EGFP plasmid was obtained from Addgene (116485). Retrovirus production was performed on 293 T cells through cotransfection of pMSCV-KRAS$^{G12V}$-IRES-EGFP together with pCL-Ampho and pVSV-G. Lentivirus production was performed in 293 T cells through cotransfection of pMD2.G and pCMVR8.74. Transfection reagent X-tremeGENE 9 (Roche) was used to increase the transfection efficiency. The successful exogenous gene expression in urothelial cells was validated by western blot.

## Statistics and reproducibility

Statistical analysis was performed as detailed in the figure legends. Samples sizes were not predetermined by any statistical methods. In our studies, sample sizes were similar to prior studies[93] and indicated in the figures or figure legends. No data were excluded from the analyses. Mouse treatment and tumor measurements were performed by the same person blinded to the therapeutic effects. Experiments were successfully repeated with a minimum of two independent experiments. In most cases, we performed multiple experiments to address the same scientific question.

## Reporting summary

Further information on research design is available in the Nature Portfolio Reporting Summary linked to this article.

## Data availability

Raw sequencing data are publicly available from the Gene Expression Omnibus: GSE180947, GSE236370 and GSE264514. Raw mass spectrometry data of histone PTMs are available via ProteomeXchange with the identifier PXD056439. All other data supporting the findings of this study are available upon request from the corresponding authors. Source data are provided with this paper.

## Code availability

This study did not use custom code or software. All code used in the study has been published and has been cited in Methods.

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

## Acknowledgements

This work was supported by grants from the National Institute of Health (NIH) and National Cancer Institute (NCI) grants (R01CA228216, DP2CA174499 and P50CA217694 to P.C.; U54CA224079, P50CA092629, R01CA193837, U01CA224044 and R01CA208100 to Y.C.; R01CA233899 to H.A.A.-A.; P50CA221745 to Y.C., D.B.S., G.V.I., J.E.R. and H.A.A.-A.; P01CA221757 to D.B.S.; and P30CA008748 to Memorial Sloan Kettering Cancer Center (Core Grant). The Samuel Waxman Cancer Research Foundation to Y.C., G.V.I. and J.E.R. The Gladstein Family Bladder Cancer Research Fund to Y.C., G.V.I. and J.E.R. The Geoffrey Beene Cancer Research Fund to P.C. and Y.C. A Department of Defense grant (W81XWH-15-1-0124) and Francis Collins Scholar NTAP, Cycle for Survival and Linn Family Discovery Fund to P.C. The mass spectrometry proteomics data have been deposited in the ProteomeXchange Consortium via the PRIDE partner repository. We thank the following core facilities at MSKCC: Flow Cytometry, Integrated Genomics Operations, Molecular Cytology and Research Animal Resource Center.

## Author contributions

N.W. planned and carried out the experiments, conceived the idea, analyzed data, prepared figures and wrote the paper. M.R.P. did the mammary fat pad allografts, treated the mice, performed histological sectioning and staining, analyzed the ratio of Ki-67[+] cells in tissue sections and generated the BBN mouse models. D.L. analyzed the next-generation sequencing data and prepared figures. C.L. performed the genotyping, injected tamoxifen and recorded survival conditions of mouse models. S.W. and M.N.K. did the mammary fat pad allografts. W.H.C. performed histological staining and intravesical injections. C.Q. collected and dissociated the allografts. G.X. performed ChIP-seq of KMT2D and assisted with data processing. S.R.S., L.Y. and H.Y. performed and analyzed the PRO-cap assay. E.W.P.W., J.Y. and F.V.T. provided experimental methods. F.K., S.P.G. and J.L. performed the molecular signature analysis in the MIBC dataset. M.H., D.G., W.H., D.B.S., G.V.I. and J.E.R. provided experimental materials and/or gave suggestions to the paper. A.S. and S.C. provided the *Kmt2c*^{f/f} mouse model. K.G. provided the *Kmt2d*^{f/f} mouse model. H.A.A.-A. performed the histological analyses. P.C. and Y.C. directed the experimental designing, conceived the idea, analyzed data and wrote the paper.

## Competing interests

P.C. has received personal honoraria/advisory boards/consulting fees from Deciphera, Exelixis, Zai Lab, Novartis and Ningbo NewBay Medical Technology. P.C. has received institutional research funding from Pfizer/Array, Novartis, Deciphera and Ningbo NewBay Medical Technology. Y.C. has stock ownership and received royalties from Oric Pharmaceuticals. S.C. has received institutional research support from Daiichi-Sankyo, Paige.ai and AstraZeneca, is founder/shareholder of Odyssey Biosciences, has shares in Totus Medicines, and personal consulting fees from Sanofi, Novartis, Inivata, Lilly and AstraZeneca. J.E.R. has received personal honoraria/consulting fees from BMS, Merck, Pfizer, Pharmacyclics, Boheringer Ingelheim, EMD-Serono, GSK, Infinity, Janssen, Mirati, BioClin, Lilly, Tyra Biosciences, Astellas, Seagen, Bayer, AstraZeneca, Roche/Genentech and QED Therapeutics, and conducted trials from Astellas, Seagen, Bayer, AstraZeneca, Roche/Genentech and QED Therapeutics. S.R.S. is an equity holder and member of the scientific advisory board of NeuScience, Inc. and a consultant at Third Bridge Group Limited. D.B.S. has consulted/received honoraria from Rain, Pfizer, Fog Pharma, PaigeAI, BridgeBio, Scorpion Therapeutics, FORE Therapeutics, Function Oncology, Pyramid and Elsie Biotechnologies, Inc. None of these disclosures are directly related to this study. The other authors declare no competing interests.

## Additional information

**Extended data** is available for this paper at https://doi.org/10.1038/s41588-024-02015-y.

**Correspondence and requests for materials** should be addressed to Ping Chi or Yu Chen.

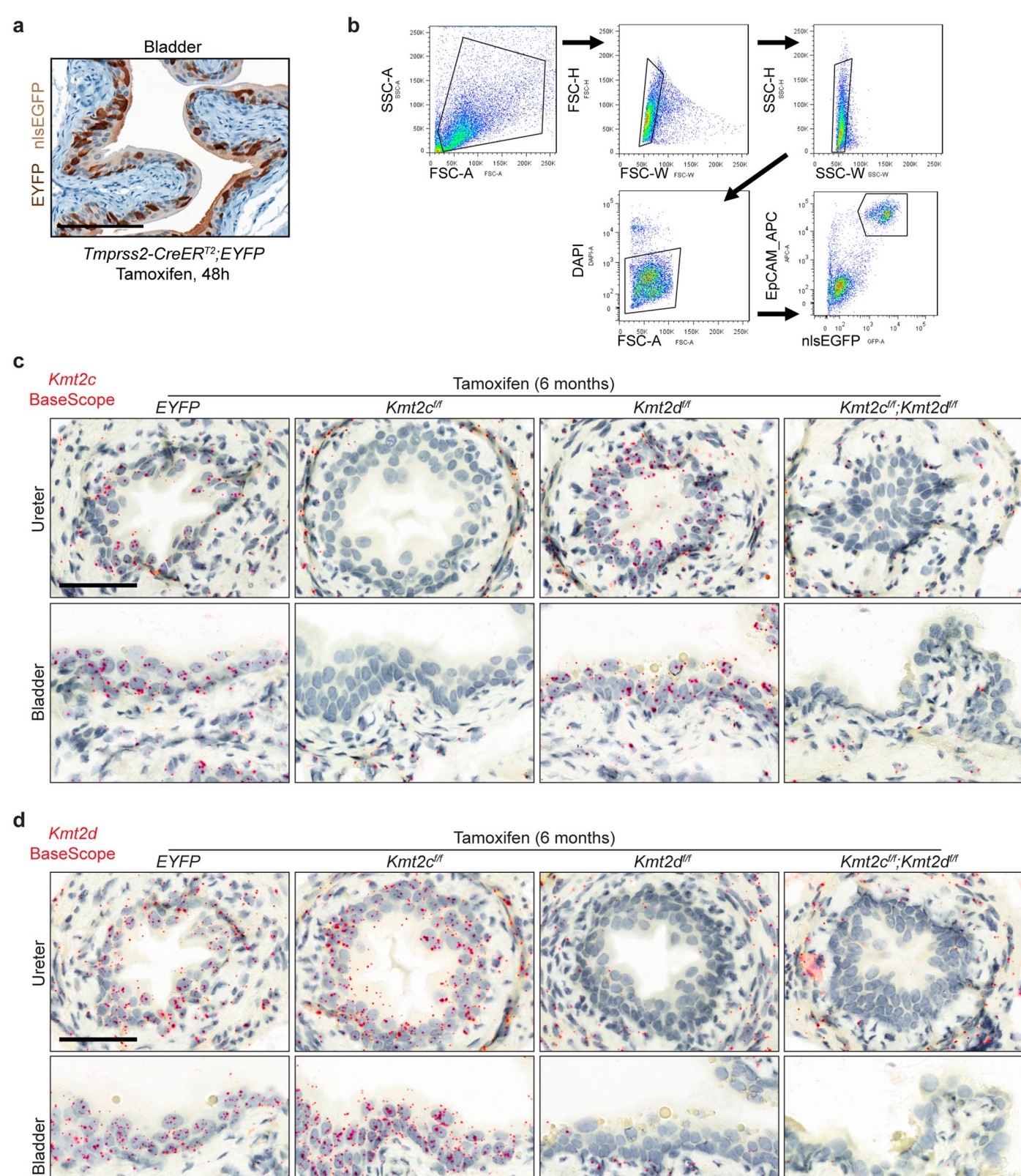

**Extended Data Fig. 1 | Characterization of Cre-mediated recombination in *Tmprss2-CreER^T2* mouse urothelium. a**, IHC using anti-GFP antibody that recognizes both nlsEGFP and EYFP in bladder sections from *Tmprss2-CreER^T2;Rosa26-CAG-LSL-EYFP* mice (n = 3 mice). Tissues were collected 48 h after a single dose of Tamoxifen (3 mg). Note that the stronger staining was from EYFP, while the weaker staining was from nlsEGFP. Scale bar, 100 μm. **b**, FACS sorting of EpCAM⁺/nlsEGFP⁺ viable urothelial cells in dissociated mouse bladder with no tamoxifen treatment. **c** and **d**, Representative BaseScope staining of ureter and bladder with probes targeting *Kmt2c* or *Kmt2d* floxed exons (n = 3 mice in each group). Scale bar, 50 μm.

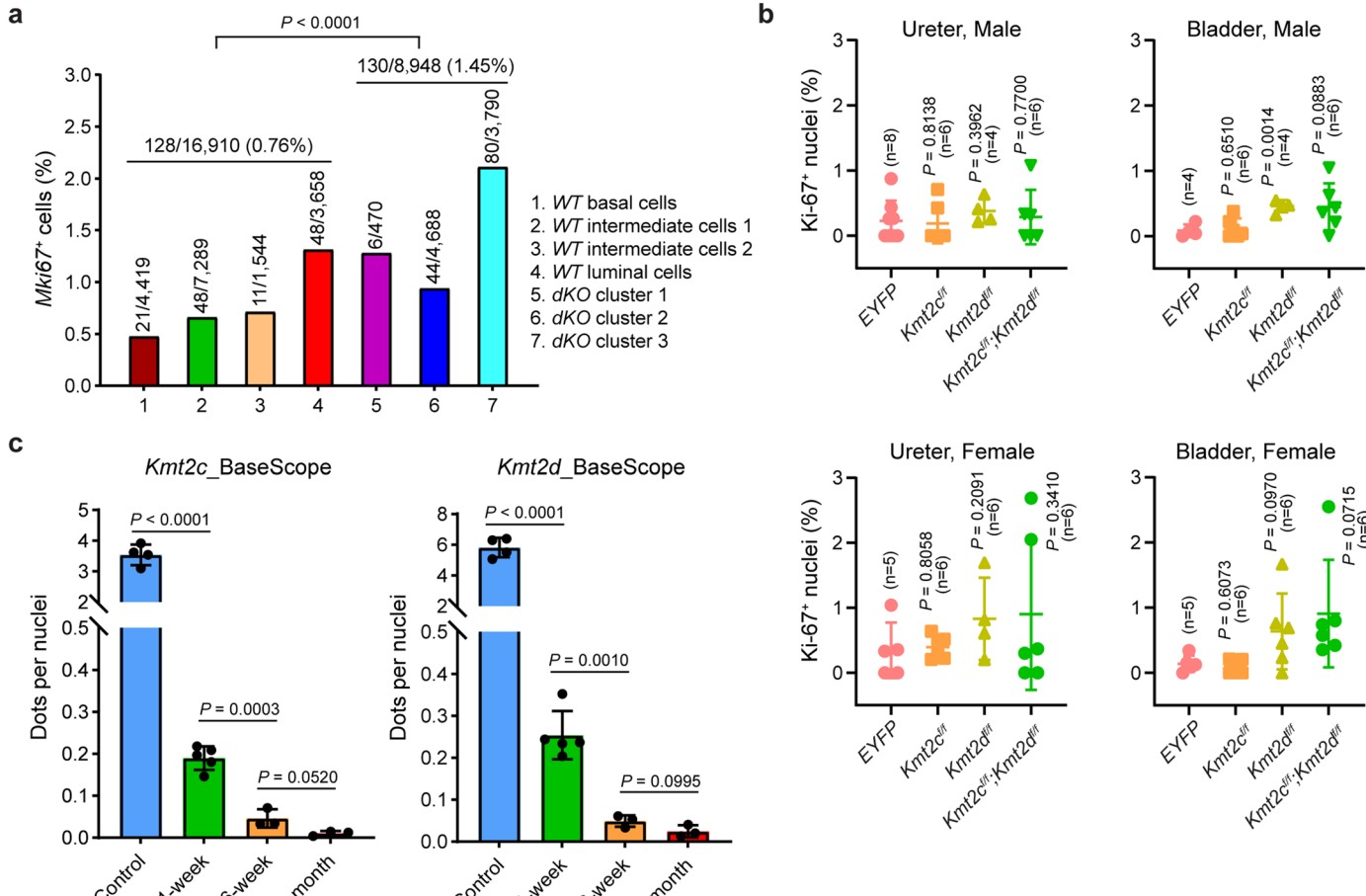

**Extended Data Fig. 2 | Urothelial cells with *Kmt2c/d* loss demonstrate growth advantage. a**, Quantification of *Mki67* positivity in cell clusters from scRNA-seq (n = 4 mice in *WT*, n = 3 mice in *dKO*). Mki67 positivity was defined as detectable reads higher than 1. Data were analyzed with two-tailed Chi-Squared test. **b**, Quantification of Ki-67 positive cells (IHC) in mouse bladder and ureter tissues collected 6 months post tamoxifen administration. Each dot indicates the Ki-67 positive ratio of multiple sections from one mouse. Data were presented as mean ± SD and analyzed with two-tailed t-test between *EYFP* and each indicated group. **c**, Quantification of BaseScope staining with probes targeting *Kmt2c* or *Kmt2d* floxed exons. Bladder tissues collected from *EYFP* groups were used as control (6 months, n = 4 mice). Bladder tissues from *Tmprss2-CreER^T2^;Kmt2c^f/f^;Kmt2d^f/f^* mice were collected 1 week (n = 5 mice), 6 weeks (n = 3 mice), and 6 months (n = 3 mice) post tamoxifen injection (2 x 3 mg). Data were presented as mean ± SD and analyzed with two-tailed t-test.

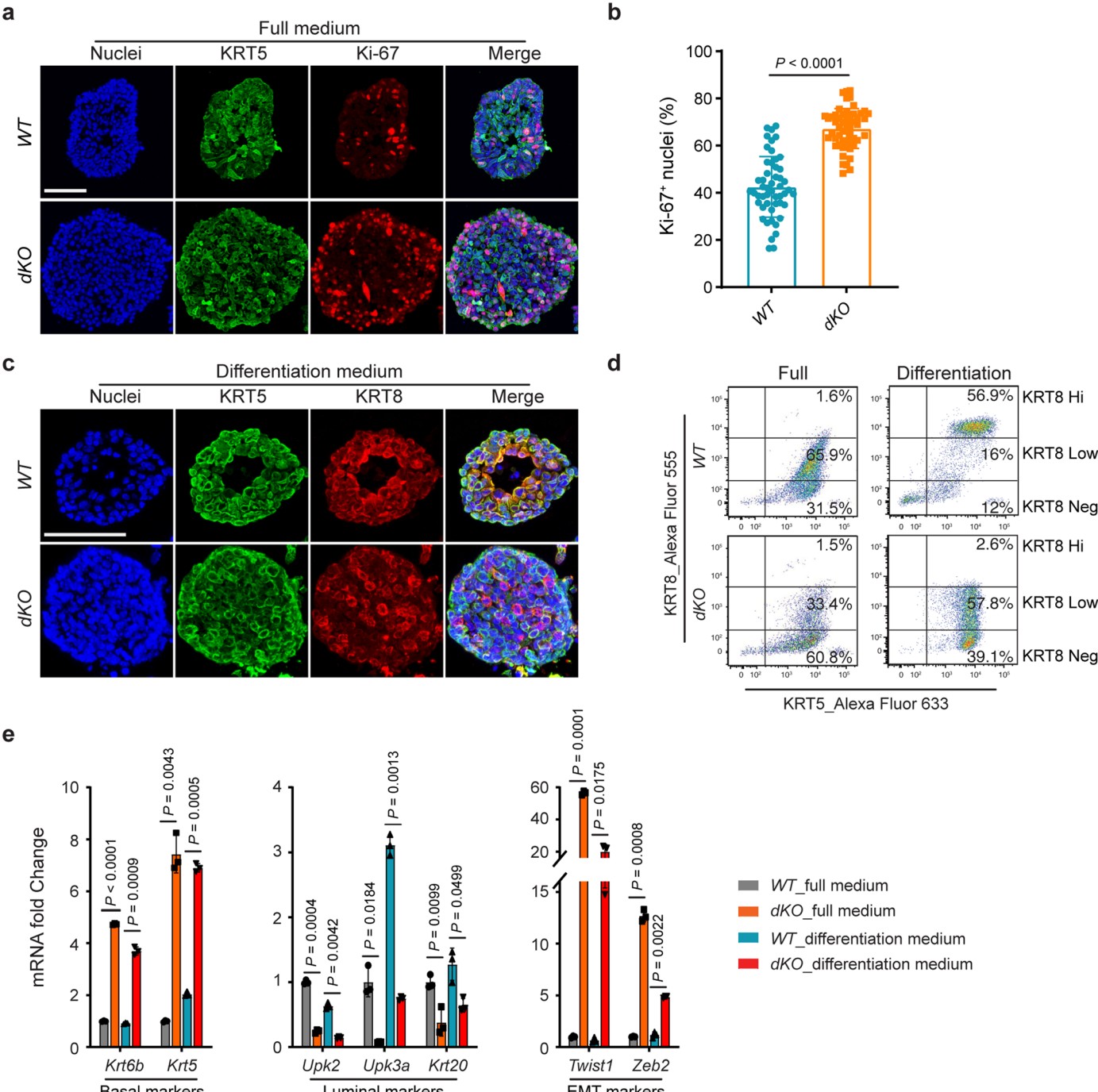

**Extended Data Fig. 3 | Characterization of cell proliferation, differentiation, and EMT in urothelial organoids. a**, Representative immunofluorescence staining of KRT5 and Ki-67 in *WT* (n = 4 mice) and *dKO* (n = 4 mice) urothelial organoid sections. Cell nuclei were counterstained with DAPI (blue). Scale bar, 50μm. **b**, Quantification of Ki-67 positive nuclei in *WT* and *dKO* urothelial organoid. Each dot indicates Ki-67 positivity in each organoid section (pooled from n = 4 mice in each group). Data were presented as mean ± SD and analyzed with two-tailed t-test. **c**, Representative immunofluorescence staining of KRT5 and KRT8 in *WT* (n = 4 mice) and *dKO* (n = 4 mice) urothelial organoid sections.

Organoids were seeded in full medium overnight and then cultured for 9 days with differentiation medium. Cell nuclei were counterstained with DAPI (blue). Scale bar, 50μm. **d**, Flow cytometry analysis of KRT5 and KRT8 expression in urothelial cells cultured under full and differentiation conditions (n = 2 independent experiments). **e**, Quantitative RT-PCR comparing differentiation and EMT markers between *WT* and *dKO* organoids (representative of n = 3 independent experiments). Data were presented as mean ± SD and analyzed with two-tailed t-test.

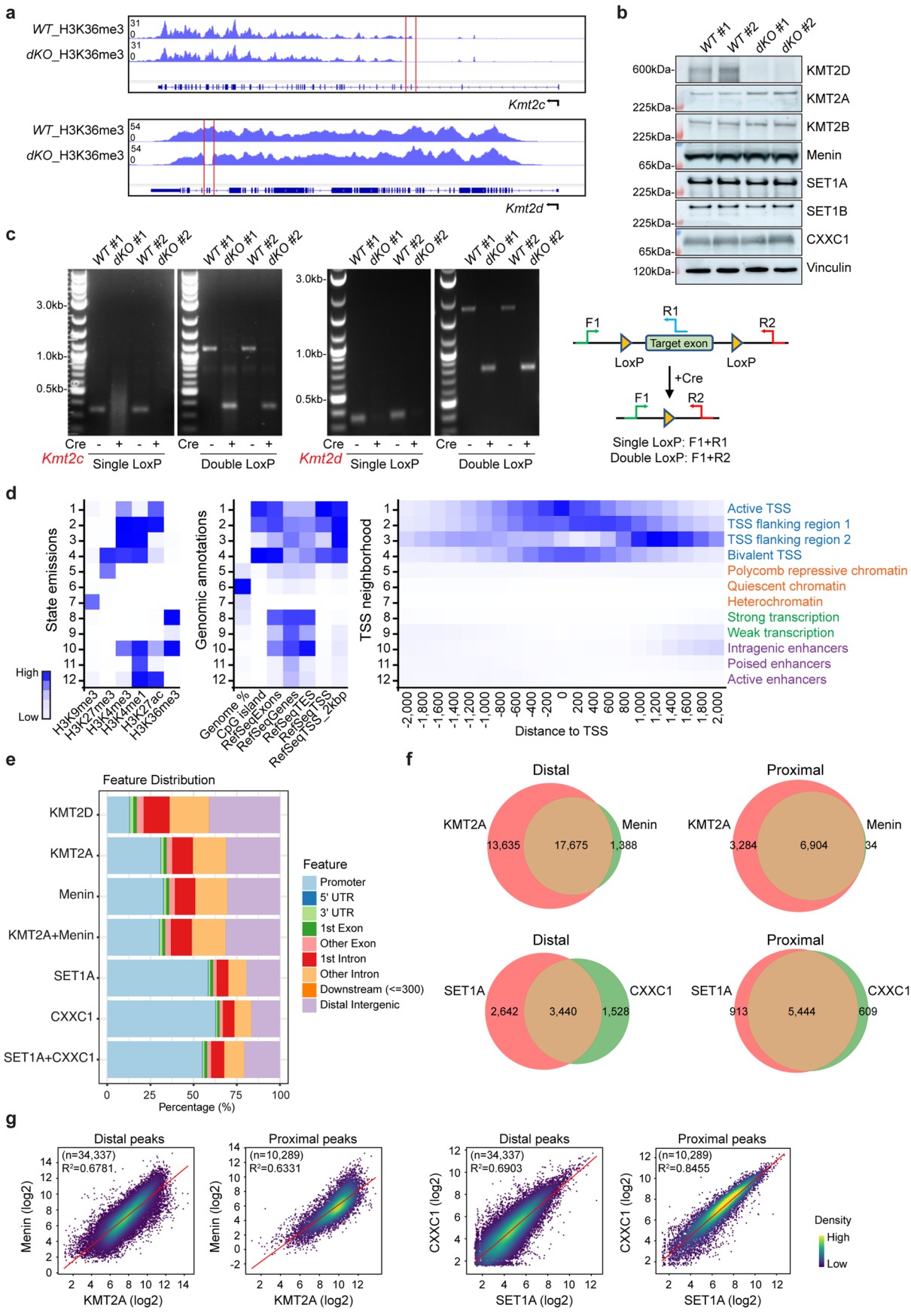

**Extended Data Fig. 4 | See next page for caption.**

**Extended Data Fig. 4 | Global chromatin state and deposition of KMT2 components in mouse urothelial cells. a**, Representative IGV tracks of H3K36me3 Cut&Run indicating the successful depletions of Exon 3 in *Kmt2c* and Exons 50-51 in *Kmt2d*. **b**, Western blot of KMT2 components in *WT* and *dKO* urothelial cells (n = 2 independent experiments). **c**, Genotyping of *Kmt2c* and *Kmt2d* floxed alleles in *WT* and *dKO* urothelial cells (n = 3 independent experiments). Primers amplifying single LoxP were used to only detect intact allele. Primers amplifying double LoxP were used to identify both intact (long) and deleted (short) allele. **d**, Overview of ChromHMM in *WT* urothelial cells.

Left, a darker blue color corresponds to a higher probability of observing specific modifications in each chromatin state. Middle and right, a darker blue color corresponds to a higher fold enrichment of chromatin state at the given coordinates. State annotations were shown on the right side of the heatmap. **e**, Genomic distributions of KMT2 components. KMT2A+Menin and SET1A + CXXC1 groups show the features of combined peaks. **f**, Overlap of KMT2A and Menin, SET1A and CXXC1 on promoter-distal and promoter-proximal peaks. **g**, Comparison of tag-counts (log2) of KMT2A and Menin, SET1A and CXXC1 at promoter-proximal and promoter-distal peaks.

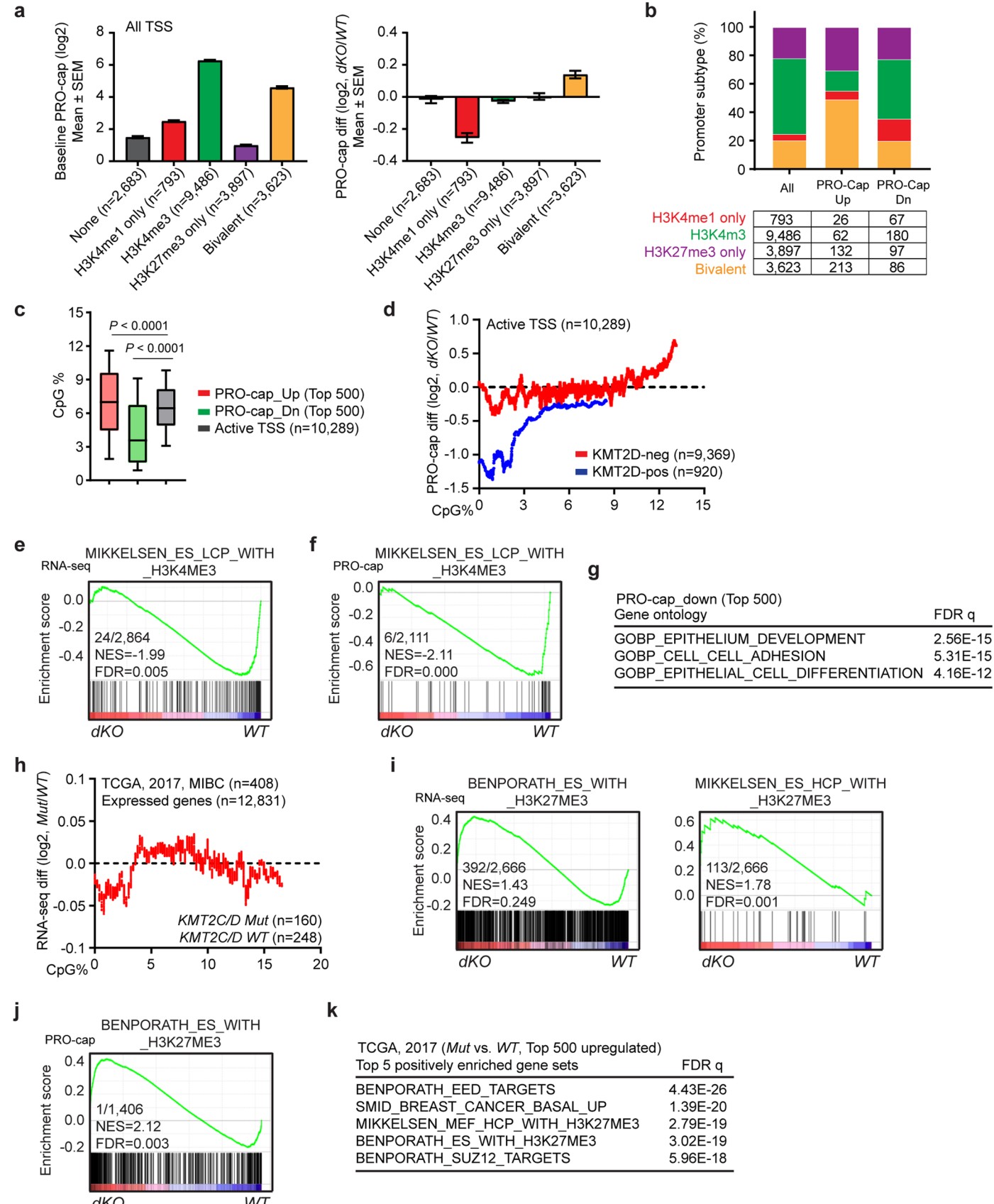

**Extended Data Fig. 5 | See next page for caption.**

**Extended Data Fig. 5 | Direct and indirect regulation of promoter activity by *Kmt2c/d* knockout. a**, Left, PRO-cap baseline signal in *WT* urothelial cells at TSS subtypes. Right, PRO-cap signal change (Log2 diff) in *dKO* urothelial cells. TSS were categorized by peak overlap and subtype as H3K4me1/H3K4me3/H3K27me3 negative (n = 2,683), H3K4me1 only (n = 793), H3K4me3 (may have H3K4me1 at shores) (n = 9,486), H3K27me3 only (n = 3,897), and bivalent promoters (H3K4me3/H3K27me3 double positive, n = 3,623). Data were shown as mean ± SEM. **b**, Distribution of promoter subtypes in H3K4me1 only (n = 793), H3K4me3 (n = 9,486), H3K27me3 only (n = 3,897), and bivalent promoters (H3K4me3/H3K27me3 double positive, n = 3,623). **c**, Fraction of CpG dinucleotide on proximal KMT2D, KMT2A/Menin, or SET1A/CXXC1 peaks with top 500 up- or down- regulated PRO-cap signal. The center line represents the median; the box limits represent the upper and lower quartiles; the minimum and maximum whiskers represent 10 and 90 percentile. Data were analyzed with two-tailed t-test. **d**, Log2 fold change of PRO-cap signal at KMT2D-bound and non-KMT2D-bound TSS with respect to CpG content. Active TSS was ranked by CpG% within proximal KMT2D, KMT2A/Menin, or SET1A/CXXC1 peaks. Bin size of 100 was then used to smoothen the log2 fold change of PRO-cap signal. **e** and **f**, Representative GSEA analyses of RNA-seq and PRO-cap showing negatively enriched gene sets with *Kmt2c/d* knockout. LCP refers to low CpG promoters. **g**, GSEA of top 500 downregulated PRO-cap signal with *Kmt2c/d* loss. **h**, Log2 fold change of RNA-seq in bladder cancer samples in *Mut* (*KMT2C* or *KMT2D* mutations) and *WT* group. Genes were ranked by CpG% within TSS ± 250 bp. Bin size of 100 was then used to smoothen the log2 fold change of RNA-seq. **i** and **j**, Representative GSEA analyses of RNA-seq and PRO-cap showing positively enriched gene sets with *Kmt2c/d* knockout. HCP refers to high CpG promoters. **k**, GSEA of top 500 upregulated genes between *Mut* (*KMT2C* or *KMT2D* mutations) and *WT* in human MIBC dataset (TCGA, 2017).

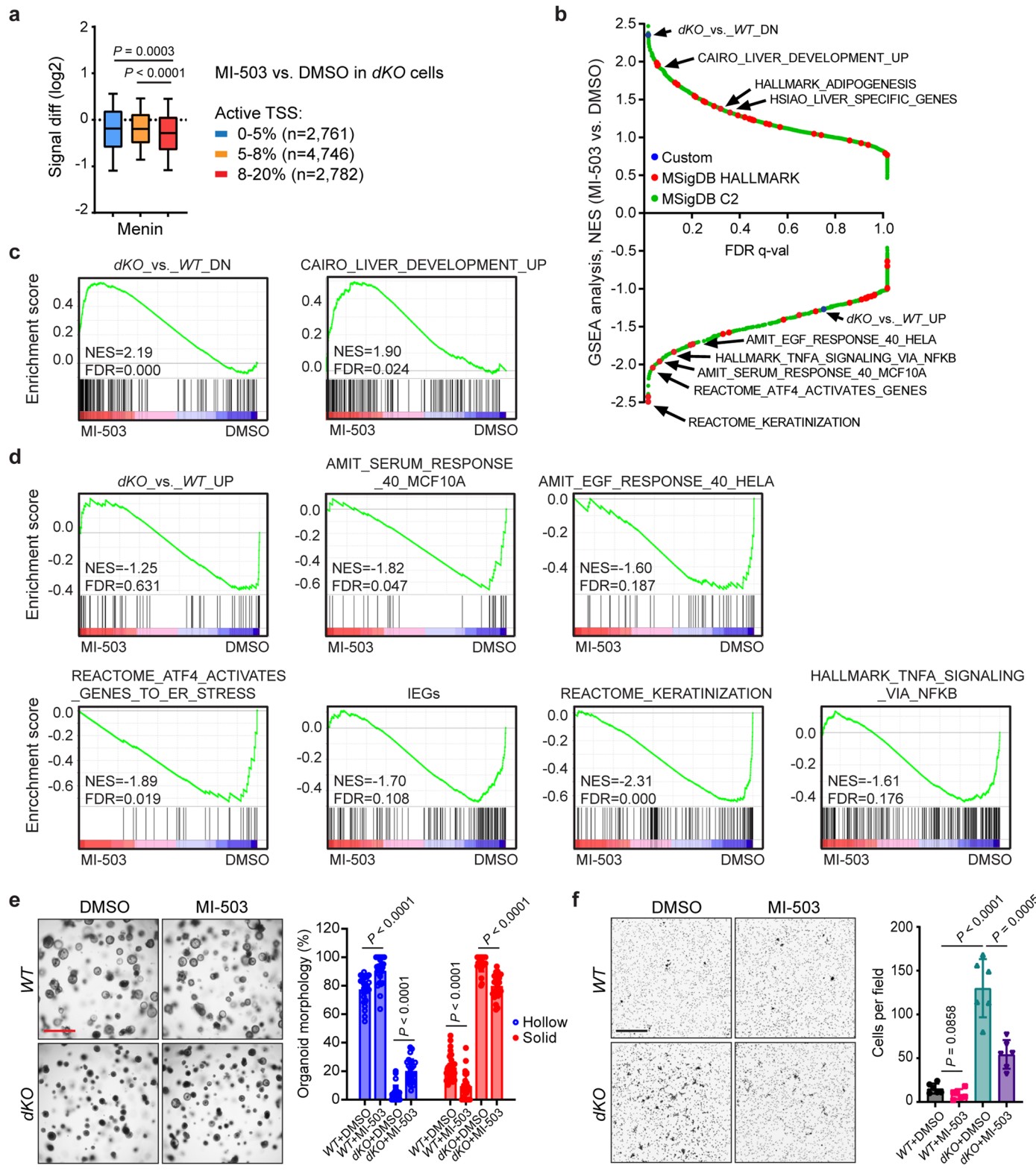

**Extended Data Fig. 6 | See next page for caption.**

**Extended Data Fig. 6 | Blockade of KMT2A/B-Menin partially rescues the transcriptome in *dKO* urothelial cells. a**, Log2 fold change of Menin deposition with MI-503 treatment (1 μM, 4 days) in *Kmt2c/d dKO* urothelial cells. The center line represents the median; the box limits represent the upper and lower quartiles; the minimum and maximum whiskers represent 10 and 90 percentile. Data were analyzed with two-tailed t-test. **b**, Plot of NES vs. FDR q-value of GSEA analyses with MSigDB Hallmark v7.4, MSigDB C2 v7.4 and custom gene sets. RNA-seq was performed in *Kmt2c/d dKO* cells treated with DMSO or MI-503 (1 μM, 4 days). **c-d**, Representative gene sets positively (c) or negatively (d) enriched with MI-503 (1 μM, 4 days) treatment in *dKO* urothelial cells (MI-503 vs. DMSO). **e**, Left, representative bright-field images of organoid from *WT* and *dKO* groups. Cells were pre-treated with DMSO or MI-503 (1 μM) for 3 days. Images were taken at day 8 with continuous presence of DMSO or MI-503. Scale bar, 100 μm. Right,

statistics of organoid morphology. The fraction of hollow and solid organoids was calculated by counting the number of all organoids in each matrigel bulb (200 cells seeded in each bulb, pooled from n = 3 independent experiments). Data were presented as mean ± SD and analyzed with two-tailed t-test. **f**, Matrigel invasion assay with fluorescence blocking transwell insert (pore size, 8 μm). Cells were pre-treated with DMSO or MI-503 (1 μM) for 2 days. On the top chamber, 100k starvation-treated (48 h in basic DMEM/F12 medium) urothelial cells were seeded with basic DMEM/F12 medium. Complete organoid culture medium was added to the bottom. DMSO or MI-503 was added on both top and bottom chamber for the treatment. Twenty-four hours later, the transwell inserts were fixed, permeabilized, and stained with DAPI. Scale bar, 200 μm. Data were presented as mean ± SD (n = 6 independent experiments) and analyzed with two-tailed t-test.

**a**

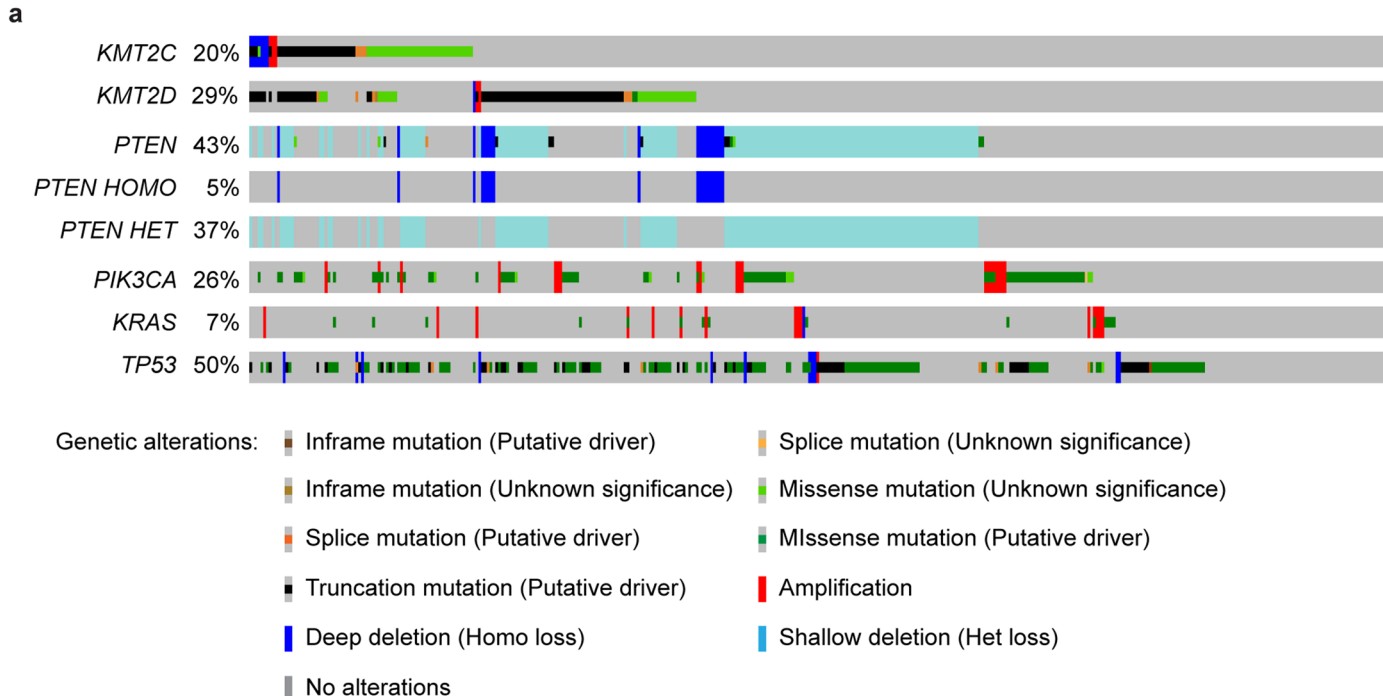

**Extended Data Fig. 7 | Oncoprint of _KMT2C, KMT2D, PTEN, PIK3CA, KRAS_, and _TP53_ in TCGA dataset. a**, Oncoprint of frequently mutated genes in TCGA MIBC dataset (2017).

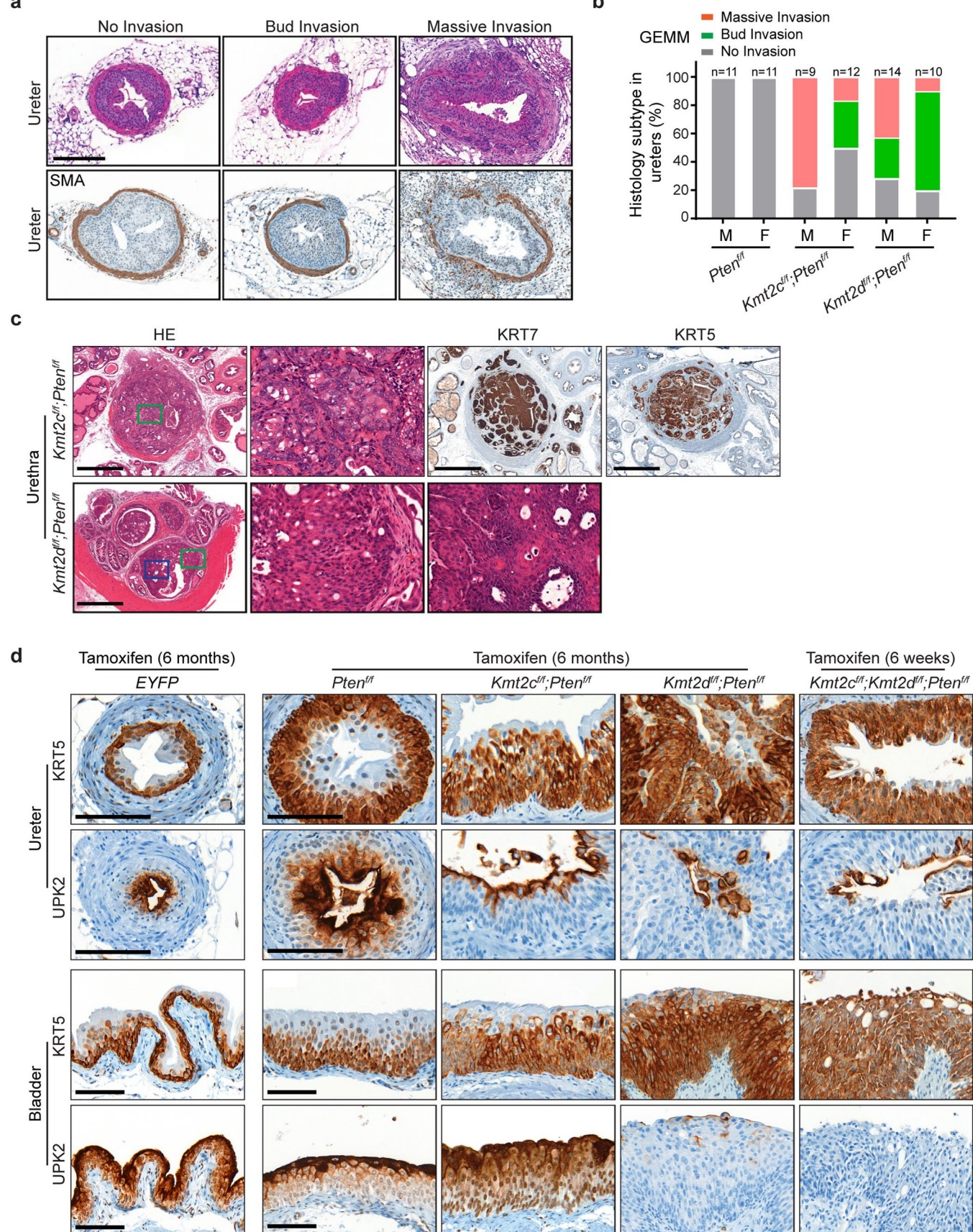

**Extended Data Fig. 8 | See next page for caption.**

**Extended Data Fig. 8 | Characterization of histology and differentiation in urothelial carcinoma GEMMs. a**, Representative histological staining of SMA (alpha Smooth Muscle Actin) in ureter tissues collected 6 months post tamoxifen administration. Scale bar, 200 μm. **b**, Quantification of non-invasion, bud invasion and massive invasion in male and female ureter sections collected 6 months post tamoxifen administration. **c**, Representative H&E staining, KRT5 IHC, and KRT7 IHC of urethral urothelial carcinoma sections collected 6 months after tamoxifen administration (*Kmt2c^{f/f};Pten^{f/f}* n = 4 mice; *Kmt2d^{f/f}; Pten^{f/f}* n = 3 mice). Note that the areas in green squares show features of urothelial carcinoma, whereas the area in blue square from the same section shows features of squamous differentiation. Scale bar, 500 μm. **d**, Representative histological staining of KRT5 and UPK2 in mouse bladder and ureter tissue sections (n = 4 mice in each group). Scale bar, 100 μm.

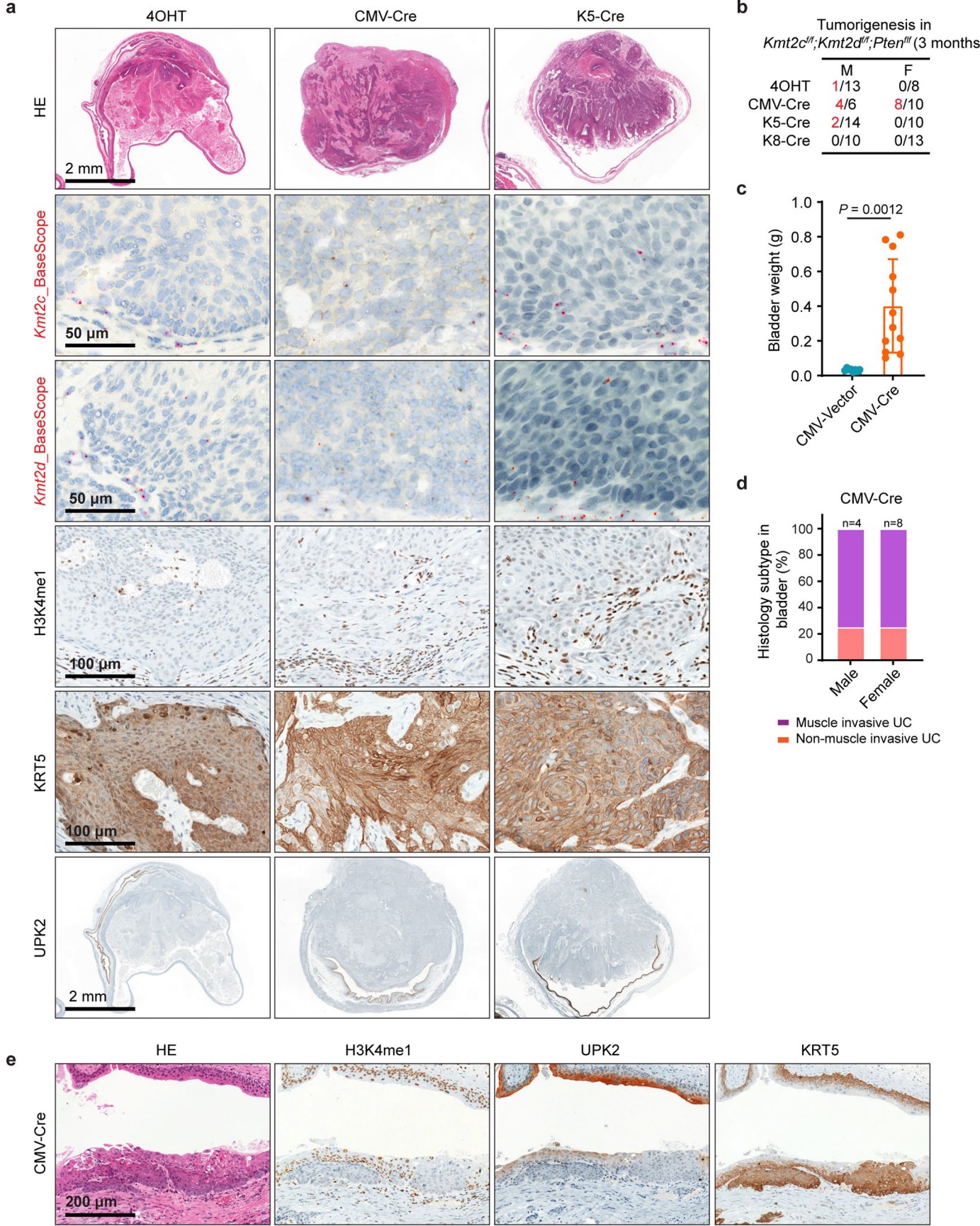

**Extended Data Fig. 9 | See next page for caption.**

**Extended Data Fig. 9 | Bladder specific deletions of *Kmt2c/Kmt2d/Pten* induces muscle invasive urothelial cancer. a**, Representative H&E, *Kmt2c/d* BaseScope, H3K4me1 IHC, KRT5 IHC, and UPK2 IHC staining in bladder tissue sections collected from *Tmprss2-CreER^T2^;Kmt2c^f/f^;Kmt2d^f/f^;Pten^f/f^* mice with 4OHT (n = 1 tumor), adeno-CMV-Cre (n = 2 tumors), or adeno-K5-Cre (n = 2 tumors) intravesical injections. Scale bars were indicated on the figure. **b**, Urothelial cancer efficiency by intravesical delivery of 4OHT or adenovirus. **c**, Statistics of bladder weight 3 months post intravesical adenovirus injection. Control adenovirus (n = 8 mice) or adeno-CMV-Cre (n = 12 mice) were injected into the bladder of *Tmprss2-CreER^T2^;Kmt2c^f/f^;Kmt2d^f/f^;Pten^f/f^* mice. Data were presented as mean ± SD and analyzed with two-tailed t-test. **d**, Histological subtypes of bladder urothelial carcinoma with intravesical adeno-CMV-Cre injections. **e**, Representative H&E, H3K4me1 IHC, KRT5 IHC, and UPK2 IHC staining in bladder tissue sections collected from *Tmprss2-CreER^T2^;Kmt2c^f/f^;Kmt2d^f/f^;Pten^f/f^* mice with adeno-CMV-Cre intravesical injection (n = 2 tumors). Compared to H3K4m1 positive cells, lower UPK2 expression and higher KRT5 expression were observed in H3K4me1 negative cells. Scale bar was indicated on the figure.

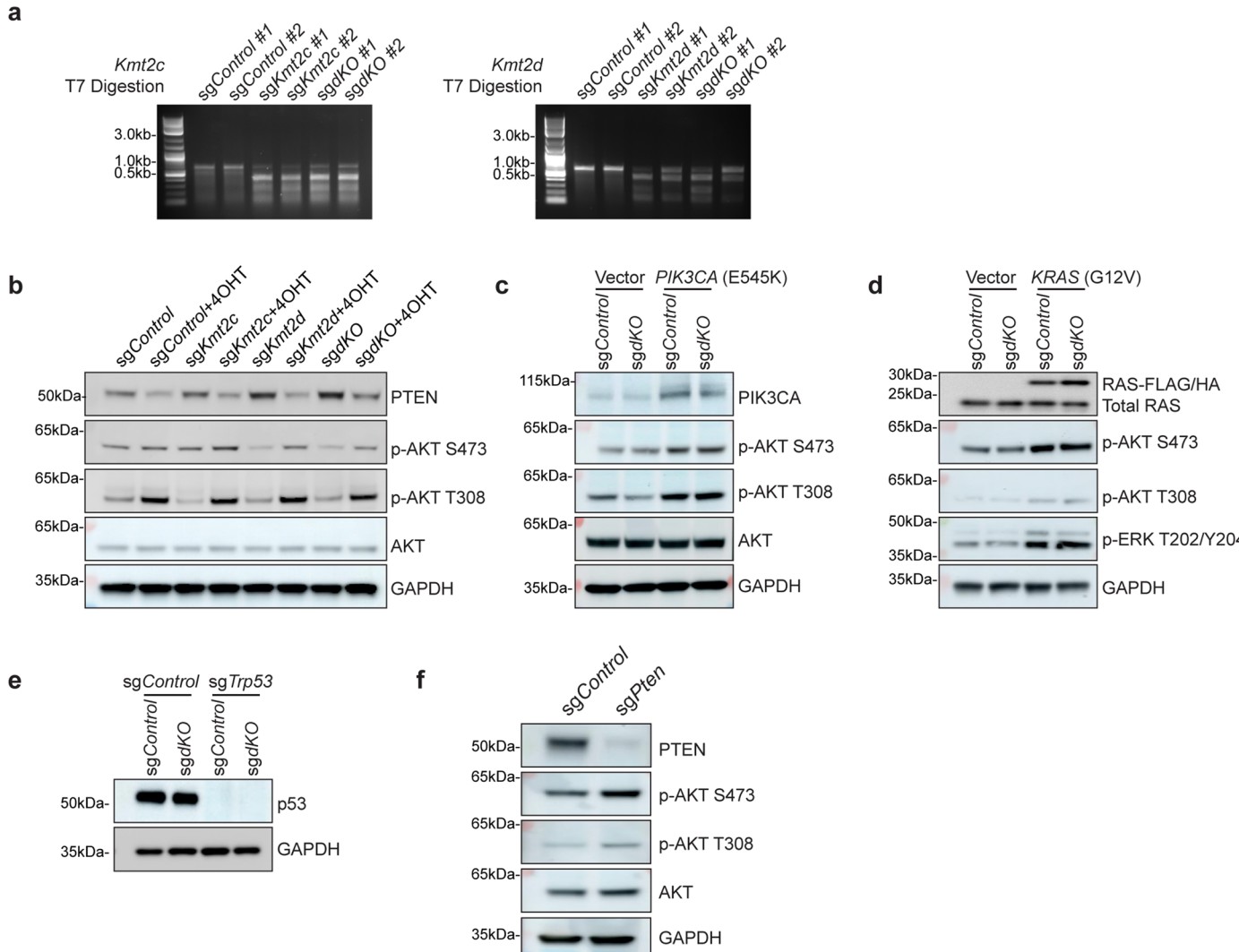

**Extended Data Fig. 10 | Validation of genetic manipulations in *WT* and *dKO* urothelial cells. a**, Surveyor assay validating the successful CRISPR editing of *Kmt2c* and *Kmt2d* (n = 3 independent experiments). Guide RNAs *sgKmt2c* #1 and *sgKmt2d* #1 were picked for the following experiments. **b**, Western blot validation of *Pten* deletion in sg*Control*, sg*Kmt2c*, sg*Kmt2d*, and sg*dKO* urothelial cells

(n = 2 independent experiments). **c-e**, Western blot validation of engineered transgene *PIK3CA*, *KRAS*, and knockout of *Trp53* in sg*Control* and sg*dKO* urothelial cells (n = 2 independent experiments). **f**, Western blot validation of CRISPR/Cas9-mediated *Pten* deletion (n = 2 independent experiments).

# Reporting Summary

## Statistics

For all statistical analyses, confirm that the following items are present in the figure legend, table legend, main text, or Methods section.

| n/a | Confirmed | |
|---|---|---|
| ☐ | ☒ | The exact sample size (*n*) for each experimental group/condition, given as a discrete number and unit of measurement |
| ☐ | ☒ | A statement on whether measurements were taken from distinct samples or whether the same sample was measured repeatedly |
| ☐ | ☒ | The statistical test(s) used AND whether they are one- or two-sided<br>*Only common tests should be described solely by name; describe more complex techniques in the Methods section.* |
| ☒ | ☐ | A description of all covariates tested |
| ☐ | ☒ | A description of any assumptions or corrections, such as tests of normality and adjustment for multiple comparisons |
| ☐ | ☒ | A full description of the statistical parameters including central tendency (e.g. means) or other basic estimates (e.g. regression coefficient) AND variation (e.g. standard deviation) or associated estimates of uncertainty (e.g. confidence intervals) |
| ☐ | ☒ | For null hypothesis testing, the test statistic (e.g. *F*, *t*, *r*) with confidence intervals, effect sizes, degrees of freedom and *P* value noted<br>*Give P values as exact values whenever suitable.* |
| ☒ | ☐ | For Bayesian analysis, information on the choice of priors and Markov chain Monte Carlo settings |
| ☒ | ☐ | For hierarchical and complex designs, identification of the appropriate level for tests and full reporting of outcomes |
| ☐ | ☒ | Estimates of effect sizes (e.g. Cohen's *d*, Pearson's *r*), indicating how they were calculated |

*Our web collection on statistics for biologists contains articles on many of the points above.*

## Software and code

Policy information about availability of computer code

| Data collection | Histology and organoids: Mirax Digital Slide Scanner, Leica TCS SP5 upright confocal microscope, Nikon ECLIPSE Ti2 inverted microscope;<br>Flow cytometry: BD LSRFortessa, BD FACSymphony S6 Cell Sorter;<br>Luminesent imaging: IVIS Spectrum imaging system;<br>scRNA-seq: Illumina NovaSeq platform S4 flow cell;<br>RNA-seq, PRO-cap, ChIP-seq, Cut&Run, and ATAC-seq: Illumina platforms HiSeq4000 and NovaSeq6000, ;<br>PCR reaction: Applied Biosystems QuantStudio 7 Flex Real-Time PCR machine;<br>Western blot: Amersham ImageQuant 800 biomolecular imager;<br>Histone PTMs: Thermo Scientific TSQ Quantum Ultra mass spectrometer, UltiMate 3000 Dionex nano-liquid chromatography; |
|---|---|
| Data analysis | Flow cytometry data were analyzed with BD FACSDiva software v6.2 and FlowJo 10.7.1;<br>IHC and BaseScope quantification: QuPath-0.4.4;<br>scRNA-seq data were processed with Cell Ranger (7.0.0). True cells were identified using scCB2 package (1.14.0). Putative doublets were detected and filtered out using doublet detection package (4.2). Downstream analyses and figure plotting were performed using Scanpy (1.6.1). Differentially expressed genes were compared with MAST package (1.30.0). Imputed data were used to generate heatmap using MAGIC (3.0.0) package.<br>RNA-seq data were processed with STAR (2.7.10b);<br>PRO-cap data were mapped with STAR (2.7.10b). Peaks were called with PINTS 1.1.8 (pints_caller). Bigwig files of plus and minus strand alignments were generated with PINTS 1.1.8 (pints_visualizer);<br>GSEA analyses were performed using JAVA GSEA 4.1.0 program;<br>TCGA human MIBC data were analyzed with ssGSEA v4.0;<br>Data from ATAC-seq, ChIP-seq, Cut&Run were trimmed with trim_galore and mapped to GRCm38(mm10) using bowtie2 (2.4.5). Duplicates |

were marked with samblaster (0.1.26). Mapping quality was analyzed with qualimap (2.2.2-dev). Peaks were called using MACS3 (3.0.0). Bigwig files were generated with bamCoverage (3.5.1). Read counts were measured with featureCounts (v2.0.1); Heatmap and aggregation plots were generated using deepTools (3.5.1).
In KMT2D ChIP-seq, Sicer2 (1.0.3) (sicer_df) was used to call differential peaks between WT and dKO conditions;
Homer (4.11.1) was employed to merge (Homer mergePeaks) and annotate (Homer annotatepeaks.pl) peaks from PRO-cap, ATAC-seq, ChIP-seq, and Cut&Run;
ChromHMM (v1.25) LearnModel was performed to investigate chromatin state in WT and dKO cells. ChromHMM OverlapEnrichment was conducted to compare enrichment of chromatin states at given coordinates.
Histone PTMs were analyzed with Skyline.

For manuscripts utilizing custom algorithms or software that are central to the research but not yet described in published literature, software must be made available to editors and reviewers. We strongly encourage code deposition in a community repository (e.g. GitHub). See the Nature Portfolio guidelines for submitting code & software for further information.

## Data

Policy information about availability of data

All manuscripts must include a data availability statement. This statement should provide the following information, where applicable:
- Accession codes, unique identifiers, or web links for publicly available datasets
- A description of any restrictions on data availability
- For clinical datasets or third party data, please ensure that the statement adheres to our policy

Raw sequencing data are publicly available from Gene Expression Omnibus: GSE180947, GSE236370, and GSE264514. Raw mass spectrometry data of histone post translational modifications are available via ProteomeXchange with identifier PXD056439. All other data supporting the findings of this study are available upon request from the corresponding authors.
This study did not use custom code or software. All code used in the study has been published and has been cited in the relevant section of the Methods.

## Research involving human participants, their data, or biological material

Policy information about studies with human participants or human data. See also policy information about sex, gender (identity/presentation), and sexual orientation and race, ethnicity and racism.

| | |
|---|---|
| Reporting on sex and gender | N/A |
| Reporting on race, ethnicity, or other socially relevant groupings | N/A |
| Population characteristics | N/A |
| Recruitment | N/A |
| Ethics oversight | N/A |

Note that full information on the approval of the study protocol must also be provided in the manuscript.

# Field-specific reporting

Please select the one below that is the best fit for your research. If you are not sure, read the appropriate sections before making your selection.

☒ Life sciences          ☐ Behavioural & social sciences          ☐ Ecological, evolutionary & environmental sciences

For a reference copy of the document with all sections, see nature.com/documents/nr-reporting-summary-flat.pdf

# Life sciences study design

All studies must disclose on these points even when the disclosure is negative.

| | |
|---|---|
| Sample size | Sample sizes were not predetermined by any statistical methods. In our experiments, sample sizes were similar to prior studies (PMID: 37084735) and indicated in the figures or figure legends. |
| Data exclusions | No data were excluded from the analyses. |
| Replication | Experiments were successfully repeated with a minimum of two independent experiments. In most cases, we performed multiple experiments to address the same scientific question. |
| Randomization | For the genetically engineered mouse models, we randomly gave tamoxifen to both male and female mice with the matched ages. In the therapeutic experiments, tumor-bearing mice were randomized before the treatment to ensure comparable tumor sizes among groups at the beginning timepoint. |

| Blinding | Next-generation sequencing data were obtained and analyzed unbiasedly in this study. Mouse treatment and tumor measurements were performed by the same person blinded to the therapeutic effects. Statistics of Ki67 positivity were conducted with the whole bladder sections using QuPath program. |
|---|---|

# Reporting for specific materials, systems and methods

We require information from authors about some types of materials, experimental systems and methods used in many studies. Here, indicate whether each material, system or method listed is relevant to your study. If you are not sure if a list item applies to your research, read the appropriate section before selecting a response.

## Materials & experimental systems

| n/a | Involved in the study |
|---|---|
| ☐ | ☒ Antibodies |
| ☒ | ☐ Eukaryotic cell lines |
| ☒ | ☐ Palaeontology and archaeology |
| ☐ | ☒ Animals and other organisms |
| ☒ | ☐ Clinical data |
| ☒ | ☐ Dual use research of concern |
| ☒ | ☐ Plants |

## Methods

| n/a | Involved in the study |
|---|---|
| ☐ | ☒ ChIP-seq |
| ☐ | ☒ Flow cytometry |
| ☒ | ☐ MRI-based neuroimaging |

## Antibodies

Antibodies used

KRT5, #905501, Biolegend (1:500 in IHC, 1:400 in IF, 1:400 in Flow)
KRT5, #905901, Biolegend (1:400 in IF)
KRT7, #ab181598, lot# GR3214132-2, Abcam (1:1000 in IHC)
KRT8, #904801, Biolegend (1:400 in IF, 1:400 in Flow)
KRT14, #906004, Biolegend (1:400 in IF)
KRT20, #M7091, clone# KS20.8, lot# 20046893, Dako Omnis (1:500 in IF)
UPK2, #ab213655, lot# GR284813-6, Abcam (1:500 in IHC and IF)
PTEN, #9188, clone# D4.3, lot# 6, Cell Signaling Technology (1:100 in IHC, 1:2000 in WB)
GFP, #2956, clone# D5.1, lot# 6, Cell Signaling Technology (1:200 in IHC)
Ki-67, #ab16667, Abcam (1:100 in IHC, 1:100 in IF)
SMA, #ab5694, lot# GR3356867-5, Abcam (1:1000 in IHC)
p-EGFR Y845, #2231, lot# 8, Cell Signaling Technology (1:50 in IHC)
H-2Kb/Db-APC, #114614, clone# 28-8-6, lot# B312433, Biolegend (1:200 in flow cytometry)
PD-L1-APC, #124311, clone# 10F.9G2, lot# B357778, Biolegend (1:200 in flow cytometry)
EpCAM-APC, #17579180, clone# G8.8, lot# 2202308, Thermo Fisher Scientific (1:200 in flow cytometry)
H3K4me1, #710795, lot# 1998633, Thermo Fisher Scientific (1ug per 250k cells in Cut&Run)
H3K4me1, #5326, clone# D1A9, lot# 5, Cell Signaling Technology (1:100 in IHC)
H3K4me2, #710796, lot# 2059496, Thermo Fisher Scientific (1ug per 250k cells in Cut&Run)
H3K4me3, #PA57-27029, lot# TI4042096A, Thermo Fisher Scientific (1ug per 250k cells in Cut&Run)
H3K27ac, #ab4729, lot# GR3231887-1, Abcam (2ug per 10million cells in ChIP)
H3K27me3, #9733, clone#C36B11, lot# 16, Cell Signaling Technology (1ug per 250k cells in Cut&Run)
H3K9me3, #ab176916, lot# 1011476-14, Abcam (1ug per 250k cells in Cut&Run)
H3K36me3, #61021, clone# 0333, lot# 11721013, Active Motif (1ug per 250k cells in Cut&Run)
SET1A, #ab70378, lot# GR3352449-5, Abcam (1ug per 250k cells in Cut&Run)
SET1A, #50805, clone# E3E2S, lot# 1, Cell Signaling Technology (1:1000 in WB)
SET1B, #44922, clone# D1U5D, lot# 1, Cell Signaling Technology (1:1000 in WB)
CXXC1, #ab198977, lot# GR3245956-8, Abcam (1ug per 250k cells in Cut&Run, 1:1000 in WB)
KMT2A, #A300-086A, lot# 7, Bethyl Laboratories (1ug per 250k cells in Cut&Run)
KMT2A, #14689, clone# D2M7U, lot# 1, Cell Signaling Technology (1:1000 in WB)
KMT2B, #47097, clone# E3M1U, lot# 1, Cell Signaling Technology (1:1000 in WB)
Menin, #A300-105A, lot# 12, Bethyl laboratories (1ug per 250k cells in Cut&Run, 1:1000 in WB)
KMT2D, a kind gift from Dr. Kai Ge's lab (2ug per 10million cells in ChIP, 1:1000 in WB)
GAPDH, #2118, clone# 14C10, lot# 16, Cell Signaling Technology (1:5000 in WB)
Vinculin, #13901, clone# E1E91, lot# 7, Cell Signaling Technology (1:5000 in WB)
PIK3CA, #4249, clone# C73F8, lot# 9, Cell Signaling Technology (1:2000 in WB)
p-AKT S473, #4060, clone# D9E, lot# 24, Cell Signaling Technology (1:2000 in WB)
p-AKT T308, #13038, clone# D25E6, lot#7, Cell Signaling Technology (1:2000 in WB)
AKT, #4691, clone# C67E7, lot#28, Cell Signaling Technology (1:5000 in WB)
p-ERK T202/Y204, #4370, clone# D13.14.4E, lot# 28, Cell Signaling Technology (1:2000 in WB)
RAS, #8832, lot# 9, Cell Signaling Technology (1:500 in WB)
p53, #2524, clone# 1C12, lot# 17, Cell Signaling Technology (1:1000 in WB)
Goat anti-rabbit Alexa Fluor 488, #A11008, lot# 2382186, Thermo Fisher Scientific (1:500 in IF)
Goat anti-rabbit Alexa Fluor 555, #A21428, lot# 2395213, Thermo Fisher Scientific (1:500 in IF)
Goat anti-rabbit Alexa Fluor 633, #A21071, lot# 1073053, Thermo Fisher Scientific (1:500 in IF and Flow)
Goat anti-chicken Alexa Fluor 488, #A11039, lot# 2566343, Thermo Fisher Scientific (1:500 in IF)

Goat anti-mouse Alexa Fluor 488, #A11001, lot# 2379467, Thermo Fisher Scientific (1:500 in IF)
Goat anti-mouse Alexa Fluor 555, #A21422, lot# 2377305, Thermo Fisher Scientific (1:500 in IF and Flow)

**Validation**

All antibodies except anti-KMT2D antibody were obtained and validated from commercially available sources. All antibodies were applied to mouse cells in this study.
KMT2D antibody was generated by Dr. Kai Ge's lab and validated in KMT2D KO mouse cells (PMID: 37012455).
KRT5: https://www.biolegend.com/en-gb/products/keratin-5-polyclonal-antibody-purified-10956
KRT5: https://www.biolegend.com/fr-ch/products/keratin-5-polyclonal-chicken-antibody-purified-10957
KRT7: https://www.abcam.com/en-us/products/primary-antibodies/cytokeratin-7-antibody-epr17078-cytoskeleton-marker-ab181598
KRT8: https://www.biolegend.com/fr-ch/products/purified-anti-cytokeratin-8-antibody-13078?GroupID=GROUP26
KRT14: https://www.biolegend.com/fr-ch/products/purified-anti-keratin-14-antibody-13379
KRT20: https://www.agilent.com/en/product/immunohistochemistry/antibodies-controls/primary-antibodies/cytokeratin-20-(dako-omnis)-76273
UPK2: https://www.abcam.com/en-us/products/primary-antibodies/uroplakin-ii-upii-antibody-epr18799-ab213655
PTEN: https://www.cellsignal.com/products/primary-antibodies/pten-d4-3-xp-rabbit-mab/9188
GFP: https://www.cellsignal.com/products/primary-antibodies/gfp-d5-1-rabbit-mab/2956
Ki-67: https://www.abcam.com/en-us/products/primary-antibodies/ki67-antibody-sp6-ab16667
SMA: https://www.abcam.com/en-us/products/primary-antibodies/alpha-smooth-muscle-actin-antibody-ab5694
p-EGFR Y845: https://www.cellsignal.com/products/primary-antibodies/phospho-egf-receptor-tyr845-antibody/2231
H-2Kb/Db-APC: https://www.biolegend.com/en-ie/products/apc-anti-mouse-h-2kb-h-2db-antibody-16327
PD-L1-APC: https://www.biolegend.com/fr-ch/products/apc-anti-mouse-cd274-b7-h1-pd-l1-antibody-6655
EpCAM-APC: https://www.thermofisher.com/antibody/product/CD326-EpCAM-Antibody-clone-G8-8-Monoclonal/17-5791-82
H3K4me1: https://www.thermofisher.com/antibody/product/H3K4me1-Antibody-Recombinant-Polyclonal/710795
H3K4me1: https://www.cellsignal.com/products/primary-antibodies/mono-methyl-histone-h3-lys4-d1a9-xp-rabbit-mab/5326
H3K4me2: https://www.thermofisher.com/antibody/product/H3K4me2-Antibody-Recombinant-Polyclonal/710796
H3K4me3: https://www.thermofisher.com/antibody/product/H3K4me3-Antibody-Polyclonal/PA5-27029
H3K27ac: https://www.abcam.com/en-us/products/primary-antibodies/histone-h3-acetyl-k27-antibody-chip-grade-ab4729
H3K27me3: https://www.cellsignal.com/products/primary-antibodies/tri-methyl-histone-h3-lys27-c36b11-rabbit-mab/9733
H3K9me3: https://www.abcam.com/en-us/products/primary-antibodies/histone-h3-tri-methyl-k9-antibody-epr16601-chip-grade-ab176916
H3K36me3: https://www.activemotif.com/catalog/details/61021
SET1A: https://www.abcam.com/en-nc/products/primary-antibodies/hset1-set1-antibody-ab70378
SET1A: https://www.cellsignal.com/products/primary-antibodies/set1a-e3e2s-rabbit-mab/50805
SET1B: https://www.cellsignal.com/products/primary-antibodies/set1b-d1u5d-rabbit-mab/44922
CXXC1: https://www.abcam.com/en-us/products/primary-antibodies/cgbp-antibody-epr19199-chip-grade-ab198977
KMT2A: https://www.thermofisher.com/antibody/product/MLL1-Antibody-Polyclonal/A300-086A
KMT2A: https://www.cellsignal.com/products/primary-antibodies/mll1-d2m7u-rabbit-mab-amino-terminal-antigen/14689
KMT2B: https://www.cellsignal.com/products/primary-antibodies/mll2-kmt2b-e3m1v-rabbit-mab-amino-terminal-antigen/47097
Menin: https://www.thermofisher.com/antibody/product/Menin-Antibody-Polyclonal/A300-105A
GAPDH: https://www.cellsignal.com/products/primary-antibodies/gapdh-14c10-rabbit-mab/2118
Vinculin: https://www.cellsignal.com/products/primary-antibodies/vinculin-e1e9v-xp-rabbit-mab/13901
PIK3CA: https://www.cellsignal.com/products/primary-antibodies/pi3-kinase-p110a-c73f8-rabbit-mab/4249
p-AKT S473: https://www.cellsignal.com/products/primary-antibodies/phospho-akt-ser473-d9e-xp-rabbit-mab/4060
p-AKT T308: https://www.cellsignal.com/products/primary-antibodies/phospho-akt-thr308-d25e6-xp-rabbit-mab/13038
AKT: https://www.cellsignal.com/products/primary-antibodies/akt-pan-c67e7-rabbit-mab/4691
p-ERK T202/Y204: https://www.cellsignal.com/products/primary-antibodies/phospho-p44-42-mapk-erk1-2-thr202-tyr204-d13-14-4e-xp-rabbit-mab/4370
RAS: https://www.cellsignal.com/products/cellular-assay-kits/active-ras-detection-kit/8821
p53: https://www.cellsignal.com/products/primary-antibodies/p53-1c12-mouse-mab/2524
Goat anti-rabbit Alexa Fluor 488: https://www.thermofisher.com/antibody/product/Goat-anti-Rabbit-IgG-H-L-Cross-Adsorbed-Secondary-Antibody-Polyclonal/A-11008
Goat anti-rabbit Alexa Fluor 555: https://www.thermofisher.com/antibody/product/Goat-anti-Rabbit-IgG-H-L-Cross-Adsorbed-Secondary-Antibody-Polyclonal/A-21428
Goat anti-rabbit Alexa Fluor 633: https://www.thermofisher.com/antibody/product/Goat-anti-Rabbit-IgG-H-L-Highly-Cross-Adsorbed-Secondary-Antibody-Polyclonal/A-21071
Goat anti-chicken Alexa Fluor 488: https://www.thermofisher.com/antibody/product/Goat-anti-Chicken-IgY-H-L-Secondary-Antibody-Polyclonal/A-11039
Goat anti-mouse Alexa Fluor 488: https://www.thermofisher.com/antibody/product/Goat-anti-Mouse-IgG-H-L-Cross-Adsorbed-Secondary-Antibody-Polyclonal/A-11001
Goat anti-mouse Alexa Fluor 555: https://www.thermofisher.com/antibody/product/Goat-anti-Mouse-IgG-H-L-Cross-Adsorbed-Secondary-Antibody-Polyclonal/A-21422

# Animals and other research organisms

Policy information about studies involving animals; ARRIVE guidelines recommended for reporting animal research, and Sex and Gender in Research

**Laboratory animals**

1. Tmprss2-CreERT2-IRES-nlsEGFP (Tmprss2tm1.1(cre/ERT2)Ychen, MGI:5911389) was generated in our lab.
2. Pten flox (Ptentm2.1Ppp, MGI:2679886).
3. Kmt2c flox strain with Exon 3 flanked by LoxP sites was obtained from the Sarat Chandarlapaty's lab.
4. Kmt2d flox strain with Exon 50-51 flanked by LoxP sites was obtained from the Kai Ge's lab.
5. Rosa26-CAG-LSL-EYFP strain was obtained from Jackson Laboratory (B6.Cg-Gt(ROSA)26Sortm3(CAG-EYFP)Hze, Stock No: 007903).
6. Female NOD-SCID mice (6-8 weeks old) were obtained from Jackson Laboratory (Strain/Stock: NOD.CB17-Prkdc<scid>, Stock #:

001303).
To induce gene knockout in GEMM, tamoxifen was given to mice of 6-12 weeks old. Intravesical delivery of 4OHT and adenovirus were performed on mice of 6-12 weeks old.
Mice were maintained under 12h light/dark cycle (lights on/off at 6am/pm), with controlled temperature and humidity, and with access to regular chow and sterilized water ad libitum.

| | |
|---|---|
| Wild animals | Wild animals were not used in the study. |
| Reporting on sex | This study included both male and female mice as detailed in the figures or figure legends. |
| Field-collected samples | No field collected samples were used in the study. |
| Ethics oversight | Mouse experiments were conducted under protocol 11-12-027 approved by Institutional Animal Care and Use Committee (IACUC) of MSKCC, New York. |

Note that full information on the approval of the study protocol must also be provided in the manuscript.

# Plants

| | |
|---|---|
| Seed stocks | *Report on the source of all seed stocks or other plant material used. If applicable, state the seed stock centre and catalogue number. If plant specimens were collected from the field, describe the collection location, date and sampling procedures.* |
| Novel plant genotypes | *Describe the methods by which all novel plant genotypes were produced. This includes those generated by transgenic approaches, gene editing, chemical/radiation-based mutagenesis and hybridization. For transgenic lines, describe the transformation method, the number of independent lines analyzed and the generation upon which experiments were performed. For gene-edited lines, describe the editor used, the endogenous sequence targeted for editing, the targeting guide RNA sequence (if applicable) and how the editor was applied.* |
| Authentication | *Describe any authentication procedures for each seed stock used or novel genotype generated. Describe any experiments used to assess the effect of a mutation and, where applicable, how potential secondary effects (e.g. second site T-DNA insertions, mosiacism, off-target gene editing) were examined.* |

# ChIP-seq

## Data deposition

☒ Confirm that both raw and final processed data have been deposited in a public database such as GEO.

☒ Confirm that you have deposited or provided access to graph files (e.g. BED files) for the called peaks.

| | |
|---|---|
| Data access links<br>*May remain private before publication.* | https://www.ncbi.nlm.nih.gov/geo/query/acc.cgi (GSE180947, GSE236370, GSE264514) |
| Files in database submission | GSM8219288 RNAseq_WT_DMSO_Rep1<br>GSM8219289 RNAseq_WT_DMSO_Rep2<br>GSM8219290 RNAseq_WT_DMSO_Rep3<br>GSM8219291 RNAseq_WT_MI503_Rep1<br>GSM8219292 RNAseq_WT_MI503_Rep2<br>GSM8219293 RNAseq_WT_MI503_Rep3<br>GSM8219294 RNAseq_dKO_DMSO_Rep1<br>GSM8219295 RNAseq_dKO_DMSO_Rep2<br>GSM8219296 RNAseq_dKO_DMSO_Rep3<br>GSM8219297 RNAseq_dKO_MI503_Rep1<br>GSM8219298 RNAseq_dKO_MI503_Rep2<br>GSM8219299 RNAseq_dKO_MI503_Rep3<br>GSM8219300 ATACseq_WT_Rep1<br>GSM8219301 ATACseq_WT_Rep2<br>GSM8219304 ATACseq_dKO_Rep1<br>GSM8219305 ATACseq_dKO_Rep2<br>GSM8219308 ChIPseq_H3K27ac_WT_Rep1<br>GSM8219309 ChIPseq_H3K27ac_WT_Rep2<br>GSM8219311 ChIPseq_H3K27ac_dKO_Rep1<br>GSM8219312 ChIPseq_H3K27ac_dKO_Rep2<br>GSM8219315 CutRun_Kmt2a_WT_Rep1<br>GSM8219316 CutRun_Kmt2a_WT_Rep2<br>GSM8219318 CutRun_Kmt2a_dKO_Rep1<br>GSM8219319 CutRun_Kmt2a_dKO_Rep2<br>GSM8219320 CutRun_H3K4me1_WT_Rep1<br>GSM8219321 CutRun_H3K4me1_WT_Rep2<br>GSM8219330 CutRun_H3K4me1_dKO_Rep1<br>GSM8219331 CutRun_H3K4me1_dKO_Rep2<br>GSM8219323 CutRun_H3K4me2_WT_1<br>GSM8219324 CutRun_H3K4me2_WT_2<br>GSM8219332 CutRun_H3K4me2_dKO_1 |

GSM8219333 CutRun_H3K4me2_dKO_2
GSM8219327 CutRun_H3K27me3_WT_Rep1
GSM8219328 CutRun_H3K27me3_WT_Rep2
GSM5478809 CutRun_H3K27me3_WT_Rep3
GSM8219335 CutRun_H3K27me3_dKO_Rep1
GSM8219336 CutRun_H3K27me3_dKO_Rep2
GSM5478810 CutRun_H3K27me3_dKO_Rep3
GSM5478807 WT_CutRun_H3K4me3_Rep1
GSM7528793 WT_CutRun_H3K4me3_Rep2
GSM5478808 dKO_CutRun_H3K4me3_Rep1
GSM7528794 dKO_CutRun_H3K4me3_Rep2
GSM7528795 WT_MI503_CutRun_H3K4me3
GSM7528796 dKO_MI503_CutRun_H3K4me3
GSM5478811 WT_CutRun_Menin_Rep1
GSM7528797 WT_CutRun_Menin_Rep2
GSM5478812 dKO_CutRun_Menin_Rep1
GSM7528798 dKO_CutRun_Menin_Rep2
GSM7528799 WT_MI503_CutRun_Menin
GSM7528800 dKO_MI503_CutRun_Menin
GSM5478815 WT_scRNAseq_Rep1
GSM5478816 WT_scRNAseq_Rep2
GSM5478817 WT_scRNAseq_Rep3
GSM5478818 WT_scRNAseq_Rep4
GSM5478819 dKO_scRNAseq_Rep1
GSM5478820 dKO_scRNAseq_Rep2
GSM5478821 dKO_scRNAseq_Rep3
GSM7528773 WT_PROcap_Rep1
GSM7528774 WT_PROcap_Rep2
GSM7528775 dKO_PROcap_Rep1
GSM7528776 dKO_PROcap_Rep2
GSM7528777 WT_Kmt2d_ChIP_Rep1
GSM7528778 WT_Kmt2d_ChIP_Rep2
GSM7528779 dKO_Kmt2d_ChIP_Rep1
GSM7528780 dKO_Kmt2d_ChIP_Rep2
GSM7528781 WT_ChIP_Input_Rep1
GSM7528782 WT_ChIP_Input_Rep2
GSM7528783 dKO_ChIP_Input_Rep1
GSM7528784 dKO_ChIP_Input_Rep2
GSM7528801 WT_CutRun_Cxxc1_Rep1
GSM7528802 WT_CutRun_Cxxc1_Rep2
GSM7528803 dKO_CutRun_Cxxc1_Rep1
GSM7528804 dKO_CutRun_Cxxc1_Rep2
GSM7528805 WT_CutRun_Set1a_Rep1
GSM7528806 WT_CutRun_Set1a_Rep2
GSM7528807 dKO_CutRun_Set1a_Rep1
GSM7528808 dKO_CutRun_Set1a_Rep2
GSM7528809 WT_CutRun_H3K9me3
GSM7528810 dKO_CutRun_H3K9me3
GSM7528811 WT_CUTnRUN_H3K36me3
GSM7528812 dKO_CUTnRUN_H3K36me3

Genome browser session
(e.g. UCSC)

Not applicable.

## Methodology

Replicates

WT and dKO, scRNA-seq (n=4 mice in WT, n=3 mice in Kmt2c/d dKO)
WT and dKO, Bulk RNA-seq, DMSO (3 replicates)
WT and dKO, Bulk RNA-seq, MI503 (3 replicates)
WT and dKO, PRO-cap (2 replicates)
WT and dKO, KMT2D (2 replicates in ChIP-seq)
WT and dKO, H3K4me1 (2 replicates in Cut&Run)
WT and dKO, H3K4me2 (2 replicates in Cut&Run)
WT and dKO, H3K4me3 (2 replicates in Cut&Run)
WT and dKO, H3K27ac (2 replicate in ChIP-seq)
WT and dKO, ATAC-seq (3 replicates)
WT and dKO, H3K27me3 (3 replicate in Cut&Run)
WT and dKO, H3K9me3 (1 replicate in Cut&Run)
WT and dKO, H3K36me3 (1 replicate in Cut&Run)
WT and dKO, SET1A (2 replicates in Cut&Run)
WT and dKO, CXXC1 (2 replicates in Cut&Run)
WT and dKO, KMT2A (2 replicates in Cut&Run)
WT and dKO, Menin (2 replicate in Cut&Run)
WT and dKO, Menin_MI503 (1 replicate in Cut&Run)
WT and dKO, H3K4me3_MI503 (1 replicate in Cut&Run)

| | |
|---|---|
| | WT and dKO, Input (3 replicates in ChIP-seq) |
| Sequencing depth | In poly-A bulk RNA-seq, paired end 50bp or 100bp, 30-40 million reads.<br>In scRNA-seq, paired end 28/91, median reads count 12,984 per cell.<br>In PRO-cap, paired end 150bp, 60-70 million reads.<br>In ATAC-seq, paired end 50bp, 40-50 million reads.<br>In ChIP-seq, paired end 50bp or 100bp, 30-40 million reads, or more.<br>In Cut&Run, paired end 50bp or 100bp, 10-20 million reads, or more. |
| Antibodies | H3K4me1, #710795, Thermo Fisher Scientific (1ug per 250k cells in Cut&Run)<br>H3K4me2, #710796, Thermo Fisher Scientific (1ug per 250k cells in Cut&Run)<br>H3K4me3, #PA57-27029, Thermo Fisher Scientific (1ug per 250k cells in Cut&Run)<br>H3K27ac, #ab4729, Abcam (2ug per 10million cells in ChIP)<br>H3K27me3, #9733, Cell Signaling Technology (1ug per 250k cells in Cut&Run)<br>H3K9me3, #ab176916, Abcam (1ug per 250k cells in Cut&Run)<br>H3K36me3, #61021, Active Motif (1ug per 250k cells in Cut&Run)<br>SET1A, #ab70378, Abcam (1ug per 250k cells in Cut&Run)<br>CXXC1, #ab198977, Abcam (1ug per 250k cells in Cut&Run)<br>KMT2A, #A300-086A, Bethyl Laboratories (1ug per 250k cells in Cut&Run)<br>Memin, #A300-115A, Bethyl laboratories (1ug per 250k cells in Cut&Run)<br>KMT2D, a kind gift from Dr. Kai Ge's lab (2ug per 10million cells in ChIP) |
| Peak calling parameters | MACS3 (3.0.0) was used for peak calling in ChIP-seq, ATAC-seq, Cut&Run, -q e-2.<br>PINTS (1.1.8) was used for peak calling in PRO-cap, default parameters. |
| Data quality | All peaks in our study were called with FDR < 0.01. |
| Software | scRNA-seq data were processed with Cell Ranger (7.0.0). True cells were identified using scCB2 package (1.14.0). Putative doublets were detected and filtered out using doublet detection package (4.2). Downstream analyses and figure plotting were performed using Scanpy (1.6.1). Differentially expressed genes were compared with MAST package (1.30.0). Imputated data were used to generate heatmap using MAGIC (3.0.0) package.<br>RNA-seq data were processed with STAR (2.7.10b);<br>PRO-cap data were mapped with STAR (2.7.10b). Peaks were called with PINTS 1.1.8 (pints_caller). Bigwig files of plus and minus strand alignments were generated with PINTS 1.1.8 (pints_visualizer);<br>Data from ATAC-seq, ChIP-seq, Cut&Run were trimmed with trim_galore and mapped to GRCm38(mm10) using bowtie2 (2.4.5). Duplicates were marked with samblaster (0.1.26). Mapping quality was analyzed with qualimap (2.2.2-dev). Peaks were called using MACS3 (3.0.0). Bigwig files were generated with bamCoverage (3.5.1); Read counts were measured with featureCounts (v2.0.1); Heatmap and aggregation plots were generated using deepTools (3.5.1).<br>In KMT2D ChIP-seq, Sicer2 (1.0.3) (sicer_df) was used to call differential peaks between WT and dKO conditions;<br>Homer (4.11.1) was employed to merge (Homer mergePeaks) and annotate (Homer annotatepeaks.pl) peaks from PRO-cap, ATAC-seq, ChIP-seq, and Cut&Run;<br>ChromHMM (v1.25) LearnModel was performed to investigate chromatin state in WT and dKO cells. ChromHMM OverlapEnrichment was conducted to compare enrichment of chromatin states at given coordinates. |

# Flow Cytometry

## Plots

Confirm that:

☒ The axis labels state the marker and fluorochrome used (e.g. CD4-FITC).

☒ The axis scales are clearly visible. Include numbers along axes only for bottom left plot of group (a 'group' is an analysis of identical markers).

☒ All plots are contour plots with outliers or pseudocolor plots.

☒ A numerical value for number of cells or percentage (with statistics) is provided.

## Methodology

| | |
|---|---|
| Sample preparation | For urothelial cell sorting, bladders were dissected out and minced with scalpel, and then processed for 1h digestion with collagenase/hyaluronidase (#07912, STEMCELL Technologies) and 15min digestion with TrypLE (#12605010, Gibco). Dissociated were staining with EpCAM-APC (#17579180, Thermo Fisher Scientific) for 30min on ice. Live single urothelial cells were sorted out by flow cytometry as DAPI-/EpCAM+/nlsEGFP+.<br>For intracellular flow cytometry analysis, organoids were digested with Dispase for 30min and then further dissociated into single cells with TrypLE for 15min on a shaker in the cell culture incubator. Single urothelial cells were then fixed with 4% PFA for 10min and permeabilized with 0.5% Triton-X 100 for 10min. Primary antibodies against KRT5 and KRT8, secondary antibodies Alexa fluor 633 conjugated goat anti-rabbit and Alexa fluor 555 conjugated goat anti-mouse were then applied in order, 30min on ice.<br>For cell surface flow cytometry analysis, fluorescence-conjugated antibodies against H-2Kb/H-2Db-APC (#114614, Biolegend) and PD-L1-APC (#124311, Biolegend) were directly stained with viable cells, 30min on ice. |
| Instrument | BD LSRFortessa, BD FACSymphony S6 Cell Sorter |

| | |
|---|---|
| Software | Data were collected with BD FACSDiva (v6.2) software and analyzed with FlowJo (10.7.1). |
| Cell population abundance | In the cell sorting, viable urothelial cells were determined by double positivity of pan-epithelial cell marker EpCAM-APC and nlsEGFP expression. |
| Gating strategy | FSC and SSC were used to gate dissociated single cells. Cell viability dye (DAPI) was used to identify viable single cells. Isotype control antibodies were used to define background. For cell sorting, viable urothelial cells were gated as EpCAM-APC and nlsEGFP double positive. For cell surface or intracellular marker analyses, the mean fluorescence intensity (MFI) of indicated markers were analyzed on urothelial cells. |

☒ Tick this box to confirm that a figure exemplifying the gating strategy is provided in the Supplementary Information.

