## [Peer Review File · Nature Genetics]

Kmt2c/d loss primes urothelium for tumorigenesis and redistributes KMT2A to bivalent promoters

Corresponding Author: Professor Yu CHEN

This manuscript has been previously reviewed at another journal. This document only contains information relating to versions considered at Nature Genetics.

Version 0:

Decision Letter:

9th Aug 2023

Dear Dr Chen,

Your Article, "Kmt2c/d loss primes urothelium for tumorigenesis and redistributes Menin to bivalent promoters" has now been seen by 3 referees. You will see from their comments copied below that while they find your work of considerable potential interest, they have raised quite substantial concerns that must be addressed. In light of these comments, we cannot accept the manuscript for publication, but would be very interested in considering a revised version that addresses these serious concerns.

We hope you will find the referees' comments useful as you decide how to proceed. If you wish to submit a substantially revised manuscript, please bear in mind that we will be reluctant to approach the referees again in the absence of major revisions. Any revision would need to robustly address the concerns about the specificity of the Cre driver in your GEMM model as a priority, and all technical points. We would also expect all other issues to be addressed in full, either experimentally where possible or textually where appropriate.

If you choose to revise your manuscript taking into account all reviewer and editor comments, please highlight all changes in the manuscript text file. At this stage we will need you to upload a copy of the manuscript in MS Word .docx or similar editable format.

*2) If you have not done so already please begin to revise your manuscript so that it conforms to our Article format instructions, available here. Refer also to any guidelines provided in this letter.

*3) Include a revised version of any required Reporting Summary: <https://www.nature.com/documents/nr-reporting-summary.pdf>

Please be aware of our guidelines on digital image standards.

We would like to thank the reviewers for their detailed and constructive comments. We have now conducted a number of experiments and analyses to address these concerns and improve our manuscript.

1. We have extensively rewritten the manuscript to improve the conciseness and language.
2. To determine if there is a competitive advantage of *Kmt2c/d* deletion in bladder urothelium, which is one criterion for field cancerization, we followed deleted cells by BaseScope over time and found increase in deleted cells from 1 week to 6 weeks to 6 months.
3. To determine if loss of *Kmt2c/d* is needed to proceed *Pten* loss, we performed in vitro deletion of *Kmt2c/d* or of *Pten* using CRISPR/Cas9 followed by grafting and in vivo deletion of the other gene using Tamoxifen. We observed significantly improved tumor formation when *Kmt2c/d* loss occurs first.
4. We repeated all CHIP-seq and CUT&RUN experiments that were not replicated in duplicate.
5. We added CUT&RUN of Kmt2a, which further strengthens our observation initially with Menin and both components of the Kmt2a/b-Menin complex are redistributed from Kmt2d-positive enhancers to CpG-high promoters after *Kmt2c/d* deletion.
6. Given that *Tmprss2-CreER^{T2}* is active in other epithelial cells beyond the bladder, we performed several methods to locally activate Cre in urothelial cells—bladder infusion with 4-hydroxytamoxifen, adenoviral CMV-Cre, adenoviral K5-Cre, and adenoviral K8-Cre. After 3 months, we found invasive UC with all methods except adenoviral K8-Cre.

Reviewers' Comments:

Reviewer #1:

Remarks to the Author:

Wang et al. report on the role of Kmt2c and Kmt2d in the urothelium using a variety of tools including genetic mouse models, chemical carcinogenesis, and genomics. The work is novel, original, and comprehensive. Their integrative epigenetic analysis linking Kmt2c/d to chromatin landscapes and gene regulation is a real tour de force and provides novel clues as to the mechanisms through which these two genes contribute to urothelial cancer. Specifically, they show that Menin plays a crucial role through its relocation to bivalent promoters associated with transcriptional de-repression. The GEMM experiments are also very strong but there is scarce evidence that the deletions are uniform and involve both alleles: only indirect measures of the effect of gene deletion is provided.

General comments

While the work is highly relevant, I have a few general concerns.

1. Most of the epigenetic analysis use an n=1 (for most of their ChIP-seq experiments). I don't think that this is adequate for a paper submitted to Nat Genet. First, it does not allow assessing the reproducibility of the findings; second, it does not allow to perform adequate statistical analyses. This is particularly important considering that multiple alleles are target for Cre but there is not definite evidence that the alleles are consistently and uniformly deleted in all cells. For all critical experiments, they should perform ideally 3 biological replicates; otherwise, all critical data should be supported by targeted analyses (ChIP-Seq or else, as required). They even perform the studies on bulk RNA-Seq only in duplicates, which is not acceptable.

Response:

To address the concerns of the reviewer, we have performed NEW duplicate experiments for all critical next-generation epigenetic sequencing data (ATAC-seq, ChIP-seq of H3K27ac, Cut&Run of H3K4me1, H3K4me2, H3K27me3, and Kmt2a). We also performed NEW triplicate RNA-seq experiments. The NEW replicate data are consistent with the prior non-replicate data. The analyses presented in Figs. 4-5, Extended Data Figs. 4-6, and Supplementary Figs. 4-8 are all redone using the new replicate data.

We defined peaks as those called by MACS3 in both replicates with FDR<0.01. Differential expression, signal enrichment, heatmap, and IGV tracks were shown as average of all replicates in the main Figs. 4g,5g and individual replicates are shown in Supplementary Figs. 5a,7a. For all sequencing data, we ran the principal component analysis (PCA) and found good correlations between duplicates under each condition (**Reviewer 1, Figure 1**). For the representative ChIP-seq data, we have validated the results with ChIP-qPCR (Shown in Supplementary Figs. 6c,8f).

Reviewer 1, Figure 1: PCA analysis of all critical next-generation sequencing data.

2. The Cre driver used in this work is not urothelial-specific; therefore, it is not possible rule out effects from deletion of target genes in other tissues. This should be addressed in the Discussion and the precise tissue activity of the Cre should be indicated.

Response:

We thank the reviewer for pointing out this critical question. As reported previously (PMID: 27536883), *Tmprss2* is highly expressed in urothelial cells of the bladder, the luminal epithelial cells of the prostate, and the epithelial cells of gastrointestinal tract (**Reviewer 1, Figure 2A** from PMID: 27536883). While the expression of *Tmprss2* is not confined to the bladder, its expression within the bladder is specific to urothelial cells. Experimentally, we have not observed *Tmprss2-CreER^{T2}* mediated recombination in any non-urothelial cells, whether assayed by IHC/IF, FACS, or single cell sequencing. Thus, within the bladder, the Cre does not drive recombination in stromal cells that can affect the microenvironment (**Reviewer 1, Figure 2B**, shown in Fig. 1b,g and Extended Data Fig. 1c,d).

To formally determine if urothelial-only Cre-mediated recombination is sufficient to generate tumorigenic phenotype, we performed intravesical injections of 4OHT (4-hydroxytamoxifen), adeno-CMV-Cre (Cre driven by CMV promoter), adeno-K5-Cre (Cre driven by bovine keratin 5 promoter), and adeno-K8-Cre (Cre driven by mouse keratin 8 promoter) in *Tmprss2-CreER^{T2};Kmt2c^{fl/fl};Kmt2d^{fl/fl};Pten^{fl/fl}* mice. Pilot experiments using *Tmprss2-CreER^{T2};LSL-EYFP* mice showed observable but low recombination efficiency in 4OHT group specific to the bladder and no other tissues (**Reviewer 1, Figure 2C**). Despite the diverse tumorigenic efficiency, muscle invasive urothelial cancer was observed in three local delivery methods (**Reviewer 1, Figure 2D**, shown in Extended Data Fig. 9). Consistent with GEMMs and allograft models, increased basal differentiation and decreased luminal differentiation were observed in these tumors (**Reviewer 1, Figure 2D**, shown in Extended Data Fig. 9a,e).

As these are models of bladder specific target gene deletion, we did not see tumorigenesis in adjacent prostate tissue (Reviewer 1, Figure 2E with H&E of prostate and prostate circled in red on gross image) or distal stomach tissue (data not shown).

Reviewer 1, Figure 2A: *Tmprss2* mRNA levels in various mouse tissues (Fig. 3b from PMID: 27536883).

Reviewer 1, Figure 2B: EYFP IHC in *Tmprss2-CreER^{T2};R26-EYFP* mice. Ureter and bladder tissues were collected 1 week post tamoxifen administration. EYFP expression was only observed in urothelial cells but not in adjacent stromal cells (Shown in Fig. 1b)

Reviewer 1, Figure 2C: IP tamoxifen (top row) causes deletion of LSL resulting in EYFP fluorescence in bladder and prostate epithelial cells. Intravesical 4-OHT causes sporadic EYFP fluorescence cells in the bladder and no observable EYFP-positive cells in the prostate.

Reviewer 1, Figure 2D: Bladder specific deletion of *Kmt2c/Kmt2d/Pten* using intravesical delivery of 4OHT, Adeno-CMV-Cre, and Adeno-K5-Cre induced muscle invasive UC (also shown in Extended Data Fig. 9). BaseScope shows biallelic loss of both *Kmt2c* and *Kmt2d* transcript and IHC shows loss of H3K4me1 in tumor cells with retention in stromal cells.

Reviewer 1, Figure 2E: H&E of prostate with intravesical 4OHT injection shows no evidence of prostate tumorigenesis and gross images with intravesical adenovirus injection shows nodular bladder lesions (Prostate circled in red).

3. The authors do not provide direct evidence of loss of protein expression in the experiments, not even in the critical ones. This is a severe limitation of the work.

Response:

Loss of Kmt2c and Kmt2d proteins: Thank you for making this important point. The inability to detect Kmt2c and Kmt2d protein loss in tissues has been a longstanding limitation in both clinical specimens and in models. We have attempted Kmt2c and Kmt2d IHC with multiple antibodies without success. We have several types of evidence on the loss of Kmt2c and Kmt2d in the mouse urothelium and in organoids.

1) One indirect evidence we have shown is the urothelial-specific loss of H3K4me1 IHC, with retention in the stromal cells (**Reviewer 1, Figure 3A**, shown in Fig. 1g and Supplementary Fig. 9c).

2) We optimized in situ hybridization (BaseScope™) with probes targeting floxed exons (*Kmt2c* Exon3, *Kmt2d* Exon50-51) in mouse ureter and bladder tissues (**Reviewer 1, Figure 3B**, shown in reproduced Extended Data Fig. 1c,d and Supplementary Fig. 9a-b). Six months post tamoxifen administration, we observed efficient deletions of *Kmt2c/d* in both bladder and ureter urothelial cells. IHC of H3K4me1 and BaseScope of *Kmt2c/d* were also performed on other GEMMs (shown in Extended Data Fig. 9a and Supplementary Fig. 10d). The interspersed cells with retained BaseScope in tumors are from infiltrating immune cells (**Reviewer 1, Figure 3C**).

3) In organoids, Cut&Run of H3K36me3 that marks gene bodies showed loss of reads over the floxed exons. (Reviewer 1, Figure 3D, panel a, shown in Extended Data Fig. 4a).

4) In organoids, due to the lack of suitable anti-Kmt2c antibody, we only performed Western blot of Kmt2d (Reviewer 1 Figure 3D, panel b, shown in Extended Data Fig. 4b). Additionally, protein levels of other Kmt2 components (Kmt2a, Kmt2b, Menin, Set1a, Set1b, and Cxxc1) were provided to better support the mechanistic conclusion.

5) In urothelial organoids, we performed genomic PCR with primers amplifying the Floxed and excised alleles (Reviewer 1 Figure 3D, panel c, shown in reproduced Extended Data Fig. 4c).

Loss of Pten protein: We performed Pten IHC in Supplementary Fig. 9c, and Pten WB in Extended Data Fig. 10b,f.

Reviewer 1, Figure 3A: H3K4me1 IHC showing loss of the mark in urothelial cells, but retention in stromal cells. (Shown in Fig. 1g and Supplementary Fig. 9c)

Reviewer 1, Figure 3B: BaseScope for *Kmt2c* and *Kmt2d* showing deletion of the floxed alleles in urothelial cells. (Shown in Extended Data Fig. 1c,d and Supplementary Fig. 9a,b)

Reviewer 1, Figure 3C: IHC of CD45 showing immune cell infiltration in mouse UC models.

Reviewer 1, Figure 3D: Validation of *Kmt2c* and *Kmt2d* knockout in cultured urothelial cells. (Shown in reproduced Extended Data Fig. 4a-c)

4. The role of Menin is an important finding proposed in this work; however, there is a dearth of mechanistic insight about how this takes place. This should be addressed (or at least discussed) by the authors.

Response:

In our initial submission, we performed Menin Cut&Run. After loss of *Kmt2c/d*, Menin redistributed from *Kmt2d* positive enhancers to high-CpG promoters. This correlates with decreased transcription from enhancers and increased transcription from high-CpG promoters. Two mechanistic possibilities include change of Menin binding to *Kmt2a/b* or redistribution of *Kmt2a/b* complexes. To distinguish these two, we performed NEW Cut&Run of *Kmt2a*. We observed good overlap between components within the same *Kmt2* complexes (*Kmt2a* and Menin, *Set1a* and *Cxxc1*). *Set1a/Cxxc1* was most promoter enriched, *Kmt2d* was most enhancer enriched, and *Kmt2a/Menin* bound to a large number of both promoter and enhancer sites (**Reviewer 1, Figure 4A**, shown in Extended Data Fig. 4e-g).

Kmt2c/d knockout decreased depositions of *Kmt2* components (*Kmt2a*, Menin, *Set1a*, *Cxxc1*) on active enhancers (**Reviewer 1, Figure 4B**, shown in Fig. 4f-h). We further performed Western Blot of *Kmt2a*, *Kmt2b*, Menin, *Set1a*, *Set1b*, and *Cxxc1* in *WT* and *dKO* urothelial cells, and we did not see significant change on protein levels (**Reviewer 1, Figure 3D** from the previous question, shown in Extended Data Fig. 4b), suggesting that decreased deposition of *Kmt2* components was not caused by decreased protein levels but by loss of *Kmt2c/d*. In contrast, we found increased depositions of *Kmt2a* and Menin at CpG-high TSS (**Reviewer 1, Figure 4C, D**, shown in Fig. 5g,h and Supplementary Fig. 7b,c). These data suggest that *Kmt2c/d* loss redistributes *Kmt2a/Menin* complexes from active enhancer to CpG-high TSS and positively regulates transcriptional activities of these promoters.

Reviewer 1, Figure 4A: Genomic distribution of Kmt2a, Menin, Set1a, and Cxxc1. (Shown in reproduced Extended Data Fig. 4e-g)

Reviewer 1, Figure 4B: *Kmt2c/d* loss decreased *Kmt2a*/*Menin*/*Set1a*/*Cxxc1* depositions at active enhancers. (Shown in reproduced Fig. 4f-h)

Reviewer 1, Figure 4C: Kmt2a/Menin were redistributed to CpG-high TSS and upregulate transcriptional activity. (Shown in reproduced Fig. 5g-h)

Reviewer 1, Figure 4D: CpG content is a determinant for Kmt2 components binding at active TSS. (Shown in reproduced Supplementary Fig. 7b,c)

5. The Results are described in detail, but this section should be distilled to provide more specific messages to the reader. Need be, they can add some explanations in the Supplementary Data.

Response:

We have reorganized the figures and rewrote the manuscript to improve the conciseness.

6. The authors fail to place their findings in the context of bladder cancer in humans. Inactivation of KMT2C and KMT2D occurs frequently in non muscle-invasive bladder cancer in patients, which means that it is associated with tumors showing urothelial - rather than basal - differentiation. Therefore, the authors should discuss how their model contributes to an understanding of the role of these mutations in human cancer. In addition, they should discuss how KMD6A/UTX and/or PTEN mutations cooperate with KMT2C/KMT2D mutations in humans. The relevance of the mouse findings to human cancer is insufficiently developed.

Response:

Thank you, this insightful comment. Most human bladder cancers are urothelial not otherwise specified (UCNOS) or urothelial with variant differentiation. The molecular subtypes defined by TCGA (Luminal, Luminal-infiltrated, Luminal-papillary, Basal-squamous, Neuronal) were not necessarily pathologic subtypes and are defined by relative expression. For example, most TCGA tumors express some level of basal marker KRT14, luminal/umbrella markers KRT18 and UPK2, and many express the squamous marker DSG3 (**Reviewer 1, Figure 5A**). Our mouse model shows relative increase in basal markers and relative decrease in luminal/umbrella markers when *Kmt2c* and *Kmt2d* are lost (Fig. 2c-f). The tumor generated in our mouse models are UC NOS or UC with squamous differentiation (Fig. 6, Extended Data Fig. 8-9, Supplementary Fig. 10). The diffuse positivity of Krt5 is typical feature of UC NOS though the presence of Keratin pearls indicates UC with squamous differentiation.

In our study, *Kmt2c/d* loss sensitized tumorigenic susceptibility to *Pten* loss, *Trp53* loss, *PIK3CA E545K*, *KRAS G12V*, and carcinogen BBN treatment. These data suggest that *KMT2C/D* loss may induce broad tumorigenic sensitization to a wide range of secondary oncogenic drivers. As suggested, we further discussed the relevance of *KMT2C/D* mutation with other frequent mutations.

We added a discussion paragraph to compare the molecular signature, frequency and cooccurrence of *KMT2C/D* mutations with other frequent mutations. We also investigated the mutational frequencies of *KMT2C* and *KMT2D* among different stages of human bladder cancer, but we could not identify the obvious difference among stages Ta, T1, and T2-T4 (**Reviewer 1, Fig 5B-D**, PMID: 37884563, 28583311, and 36543146).

Reviewer 1, Figure 5A: Expression of DSG3, KRT14, KRT18, and UPK2 in bladder cancer (red) and normal bladder (grey) from TCGA. Data extracted using GEPIA2.

Reviewer 1, Figure 5B: (Fig. 1C from PMID: 36543146)

Reviewer 1, Figure 5C: (Fig. 1 from PMID: 28583311)

Reviewer 1, Figure 5D: (Table 2 from PMID: 37884563)

Gene	Chromosome	Frequency (%)			Alteration	Functions affected
		Ta	T1	T2+		
CDKN2A	9p21	30	60	60	Loss of heterozygosity, deletion	Cell cycle
		≤2	12	22	Homozygous deletion	
		1	7	7	Mutation	
RB1	13q14	0	14	17	Inactivating mutation	Cell cycle
ATM	11q22	12	16	14	Inactivating mutation	Cell cycle
CDKN1A	6p21	11	11	9	Inactivating mutation	Cell cycle
TP53	17p13	4	24	48	Inactivating mutation	Transcription
ELF3	1q32	8	22	12	Inactivating mutation	Transcription
ZFP36L1	14q24	12	11	6	Inactivating mutation	Transcription
KDM6A	Xp11	40	40	26	Inactivating mutation	Chromatin regulation
KMT2D	12q13	35	27	28	Inactivating mutation	Chromatin regulation
CREBBP	16p13	23	20	12	Inactivating mutation	Chromatin regulation
KMT2C	7q36	23	14	18	Inactivating mutation	Chromatin regulation
STAG2	Xq25	30	9	14	Inactivating mutation	Chromatin regulation
ARID1A	1p36	11	27	25	Inactivating mutation	Chromatin regulation
KMT2A	11q23	11	15	11	Inactivating mutation	Chromatin regulation
EP300	22q13	15	11	15	Inactivating mutation	Chromatin regulation
ASH1L	1q22	10	12	7	Inactivating mutation	Chromatin regulation
ARID2	12q12	7	11	8	Inactivating mutation	Chromatin regulation
ERCC2	19q13	4	24	18	Inactivating mutation	DNA repair
BRCA2	13q13	10	10	9	Inactivating mutation	DNA repair
PTEN	10q23	7-12	20-30	50	Loss of heterozygosity, deletion, mutation	Regulator of AKT signalling
TSC1	9q34	12	15	8	Inactivating mutation	Regulator of mTOR signalling
RBM10	Xp11	7	13	5	Inactivating mutation	RNA splicing

Genes affected in >10% of at least one bladder cancer stage are shown. Large genes not formally identified as significantly mutated or with unknown function are not listed.

Specific comments

1. Page 3, lines 109-125: show protein expression in control urothelium as well as upon TMX-induced recombination.

Response:

Due to the lack of antibodies that work on IF or IHC, we performed BaseScope with probes targeting floxed exons of *Kmt2c* and *Kmt2d* in mouse ureter and bladder tissues (**Reviewer 1, Figure 3B**, shown in Extended Data Fig. 1c,d). Six months post tamoxifen administration, we observed efficient deletions of *Kmt2c/d* and overall decrease of H3K4me1 in urothelial cells (**Reviewer 1, Figure 3A, B**).

2. Supplementary Fig. 2: cluster 7 is enriched in transcripts coding for ribosomal proteins- artifact? is this related to quality of these cells?

Response:

Thank you for this keen observation. To validate the ribosomal protein expression in urothelial cells, we performed immunofluorescent staining of Rpl36 and Rps21 in *WT* and *dKO* bladder tissue that collected 6 months post tamoxifen administration. Compared to *WT*, increased staining intensity was observed in *dKO* group (**Reviewer 1, Figure 6**). To directly compare ribosomal biogenesis, we performed puromycin incorporation assay (200 μ L per mouse, 2.5mM, 1h) in *WT* and *dKO* mice 3 weeks post tamoxifen administration. We next performed immunohistochemistry to detect incorporated puromycin and found increased puromycin staining intensity in *dKO* cells (n=3 mice), suggesting the increased ribosomal biogenesis (**Reviewer 1, Figure 6**). Notably, increased ribosome protein expression is also a signature of the pre-tumorigenic transcriptome (PMID: 29057876), consistent with primed tumorigenic state after *Kmt2c/d* KO. The increased ribosomal biogenesis in *dKO* condition is the focus of ongoing research in the lab and too preliminary to be included in this current study.

Reviewer 1, Figure 6: Rpl36 and Rps21 immunofluorescence and puromycin labeling of *WT* and *dKO* urothelium.

3. Page 5, line 166: I'm surprised that, in general, there are no big expression differences amongst the clusters for the 5 basal marker genes in the *WT* populations. Are there significant differences?

Response:

We too were surprised by the relative high levels of *Krt5* and *Trp63* into the luminal layers. We generated a Z-score based heatmap and color-scaled UMAP of the expression of these genes. (**Reviewer 1, Figure 7A, B**, shown in Supplementary Fig.

1d). A gradient of *Krt5*, *Trp63*, and *Col17a1* among urothelial cell clusters can be appreciated. Unlike for example the prostate where luminal and basal cells form distinct clusters, the urothelium clusters form a gradient. We next queried two previous scRNA-seq studies of mouse bladder. Consistent to our study, they found a continuum of *Krt5* and *Krt8* expression that forms a gradient from basal to luminal cells (**Reviewer 1, Figure 7C**, from PMID: 31462402).

Reviewer 1, Figure 7A: Z-score heatmap of basal, luminal, and EMT markers.

Reviewer 1, Figure 7B: Expression UMAP of basal, luminal, and EMT markers.

Reviewer 1, Figure 7C: Two other single cell RNA-seq studies of normal mouse bladder.

4. Page 5, line 174 figure 2F: please, provide quantification of Krt14+ cells in dKO vs WT needed

Response:

As suggested, we have added statistics of Krt14 positivity in *WT* and *dKO* mice. (Reviewer 1, Figure 8, shown in Fig. 2f,g)

Reviewer 1, Figure 8: Immunofluorescence of Krt5, Krt14, Upk2, and Krt20 with quantification of Krt14 positive cells.

5. Page 5, line 175 figure 2G: authors should clarify in the figure legend or method section the number of n's in each group of the comparisons. Furthermore, authors should test expression of the human orthologues of the genes in figure 2c-e for the comparisons in figure 2G.

Response:

The p values of this test have been provided as raw statistical data (below). We also used the basal and luminal markers from Fig. 2d and 2e for the comparison, but we did not see consistent changes of these genes (**Reviewer 1, Figure 9**). Probably because they were not generated by statistical comparison among human UC subtypes, and they may not be the best differentiation markers for human bladder cancer samples.

Group	Population	pvalue	mean_Z_diff	p.adjust	-log10P
KMT2C_vs._WT	Basal	0.340675629	0.12850927	0.94779368	0.467659
KMT2C_vs._WT	Luminal	0.723654075	0.018987256	0.94779368	0.140469
KMT2C_vs._WT	Squamous_diff	0.840992902	0.050275508	0.94779368	0.075208
KMT2D_vs._WT	Basal	0.083683655	0.199311247	0.167367309	1.077359
KMT2D_vs._WT	Luminal	0.361969736	-0.106137448	0.482626315	0.441328
KMT2D_vs._WT	Squamous_diff	0.041760319	0.218932076	0.167041275	1.379236
KMT2C/D_vs._WT	Basal	0.143203913	0.15686011	0.534266426	0.844045
KMT2C/D_vs._WT	Luminal	0.848944187	-0.031982952	0.848944187	0.071121
KMT2C/D_vs._WT	Squamous_diff	0.267133213	0.108835032	0.534266426	0.573272

Reviewer 1, Figure 9.

6. Page 5, line 179: How was Mki67 positivity defined based on expression? Which expression threshold was used for saying that the cell is positive?

Response:

Given the low coverage of scRNA-seq, we defined Mki67 positivity as detectable reads higher than 1.

7. Page 6, lines 195-210: the analysis of differentiation is very superficial, I suggest to be more cautious regarding these statements - just remove the term "umbrella cells". Same applies to line 221 of the text.

Response:

We agree with the reviewer, as cluster 4 in WT is not a pure umbrella cell cluster. We have changed the name of umbrella cluster/marker to luminal cluster/marker in the manuscript.

8. Page 6, line 205 figure S3D: Authors should also evaluate Krt20 expression as this is uniquely expressed in umbrella cells and not in intermediate cells.

Response:

Real-time PCR of *Krt20* has been provided in Extended Data Fig. 3e (Reviewer 1, Figure 10). We also added *Krt20* immunofluorescence (Reviewer 1, Figure 8).

Reviewer 1 Figure 10.

9. Page 6, line 225 figure 3A: Authors should also include the molecular subtype gene sets from Figure 2G or consensus subtype gene sets from Kamoun et al., 2020 in this analysis. Where do the luminal, basal, and squamous gene sets fall in this graph?

Response:

Basal markers and luminal markers from Fig. 2g (new Fig. 2h) are significantly positively and negatively enriched and have been added as gene sets in Fig. 3a (Reviewer 1, Figure 11A). But for squamous markers, we did not see significant enrichment (NES=-

1.23, FDR $q=0.669$). Next, we used the consensus subtype gene sets from Kamoun et al., we found slight increase of basal/squamous as well as luminal/papillary signatures for all three comparisons, but there were no significant differences (**Reviewer 1, Figure 11B**). These observations raised interesting questions about the function of *KMT2C/D* loss in luminal subtype of urothelial cancer. We have added this as an open question to the discussion.

Reviewer 1, Figure 11A

Reviewer 1, Figure 11B

Gene sets used in Fig. 2h:

Luminal markers: CYP2J2, ERBB2, ERBB3, FGFR3, FOXA1, GATA3, GPX2, KRT18, KRT19, KRT20, KRT7, KRT8, PPARG, XBP1, UPK1A, UPK2

Basal markers: CD44, CDH3, KRT1, KRT14, KRT16, KRT5, KRT6A, KRT6B, KRT6C

Squamous markers: DSC1, DSC2, DSC3, DSG1, DSG2, DSG3, S100A7, S100A8

10. Page 7, line 233: Why are DKO cells more responsive to EGF? Which components of the EGFR signaling pathway is upregulated in DKOs? Are EGFRs upregulated?

Response:

The cellular response to EGF is typically analyzed by observing phosphorylation of MAPKs and subsequent transcriptional activation of immediate-early genes (IEGs, PMID: 17322878). In scRNA-seq studies, gene sets of IEGs and EGF-stimulated genes were positively enriched with *Kmt2c/d* knockout (**Reviewer 1, Figure 12A**, shown in Fig.

3a,b). Increased transcription of IEGs to EGF stimulation were validated in *dKO* cells (Reviewer 1, Figure 12A, shown in Fig. 3c,d), suggesting more robust transcriptional response to EGF stimulation. We further demonstrated that IEGs such as *Egr1*, *Egr3*, *Nr4a1*, *Nr4a2*, and *Nr4a3* are characterized by higher CpG content at TSS and by elevated *Kmt2a* binding after *Kmt2c/d* knockout (Reviewer 1, Figure 12A, shown in Supplementary Fig. 7d). In *WT* and *dKO* urothelial cells, we did not see big difference of ERBB family mRNA levels (Reviewer 1, Figure 12B from triplicate RNA-seq, *ErbB4* is not expressed). To test whether *Kmt2c/d* loss may lead to dependence on EGF signaling, we treated *WT* and *dKO* urothelial cells with EGFR inhibitors.

Reviewer 1, Figure 12A

Reviewer 1, Figure 12B

11. Page 7, line 234 figure 3C,D: Is this non-organoid 2D cell experiments? How were they performed (freshly sorted urothelial cells, based of which markers, or cell lines

established from the GEMMs)? the same applies to the experiments in page 7, line 240, Figure 3F.

Response:

These experiments were performed on mouse urothelial organoid lines cultured on collagen-coated dishes (1% in cold sterile H₂O, 1h at RT). We used the same organoid culture medium for both 2D and 3D-matrigel culture. The updated procedures have been added to the methods. We apologize for missing these essential information in the initial submission.

12. Page 7, line 245: Is loss of urothelial differentiation and basalization also observed in DKOs in the pooled scRNA analysis ?

Response:

Yes, as shown in Supplementary Fig. 2a, there are decreased differentiation markers (e.g., *Upk1a*, *Upk1b*, *Upk2*, *Upk3a*, *Upk3b*, *Krt8*, *Krt18*) and increased basal markers (e.g., *Bcam*, *Krt15*, *Col17a1*, *Krt5*, *Itgb1*) in the pooled scRNA-seq data of *dKO* mice.

13. Page 7, line 258-260: are the findings in NAT samples due to KMT2C/D mutations or are they a general feature of NAT? The NAT samples in Aran et al., 2017 are from TCGA and the healthy are from Gtex and both should have information on somatic mutations for KMT2C/D. The authors should test if there is an enrichment KMT2C/D mutations in the NAT vs Healthy samples.

Response:

This is an important question that we have pondered. However, the TCGA normal is not suitable for mutational analysis. Mutational analysis of normal urothelium involves urothelial microdissection of tiny (1mm² or smaller) areas and the clone size is small. The sequencing is too shallow in TCGA to detect mutations in non-microdissected large tissues with ample non-urothelial cells. The robust enrichment of *KMT2C/D* loss in tumor adjacent tissues is striking. But based on available data, we can't exclude the possibilities that non-mutational mechanisms, such as paracrine tumor effects, can affect tumor adjacent normal transcriptomes.

14. Line 279-285: testing the chromatin marks is the appropriate way to confirm deletions. As indicated above, the authors should test for protein expression with specific antibodies.

Response:

As suggested, WB and genotyping results have been provided in the reproduced Extended Data Fig. 4a-c (**Reviewer 1, Figure 3D**).

15. Page 8, line 311: Figure S6d shows, Cxxc1 and Set1a signals in Kmt2d bound enhancers. Is there high concordance of Cxxc1 and Set1a peak intensity in all promoter-distal sites or just in Kmt2d bound enhancers? The text should be modified accordingly.

Response:

We have now corrected the description and updated the correlations of Kmt2a/Menin, and Set1a/Cxxc1 on distal and proximal peaks (**Reviewer 1, Figure 4A**, shown in reproduced Extended Data Fig. 4e-g).

16. Page 9, line 325-326: yet all except for Menin decrease significantly at Kmt2d negative enhancers. Authors should explain how this can be.

Response:

In the new data, we found consistent alterations of Kmt2a and Menin on both Kmt2d pos or neg enhancers (**Reviewer 1, Figure 4B**, shown in Fig. 4f).

17. Page 9, line 328-329: why was this type of normalization done? Has this been published before? Authors should explain.

Response:

The normalization of Cut&Run of H3K4me1 is challenging for several reasons. First, the percent of reads that are on peak vs. background is much higher with CUT&RUN than ChIP-seq and most of the reads for H3K4me1 are on peak. Second, H3K4me1 enrichment is significantly higher at Kmt2d+ enhancer compared to active TSS (**Reviewer 1, Figure 13**, data from *WT*). *Kmt2c/d* knockout induced significant decrease of H3K4me1 at active enhancers. Normalization of H3K4me1 to 1x genome coverage by assuming *WT* and *dKO* libraries at the same complexity will artificially increase the residual H3K4me1 signal in *dKO* scenario. For H3K4me2 and H3K4me3, as the signal are stronger at TSS, we can use the standard methods for normalization. Notably, by doing this additional normalization, we observed consistent alterations of H3K4me1 with H3K4me2 and H3K27ac.

Reviewer 1, Figure 13: Total read counts of H3K4me1, H3K4me2, and H3K4me3 at active TSS and Kmt2d+ enhancers.

18. Page 9, line 338 figure 4H: Figure 4H and S6H represent two single experiments but for the same ChIPs (except for H3K27me3 which only was made once). When experimental replicates are performed, the data (reads, etc. should be evaluated together not separately).

Response:

To address this question, we first generated NEW critical epigenetic sequencing experiments in duplicate. We defined peaks as those called by MACS3 in both replicates with FDR<0.01 to ensure reproducibility. Differential expression, signal enrichment, BigWig files, and heatmap were generated as average of all replicates. All sequencing data have been updated in the new manuscript.

19. Page 9, line 349: ChIP-qPCR for K4me1 & K27ac at enhancer and promoter regions for Fgfr3 and Upk3bl as well as RT-qPCR for these genes should be performed in WT and DKO to confirm the seq data.

Response:

ChIP-qPCR validations of H3K4me1 and H3K27ac have been provided in **Reviewer 1, Figure 14, Supplementary Fig. 6c.**

Reviewer 1, Figure 14

20. Page 10, line 387: authors should also test how urothelial differentiation genes behave in GSEA.

Response:

As indicated, we have performed GSEA analysis of luminal differentiation markers with Top500 downregulated TSS in PRO-cap. We saw the enrichment of *Pparg* and *Upk2* among these downregulated genes (Reviewer 1, Figure 15), consistent with their downregulated mRNA levels (Reviewer 1, Figure 15, from triplicate RNA-seq). However, due to the small number of overlapped genes, we did not put these data in manuscript.

Reviewer 1, Figure 15

21. Page 11, line 411 figure 5H: on what basis was the threshold set for defining bivalency and non-bivalency? In figure legend it says "Here, we defined TSS with H3K4me3 \geq 4 and H3K27me3 $<$ 4.5 as H3K4me3 only, while TSS with H3K4me3 \geq 4 and H3K27me3 \geq 4.5 as 1434 bivalent". Authors should explain why they didn't use the called peaks to define these states.

Response:

The initial description of bivalency used ChIP DNA hybridized to promoter microarray (ChIP-chip) and intensity cutoff (PMID: 16630819). Several subsequent studies have also used read count cutoffs at promoters of ChIP-seq data (**Reviewer 1, Figure 16A-B**). This method allows us to count reads at promoters and correlate with ATAC, PRO-cap, etc. We presented an orthogonal method using hidden Markov model (ChromHMM) showing TSS with bivalent chromatin state 4 is enriched for upregulated genes and TSS with state 12 that's more typical of enhancers is enriched for downregulated genes (**Reviewer 1, Figure 16C**, shown in Fig. 5d).

We have now performed another method to identify bivalent chromatin used by many groups. We called H3K27me3 peaks using SICER, and H3K4me3 and H3K4me1 peaks using MACS3. Promoters that overlap with H3K4me3, H3K27me3 and both are identified. In addition, among promoters with neither H3K4me3 nor H3K27me3, a small group had H3K4me1. We then calculated their baseline expression, expression change, and relative abundance in top 500 upregulated and downregulated genes (**Reviewer 1, Figure 16D**, shown in Extended Data Fig. 5a,b). The data is consistent with downregulation of Kmt2d-only promoters and upregulation of bivalent promoters after *Kmt2c/d* loss.

Reviewer 1, Figure 16A: (Figure 2 from PMID: 28796844)

Reviewer 1, Figure 16B: (Figure 4b from PMID: 32123383)

Reviewer 1, Figure 16C: (shown in Figs. 4b and 5d)

Reviewer 1, Figure 16D: (Shown in Extended Data Fig. 5a,b)

22. Page 11, line 443 figure S11 A-E: ChIP-qPCR for Menin, K4me1 & K27ac at promoter regions for *Krt16*, *Zeb2*, *Nr4a1*, *B2m*, and *H2-K1* should be performed as well as RT-qPCR for these genes in WT and DKO to confirm the seq data.

Response:

ChIP-qPCR validations of H3K4me3, H3K27ac, and *Kmt2a* have been provided in Supplementary Fig. 8f (Reviewer 1, Figure 17).

Reviewer 1, Figure 17

23. Page 12, line 472-486 figure S13a-d: non-tamoxifen treatment conditions should also be included in the IHCs.

Response:

As suggested by the reviewer, we have added a panel of EYFP mice analyzed at the same timepoint (**Reviewer 1, Figure 18**, shown in Extended Data Fig. 8d and Supplementary Fig. 9c).

Reviewer 1, Figure 18

24. Fig. 6b,c: regarding the differences between males and females, it is not clear whether this also reflects extra-urothelial effects. In humans, females have more advanced tumors and worse outcome. This is important considering the fact that the Cre driver is not urothelium-specific and is active in the prostate.

Response:

True. In GEM models, especially in *Kmt2c^{ff};Pten^{ff}* and *Kmt2d^{ff};Pten^{ff}* mice, the morbid phenotypes were exclusively caused by distended bladder and/or swollen kidney (**Reviewer 1, Figure 20A** for next question #25, shown in Supplementary Table 2). Male mice in these two groups died with kidney failure, as we found extremely high BUN and CREA in their blood (**Reviewer 1, Figure 20B** for next question #25, shown in Supplementary Table 3, n=4 males, n=2 females). But in the female mice, only slight increase of BUN was observed. In GEMM models, we feel the differential survival between males and females might be probably caused by the different physiological structures in urethra, especially by the development of UC in prostatic urethra. Notably, we did see higher Ki67 positive proliferating cells in male compared to female mice (**Reviewer 1, Figure 19A**, middle, shown in Fig. 6e), consistent with the more malignant UC subtypes in male mice (shown in Fig. 6f and Extended Data Fig. 8b).

Due to our interest in studying prostate cancer, in our GEMM models, we did collect prostate tissue for most male mice. The prostate exhibited similar *Pten*-loss mediated prostatic intraepithelial neoplasia (PIN) as we previously described, and *Kmt2c/d* loss did not accelerate prostate tumorigenesis (**Reviewer 1, Figure 19B**). We concluded that the prostate was not the cause of mortality.

In BBN models, we did not see apparent differences of tumorigenesis or hydronephrosis between male and female mice (**Reviewer 1, Figure 19A**, right, shown in Fig. 6g,h). Consistently, in models of intravesical delivery of adeno-Cre virus, we observed comparable frequency of tumorigenesis in 4/6 male and 8/10 female mice (**Reviewer 1, Figure 19A**, right, shown in Extended Data Fig. 9b).

In summary, we are unable to make the clear conclusion of sex dimorphism in our UC models. But carefully designed grafting assay might be help in answering this question.

Reviewer 1, Figure 19A

Reviewer 1, Figure 19B

25. Lines 477-480: describe better phenotype of mice in ureter and bladder. Fig. 6d-f. Cause of bladder distension? tumor-related?

Response:

We have extensively reanalyzed all the pathologic material with our pathologist Hikmat Al-Ahmadie to grade each ureter, bladder, and urethra and provide better representative images and quantification (Fig 6, Extended Data Fig. 8). To better describe the phenotypes in our GEMM models, we provided the health check reports from veterinarian of our vivarium facility. We also provided the morphology of bladder, kidney, and ureter tissues upon dissection (**Reviewer 1, Figure 20A**, shown in Supplementary Table 2). Most moribund mice in *Kmt2c^{fl/fl};Pten^{fl/fl}* or *Kmt2d^{fl/fl};Pten^{fl/fl}* mice developed bladder distension, hydronephrosis and ureter enlargement. The swollen abdomen was exclusively caused by distended bladder and/or distended kidney. Blood test of renal panel from *Kmt2c^{fl/fl};Pten^{fl/fl}* mice were also provided as Supplementary Table 3. We observed profound increase of phosphorus, BUN, and CREA in male mice (n=4 mice), suggesting the development of renal failure (**Reviewer 1, Figure 20B**, shown in Supplementary Table 3). In contrast, in the female mice (n=2 mice), we observed only slight upregulation of BUN, despite the development of hydronephrosis. We have added these details to the manuscript.

As described in the manuscript, mice in *Tmprss2-CreERT²;Kmt2c^{fl/fl};Kmt2d^{fl/fl};Pten^{fl/fl}* group developed malignant gastric cancer (**Reviewer 1, Figure 20A**, shown in Supplementary Table 2). But we did see early tumor formation and increased Ki67+ cells in both ureter and bladder at 6-week timepoint post tamoxifen administration (Shown in Fig.6c-e). We further solved this by intravesical delivery of 4OHT and Cre-expressing adenovirus and observed muscle invasive urothelial cancer in 3 months (Reviewer 1, Figure 2D, shown in Extended Data Fig. 9).

Reviewer 1, Figure 20A: Moribund phenotypes and morphology of bladder, ureter and kidney in GEMMs. (Supplementary Table 2)

	Mouse #	Months post tamoxifen administration	Gender	Phenotypes
Tmprsse-CreER^{T2};Kmt2c^{ff};Pten^{ff}	#1	7	Male	Hunched and lethargic with bladder distension and enlarged ureter
	#2	7.6	Male	Lethargic and unbalanced with bladder distension
	#3	4.4	Male	Lethargic with bladder distension
	#4	4.2	Male	Extremely lethargic with bladder distension
	#5	3	Male	Moribund with bladder distension
	#6	4.2	Male	Hunched and sluggish with bladder distension
	#7	3.5	Male	Moribund with bladder distension
	#8	4.7	Male	Extremely lethargic with bladder distension
	#9	7.2	Male	Extremely lethargic with bladder distension
	#10	6.5	Male	Hunched and sluggish with bladder distension
	#11	11	Female	Sluggish with hydronephrosis
	#12	6.7	Female	Hunched and sluggish with hydronephrosis and enlarged ureter
	#13	5.4	Female	Hunched and sluggish with hydronephrosis
	#14	7.6	Female	Distended abdomen with hydronephrosis
Tmprsse-CreER^{T2};Kmt2d^{ff};Pten^{ff}	#1	5.5	Male	Hunched and sluggish with bladder distension
	#2	6.3	Male	Distended abdomen with bladder distension, hydronephrosis and enlarged ureter
	#3	6.5	Male	Moribund with bladder distension, hydronephrosis and enlarged ureter
	#4	7	Male	Distended abdomen with hydronephrosis
	#5	6	Male	Hunched and sluggish with hydronephrosis and enlarged ureter
	#6	9.1	Male	Distended abdomen with hydronephrosis and enlarged ureter
Tmprsse-CreER^{T2};Kmt2c^{ff};Kmt2d^{ff};Pten^{ff}	#1	7 weeks	Male	Moribund with gastric cancer
	#2	7 weeks	Male	Moribund with gastric cancer
	#3	4.2 weeks	Male	Moribund with gastric cancer
	#4	4.2 weeks	Male	Moribund with gastric cancer
	#5	4 weeks	Male	Moribund with gastric cancer
	#6	6 weeks	Male	Moribund with gastric cancer
	#7	6.4 weeks	Male	Moribund with gastric cancer
	#8	6.2 weeks	Male	Moribund with gastric cancer and enlarged ureter
	#9	9 weeks	Male	Moribund with gastric cancer
	#10	12 weeks	Male	Moribund with gastric cancer and enlarged ureter
	#11	4 weeks	Female	Moribund with gastric cancer
	#12	7 weeks	Female	Moribund with gastric cancer and enlarged ureter
	#13	2 weeks	Female	Moribund with gastric cancer
	#14	3.5 weeks	Female	Moribund with gastric cancer
	#15	3.5 weeks	Female	Moribund with gastric cancer
	#16	4.2 weeks	Female	Moribund with gastric cancer
	#17	4.2 weeks	Female	Moribund with gastric cancer
	#18	7 weeks	Female	Moribund with gastric cancer

Reviewer 1, Figure 20B: Blood test of renal panel parameters in *Kmt2c^{ff};Pten^{ff}* mice. (Supplementary Table 3)

Tmprss2-CreER^{T2};Kmt2c^{ff};Pten^{ff} , #4	Test ID: 1900045901	Result values	Reference range	
Male, 4.4 months post tamoxifen Lethargic with bladder distension	[BUN (mg/dL)]	535	5.0-28	
	[CREA (mg/dL)]	10.33	0.2-0.5	
	BUN/CREA Ratio	49.4	--	
	[TP (g/dL)]	2.7	4.8-7.2	
	[ALB (g/dL)]	1.2	2.4-4.3	
	GLOB (g/dL)	1.5	1.7-2.2	
	A/G Ratio	0.8	--	
	[P (mg/dL)]	76.7	7.3-14.5	
	[Ca (mg/dL)]	8.8	9.5-12.5	
	[TCO2 (mEq/L)]	0	--	
	[Na (mEq/L)]	113	145-181	
	[K (mEq/L)]	21.6	7.3-11.1	
	[CL (mEq/L)]	72	111-134	
	[Na/K]	5	--	
	[Anion Gap]	63	--	
	Tmprss2-CreER^{T2};Kmt2c^{ff};Pten^{ff} , #5	Test ID: 1900045901	Result values	Reference range
Female, 11 months post tamoxifen Sluggish with hydronephrosis		[BUN (mg/dL)]	31	5.0-28
		[CREA (mg/dL)]	0.17	0.2-0.5
		BUN/CREA Ratio	182.4	--
		[TP (g/dL)]	6.2	4.8-7.2
		[ALB (g/dL)]	2.6	2.4-4.3
		GLOB (g/dL)	3.6	1.7-2.2
		A/G Ratio	0.7	--
		[P (mg/dL)]	6.8	7.3-14.5
		[Ca (mg/dL)]	9.9	9.5-12.5
		[TCO2 (mEq/L)]	29	--
		[Na (mEq/L)]	154	145-181
		[K (mEq/L)]	6.2	7.3-11.1
		[CL (mEq/L)]	109	111-134
		[Na/K]	25	--
		[Anion Gap]	22	--

Minor comments

- Page 3, line 72: reference 4 should be 5, instead.

Response:

Thanks for the careful reading, we have corrected this error.

- Page 6, line 190: Santos et al., 2019 does not describe Krt8 as an umbrella cell marker but an intermediate/supra-basal marker, as has been shown by other studies. This should be corrected here.

Response:

As suggested by the reviewer, we have redefined Krt8 as luminal markers.

- Page 8, lines 303-306: information on distribution should be reported in a more discriminant manner: promoters, intragenic, intergenic.

Response:

The genomic annotations of each individual Kmt2 components have now been provided in Extended Data Fig. 4e,f (**Reviewer 1, Figure 4A**).

- Page 8, line 303: should be rephrased as not all components of the complexes are being ChIP'ed (e.g. Kmt2c, Kdm6a, Kmt2a, Kmt2b, and Set1b are not being ChIPed).

Response:

As suggested, we have corrected this expression.

- Page 8, line 308: callout to Fig S6c should come later in line 310 as it shows prior data.

Response:

Since we have performed NEW Cut&Run of Kmt2a in our system, we removed this data.

- Page 9, line 356: again, should be rephrased since not all components of the complexes are being ChIPed (e.g., Kmt2c, Kdm6a, Kmt2a, Kmt2b, and Set1b are not being ChIPed).

Response:

As suggested by the reviewer, we have corrected this expression.

- Page 10, line 377: state 3 in Fig5d doesn't seem to have more down genes than state 1.

Response:

For each state, we compared their relative enrichment over genome with all TSS. Thus, states 2, 3 and 12 which marks TSS with H3K4me1 but not H3K27me3 are enriched in downregulated TSS. The length of the column does not correlate to the number of TSS among different state.

- Line 489: do the authors really mean "urethral urothelial cancer"? urethral cancers tend to be squamous and unrelated to bladder urothelial tumors.

Response:

Thank you. As in human, the proximal prostatic urethra is lined with urothelial cells while further distal urethra is squamous. We queried MSK-IMPACT for our urethral cancers and there was comparable *KMT2C* and *KMT2D* mutations in both urothelial (orange) and squamous histology (blue) (**Reviewer 1, Figure 21A**). In this study, we observed urethral urothelial cancer in the prostatic urethra of *Kmt2c^{fl/fl};Pten^{fl/fl}* and *Kmt2d^{fl/fl};Pten^{fl/fl}* mice, but we did not observe tumorigenesis in the squamous urethra. Positive IHC of Krt7 confirms the urothelial origin of these UC (**Reviewer 1, Figure 21B**, shown in Extended Data Fig. 8c).

Reviewer 1, Figure 21A: OncoPrint of urethral cancer in MSK-IMPACT cohort grouped by urothelial (orange), squamous (blue) and adenocarcinoma (red) histology.

Reviewer 1, Figure 21B: Pathology of prostatic urethral carcinoma in *Kmt2c^{fl/fl};Pten^{fl/fl}* and *Kmt2d^{fl/fl};Pten^{fl/fl}* mice 6 months after tamoxifen.

- Discuss better differences between the role of *Kmt2c* and *Kmt2d*, which appears obvious in Fig. S13.

Response:

Thank you for this suggestion and we agree. We have added a new discussion paragraph.

Reviewer #2:

Remarks to the Author:

The manuscript by Want et.al., seeks to determine how urothelial field cancers drive urothelial cancers and in a sense, “license” downstream oncogenic transformation. Overall, the manuscript is highly impactful as it provides a possible epigenetic mechanism for impairing differentiation in the urothelium, by KMT2C/D loss, and priming a basal cell state at risk for subsequent tumor initiation. The studies are considered largely original and the methodology for these studies is an innovative blend of single cell genomics, mouse modeling, and epigenetic analyses. The overall conclusion that KMT2c/d loss elicits a field effect that provides a “license” for transformation is however, not entirely substantiated by the current data. There are, several weaknesses, that if addressed experimentally would support this conclusion. These weaknesses are bulleted below (major and minor).

Major Concerns:

Modeling Field effects in the Urothelium – A major weakness is that it is unclear if there is strong field effect in the urothelium due to the high dose tamoxifen (2x IP) and long lineage tracing (6 months). Whether or not, the *Kmt2C/D* deleted urothelium clonally expands and competes alongside adjacent normal urothelium is unclear. Low dose tamoxifen followed by a robust chase /timecourse should be considered (i.e. low dose tamoxifen to infrequently label *Kmt2c/d* clones and then sacrifice 1 week, 2 weeks, 4 weeks, etc.). As it stands now, the *Kmt2c/d* experiments are fairly static and are a simple description of *Kmt2c/d* loss in the entire epithelium. In the best case scenario, the EYFP reporter allele could be included in the *Kmt2c/d* f/f experiments and compared to *Kmt2c/d* WT in order to compare and contrast field effects across the alleles.

Response:

Thank you for pointing out this critical question. While high-dose tamoxifen was highly efficient in activation of the *Rosa26-CAG-LSL-EYFP*, we observed lower efficiency in deletion of *Kmt2c/d* at early time points. This is one reason we chose the 6-month time point.

To address this field effect question, we have made a couple of efforts:

1. We initially attempted to follow EYFP positive cells after tamoxifen. We performed intravesical injection of 4-hydroxytamoxifen (4OHT, 2µg, 50µl per injection, DMEM + 1.25% Tween-80) in *Tmprss2-CreER^{T2};R26-EYFP* mice and *Tmprss2-CreER^{T2};Kmt2c^{ff};Kmt2d^{ff};R26-EYFP* mice. One week post 4OHT injection, we observed a highly variable percentage of EYFP positive urothelial cells averaging around 10-15% (**Reviewer 2, Figure 1A**). Yet, when we performed H3K4me1 IHC, we did not observe cells that robustly lost staining. There are 5 independent floxed alleles (1 *EYFP*, 2 *Kmt2c*, and 2 *Kmt2d*) and recombination can be stochastic and EYFP fluorescence may not be a good surrogate for *Kmt2c/Kmt2d* biallelic loss. To ask this, we treated *Tmprss2-CreER^{T2};Kmt2c^{ff};Kmt2d^{ff};R26-EYFP* urothelial cells with 100nM or 1000nM

4OHT for 4 hours. After medium changing, we cultured the cells for 48 hours before analyses. Then we analyzed EYFP positivity with flow cytometry and *Kmt2c/d* deletion efficiency with genomic DNA (Real-time PCR of floxed alleles). We found that deletions of *Kmt2c/d* are less efficient compared to Rosa26 locus, especially at the lower concentration (**Reviewer 2, Figure 1B**). These data indicate that we cannot use a strategy of following EYFP positivity.

2. Because of the large variability when lower dose of tamoxifen was used, we reasoned that following the same mice over time is a more robust way for monitoring changes. We thus used a Luciferase-2A-Cre expressing lentivirus. The constitutive and continuous Cre would efficiently recombine all alleles and we can follow mice by Luciferase. We performed intravesical injection of Luciferase-2A-Cre expressing lentivirus (1×10^8 IFU/ml in PBS) in *Control* and *Kmt2c/d* floxed mice (20 μ L per injection, 10 μ L virus + 10 μ L DMEM, with 10 μ g/ml polybrene at final concentration). Cre and luciferase expression was determined in vitro on Rosa26-EYFP reporter mouse urothelial cell. After virus injection, we performed live imaging by checking luminescent signal every month. But we were unable to see any signal 4 months post the injection, which might be caused by the low efficiency of lentivirus infection.
3. In attempt to quantify the efficiency of *Kmt2c* and *Kmt2d* loss in the mouse bladder, we have successfully developed BaseScope (**Reviewer 2, Figure 2A, B for major question #2**). Cells that are negative for *Kmt2c* or *Kmt2d* indicates loss of RNA and likely biallelic deletion. We still chose to use high-dose tamoxifen to follow mice because there is lower variation of deletion which allows for better statistical comparisons. We treated the mice with two doses of 3mg tamoxifen (IP) and collected bladder tissues from *Control* (*Tmprss2-CreER^{T2};R26-EYFP*, collected at 6-month) and *dKO* groups (*Tmprss2-CreER^{T2};Kmt2c^{ff};Kmt2d^{ff}* collected at 1-week, 6-week, and 6-month). We next performed BaseScope targeting *Kmt2c* and *Kmt2d* floxed exons. We observed progressive decrease of *Kmt2c* and *Kmt2d* signal over time (**Reviewer 2, Figure 1C**, shown in Extended Data Fig. 2c), suggesting the growth advantage of *Kmt2c/d* loss urothelial cells in clonal expansion.

Reviewer 2, Figure 1A: EYFP fluorescence in bladder and prostate of *Tmprss2-CreER^{T2};R26-EYFP* after systemic tamoxifen and intravesical 4-OHT administration.

Reviewer 2, Figure 1B: Manipulation of target genes in *Tmprss2-CreER^{T2};Kmt2c;Kmt2d;R26-EYFP* cells. Urothelial cells were treated by 4OHT for 4h and then followed by medium changing. Flow cytometry and genomic DNA genotyping were performed 48 hours post the treatment.

Reviewer 2, Figure 1C: Quantification of *Kmt2c* and *Kmt2d* BaseScope signal post tamoxifen administration. (Shown in Extended Data Fig. 2c)

Quantifying Field effects in the Urothelium – The authors provide clear histology demonstrating normal like structure of the urothelium in the tamoxifen induced *Kmt2c/d* urothelium. However, A major weakness is that field effects are implied in the histologically normal with a correlative downstream assessment of H3K4me1. I find these stains difficult to interpret and not altogether convincing. Although, qPCR is provided indicating robust KO, in situ confirmation is necessary. One easy alternative, would be to perform RNAScope for *Kmt2c/d* (2 color) plus markers of the urothelium.

Response:

We really appreciate this suggestion of detecting *Kmt2c/d* deletions with in situ hybridization technique. Here, because the total RNA is still expressed and only the deleted exon is lost, we had to use BaseScope instead of RNAScope. With BaseScope, we can only quantify one target per slide, using serial sections. We designed BaseScope for the floxed exons of *Kmt2c* (Exon3) or *Kmt2d* (Exon50-51). Six months post tamoxifen administration, we observed efficient deletions of *Kmt2c/d* in both bladder and ureter urothelial cells (**Reviewer 2, Figure 2A**, shown in Extended Data Fig. 1c,d and Supplementary Fig. 9a.b). BaseScope of *Kmt2c/d* and IHC of H3K4me1 were also provided for tumorigenic models (*Pten* deletion, BBN treatment) in the study (**Reviewer 2, Figure 2B, and Figure 3A, B** for major question #3).

Reviewer 2, Figure 2A: BaseScope of *Kmt2c* and *Kmt2d* floxed exons 6 months after systemic tamoxifen of the indicated mice.

Reviewer 2, Figure 2B: BaseScope of *Kmt2c* and *Kmt2d* floxed exons 6 months after systemic tamoxifen of the indicated mice with neoplasia and *Pten* deletion.

#Quantifying Field effects as a driver of “primed” or a “license” for tumorigenesis – Figure 6 cooperation experiments are considered important but do again do not unequivocally show mutual loss of PTEN/*Kmt2c*/*Kmt2d* in the epithelium. Additional in situ analysis of each allele and important differentiation markers are necessary. Figure 7 is arguably one of the most important figures presented to support the conclusion that field effects prime / license the epithelium for tumorigenesis. The organoid experiments are appropriately conceived but the analyses are lacking from the perspective of field effects and tumors. In the tissues collected could sections be used for field assessment (i.e. how does the *Kmt2c/d* field size impact tumor growth)? Similarly, these studies are difficult to assess in the autochthonous model (Figure 7hi) due to the H3k4me1 staining. Additional mRNA FISH for the dKO alleles should be considered. Moreover, the BBN treatment really assesses cooperativity and not field driven effects. A mosaic experiment (i.e. low dose tamoxifen) and careful assessment of tumors for field origin (i.e. *Kmt2c/d* loss) would be highly impactful and support the overall conclusion.

Response:

To address these important questions, we first performed BaseScope targeting *Kmt2c/d* floxed exons and observed efficient deletions of *Kmt2c/d* 6 months post tamoxifen administration (**Reviewer 2, Figure 2B**). We further provided IHC staining of H3K4me1, Pten, Krt5, and Upk2 in bladder and ureter tissues as supplement to Fig. 6 (**Reviewer 2, Figure 3A**, shown in Extended Data Fig. 8d and Supplementary Fig. 9c). BaseScope of *Kmt2c* and *Kmt2d* were also provided for BBN models (**Reviewer 2, Figure 3B**, shown in Supplementary Fig. 10d). We did appreciate some cells positive by *Kmt2c/d* BaseScope in more established tumors which we believe are from CD45⁺ immune cell infiltration in these cancer models (**Reviewer 2, Figure 3C**).

In BBN models, we started BBN administration two weeks post tamoxifen administration. We agree this model is designed to compare the tumorigenic sensitization in *WT* and *Kmt2c/d* dKO urothelial cells, and it is insufficient to prove field cancerization. However, we did observe progressive increase of *Kmt2c/d* KO cells in mouse bladder (**Reviewer 2, Figure 1C**, shown in Extended Data Fig. 2c), suggesting clonal expansion advantage over *WT* urothelial cells.

Assessment of the sequence of genetic events cannot be done well in our GEMM model and would best be assessed using different recombinases and different alleles (e.g., Cre-loxP and Flp/FRT). Due to the time required for these experiments, we chose an alternative approach. In bladder urothelial organoids:

1. To model deletion of *Pten* first followed by *Kmt2c/d*, we generated bladder organoids from *Tmprss2-CreER^{T2};Kmt2c^{fl/fl};Kmt2d^{fl/fl}* mice and used CRISPR/Cas9 to knock out *Pten* in vitro. We then deleted *Kmt2c/Kmt2d* using 4-OHT/Tamoxifen, either in vitro before grafting or in vivo after grafting.
2. To model deletion of *Kmt2c/d* first followed by *Pten*, we generated bladder organoids from *Tmprss2-CreER^{T2};Pten^{fl/fl}* mice and used CRISPR/Cas9 to knock out *Kmt2c* and *Kmt2d* in vitro. We then deleted *Pten* using 4-OHT/Tamoxifen, either in vitro before grafting or in vivo after grafting.

Compared to *Pten* loss at first, more robust tumorigenesis was observed when we first knocked out *Kmt2c/d* (**Reviewer 2, Figure 3D**, shown in Fig. 7g). Together, these results suggest that *Kmt2c/d* loss urothelial cells are primed with clonal expansion advantage and with enhanced tumorigenic susceptibility.

Reviewer 2, Figure 3A: IHC of H3K4me1, Pten, Krt5, and Upk2 in mouse ureter and bladder tissues. (Shown in Extended Data Fig. 8d and Supplementary Fig. 9c)

Reviewer 2, Figure 3B: BaseScope staining and IHC staining in representative BBN models. (Shown in Supplementary Fig. 10d)

Reviewer 2, Figure 3C: IHC of CD45 in mouse UC models.

Reviewer 2, Figure 3D: Sequence of gene deletion matters in UC initiation. (Shown in Fig. 7g)

Single cell genomics analyses – Comparing multiple datasets in scRNASeq can be challenging, and many techniques have been developed (including various data integration techniques developed by R. Satija). The current methods do not adequately describe which technique is being used. In addition, alternative non integrative techniques should also be considered and cell typing using available workflows (i.e. SCType) should be considered. The current datasets only include N = 4 (WT) and N = 3 (dKO) so drawing conclusions regarding cell state changes is difficult. Additional biological replicates are necessary – and could be performed using alternative techniques that are more cost effective (i.e. RNAscope, IF of multiple samples) + markers of each cell lineage.

Response:

In our scRNA-seq experiments, we performed all experiments on the same day and we did not see batch difference of samples under the same condition (**Reviewer 2, Figure 4A**, shown in Supplementary Fig. 1a). Also, we did not use data from other experiments for integrated analyses. To validate the decreased differentiation with *Kmt2c/d* dKO, we performed immunofluorescence staining of Upk2 and Krt20 on multiple mice. In addition to gain of cells positive for Krt14, we observed no expression of Krt20 and thinner staining of Upk2 in dKO urothelium (**Reviewer 2, Figure 4B**, shown in Fig. 2f).

Reviewer 2, Figure 4A

Reviewer 2, Figure 4B

#impaired basal cell differentiation – This conclusion requires additional experiments that include appropriate Cre drivers in the basal compartment. The authors remark that the *Tmprss2*-CreER is present in all layers of the urothelium. So, it is unclear if the basal state found in the dKO is due to impaired differentiation or induction of cellular plasticity from differentiated progeny (i.e. intermediate cells and umbrella cells).

Response:

This is an important caveat of our *in vivo* models especially given that the urothelium exhibit slow turnover. We have changed some assertions to include both possibilities and we have added this to the discussion section. *In vitro*, we also observe impaired differentiation of dKO organoids that are placed in “differentiation media” (Reviewer 2, Figure 5A, shown in Extended Data Fig. 3d).

We performed intravesical injections of Adeno-K5-Cre (Cre expression driven by bovine *Krt5* promoter) and Adeno-K8-Cre (Cre expression driven by mouse *Krt8* promoter) (Reviewer 2, Figure 5B, shown in Extended Data Fig. 9) in *Tmprss2*-

CreER^{T2};Kmt2c^{ff};Kmt2d^{ff};Pten^{ff} mice. We did not observe any tumorigenesis in Krt8-Cre group, which preclude the analysis of differentiation marker expression in this model. These results suggest at least basal cells are the tumorigenic cell or origin, consistent with prior report that K5-Cre are more efficient than K8-Cre in tumorigenic induction of *Pten^{ff};Trp53^{ff}* knockout model (PMID: 34470779). We could not exclude the possibility of de-differentiation from intermediate or umbrella cells.

Reviewer 2, Figure 5A (Shown in Extended Data Fig. 3d): Intracellular FACS of *WT* and *dKO* organoids for K5 and K8 in full and differentiation media showing decreased level of luminal marker K8.

Reviewer 2, Figure 5B: Bladder specific deletion of *Kmt2c/Kmt2d/Pten* using intravesical delivery of 4OHT, Adeno-CMV-Cre, and Adeno-K5-Cre induced muscle invasive UC (also shown in Extended Data Fig. 9). BaseScope shows biallelic loss of both *Kmt2c* and *Kmt2d* transcript and IHC shows loss of H3K4me1 in tumor cells with retention in stromal cells.

Epigenetic analyses – The epigenetic analysis appears to be quite comprehensive, but as written it is extremely dense and challenging to follow.

Response:

As suggested, we have refined the manuscript and figures.

Therapeutic vulnerability to EGFR inhibitors – The EGFR inhibition studies are considered somewhat underdeveloped and overstated. For instance, Afatinib and Gefitinib both seem to be equally potent on WT and dKO cells - multiple independent experiments and statistical assessment of EC50 would be necessary to prove otherwise. In addition, in the natural state a mixture of WT and dKO cells would likely be present in the field. How does mixing affect responses and could clonal competition / vulnerability be assessed? The experiment in 8c is still very early in the tumor growth phase. It is unclear if treatment will reduce tumor growth rates if the tumors are allowed to grow to tumor endpoint (2000 mm³). The representative stains in 8d are also of low quality.

Response:

We repeated multiple independent experiments and there is an approximately 3 fold difference in EC50 (**Reviewer 2, Figure 6A**, shown in Fig. 8a,b). In a complementary experiment, we observed significantly decreased dKO cell number after 4 days treatment of 100 nM Gefitinib, 100 nM Afatinib, or EGF withdrawal.

When we performed mixing experiments of EYFP positive dKO cells with EYFP-negative WT cells, we observed depletion of dKO cells over time-despite no significant difference in growth rates when cells were separately maintained (**Reviewer 2, Figure 6A, panel C control condition, and Reviewer 2, Figure 6B**). This is opposite of enrichment of dKO cells in vivo (**Reviewer 2, Figure 1C**). In vitro, we did observe faster depletion dKO cells with EGFR inhibitors (**Reviewer 2, Figure 6B**).

We were not surprised by the different enrichment *in vitro* and *in situ*. Unlike common tumor suppressors in many cancer types such as *TP53*, *RB1*, and *PTEN* that are growth suppressive in vitro and in vivo, lineage specific epigenetic tumor suppressors are highly context specific. We have observed similar phenomenon in our work with tumor suppressors BAP1 in melanoma (PMID: 29490280) and PRC2 components EED and SUZ12 in malignant peripheral nerve sheath tumors (MPNSTs) (PMID: 25240281).

For in vivo therapy, when we allow tumors grow bigger in our graft models, there are more and more necrosis area in the middle. We also observed ulcerated skin on tumors with size around 600-1000mm³ ($v = \frac{4\pi}{3} * \frac{a}{2} * \frac{b}{2} * \frac{c}{2}$, a, b, and c represent the length, width, and thickness of the tumor). As a result, we started the treatment at an earlier timepoint. We have replaced the representative staining of phospho-EGFR(Y845) and Ki67, and performed the statistical analyses (**Reviewer 2, Figure 6C**, shown in Fig. 8e,f).

Reviewer 2, Figure 6A: Responses of WT and dKO urothelial cells to EGF signaling inhibition. (Shown in Fig. 8a-c)

Reviewer 2, Figure 6B: Growth competition of WT and dKO urothelial cells to EGF signaling inhibition.

Reviewer 2, Figure 6C. Quantification of p-EGFR intensity and Ki67 positivity in grafted tumors. (Shown in Fig. 8e,f)

Minor Concerns:

Image analysis – several opportunities for image analysis are missed, and if included, would greatly bolster their conclusions. This includes panels like Figure 2f.

Response:

As suggested, quantification of Krt14 (Fig. 2g), Ki67 (Fig. 8f and Extended Data Fig. 3b), and p-EGFR (Fig. 8e) have been provided (**Reviewer 2, Figures 7A-C**).

Reviewer 2, Figure 7A

Reviewer 2, Figure 7B

Reviewer 2, Figure 7C

Organoids and cell lines – neither model is well described in the text. In addition, confirmation of dKO status is not clear. This is especially important when considering the bottlenecking that occurs during selection for organoid growth and cell line growth. The invasion potential seems to be underdeveloped, and contrary to the idea that the dKO are histologically normal but show increased invasion potential ex vivo? The increased EGF responsiveness is certainly interesting, but it is unclear how this changes the phenotype of the cells in their in vitro assays.

Response:

We added more details of cell generation in result and method section. In *WT* and *dKO* urothelial cells (*T2-Cre;Kmt2c^{fl/fl};Kmt2d^{fl/fl}* +/- adeno-Cre infection), we performed genotyping with primers amplifying the Floxed and excised alleles (**Reviewer 2, Figure 8A**, shown in Extended Data Fig. 4a-c). We could not find an anti-Kmt2c antibody that works for WB and we only performed Western blot of Kmt2d. Additionally, protein levels of other Kmt2 components (Kmt2a/b-Menin, Set1a/b-Cxxc1) were also assayed to better support the mechanistic conclusion.

Despite the minor histological changes, we observed increased expression of *Zeb2* and *Fn1* in scRNA-seq (**Reviewer 2, Figure 8B**). We further identified increased *Zeb2* and *Twist1* mRNA expression in cultured *dKO* urothelial organoid (**Reviewer 2, Figure 8C**, shown in Extended Data Fig. 3e).

We also tested the impact of EGFRi on differentiation and EMT marker expression, but we did not see consistency of their expression (**Reviewer 2, Figure 8D**). Since the consequence of EGFRi on differentiation is not the focus of this study, we did not pursue this further. We feel the differentiation and EGF dependence are more like independent consequences with *Kmt2c/d* KO.

Reviewer 2, Figure 8A: Validation of *Kmt2c/d* deletion in WT and dKO urothelial cells. (Shown in Extended Data Fig. 4a-c)

Reviewer 2, Figure 8B: Heatmap of EMT markers from scRNA-seq in WT and dKO urothelial cells.

Reviewer 2, Figure 8C: Quantitative PCR of EMT markers in urothelial organoid. (Shown in Extended Data Fig. 3e)

Reviewer 2, Figure 8D: Quantitative PCR of *Krt6b*, *Upk2*, and *Zeb2* expression in WT and dKO cells treated with EGFR blockade.

Figure layouts – some of the figures are challenging to follow. Consider Figure 3 as an example, the panel layout is out of order. Others, including Figure 7 are challenging to follow in the current layout.

Response:

We agree with the reviewer, we have reorganized the layout of most figures.

Model in Figure 8e – the proposed model in Figure 8 is very challenging to follow and not well described in the text / legend.

Response:

As suggested, we have drawn a new illustration model.

Reviewer #3:

Remarks to the Author:

The study by Wang et al investigates the role of KMT2C and KMT2D in the urothelium primarily using a genetically engineered mouse model. The mouse model is based on a Cre driver that the authors had previously identified for its use in prostate cancer, however, the Cre driver is not restricted to prostate but rather achieves gene recombination in a variety of other tissues, one of which being the bladder urothelium, which is the focus of the current study. It is important to note at the outset that the authors do not explicitly discuss the distribution of Cre recombination in these mice and that this reviewer had to look up several publications to piece this together. Not only do the authors not specifically address this point, the data presented in the manuscript including in the supplementary figures (S1 and S2) do not explicitly look at other tissues, only bladder. This raises serious concerns that the authors intentions are not to be forthcoming about this very important feature of their mouse model. Additionally, the fact that the Cre driver is not limited to bladder also raises the issue about the cell autonomy of the phenotype; this is partially but fully addressed by the organoid models.

Response:

We thank the reviewer for pointing out this critical question. As reported previously (PMID: 27536883) *Tmprss2* is highly expressed in the epithelium of mouse bladder, prostate, and gastrointestinal tracts (**Reviewer 3, Figure 1A**). In that publication, we noted that *Pten* deletion caused intraepithelial neoplasia in the prostate but not large intestine or bladder while *Apc* deletion cause neoplasia of the intestine but not prostate or bladder.

We agree that description of mice phenotype and morbidity was inadequate in our initial submission, and we have extensively amended the manuscript. *Tmprss2-CreER^{T2};Pten^{ff}* mice exhibited normal longevity and the prostate exhibited prostatic intraepithelial neoplasia (mPIN) with occasional invasive carcinoma that does not affect the overall health or activity of the mice. When we observed early mortality when *Kmt2c^{ff}* and/or *Kmt2d^{ff}* were added, we first examined the prostate in detail. However, we did not observe any acceleration of prostate tumorigenesis (**Reviewer 3, Figure 1B**). After establishing BaseScope to quantify *Kmt2c* and *Kmt2d* deletion, we observed that the prostate was inefficiently deleted, suggesting the lack of competitive advantage of *Kmt2c* and *Kmt2d* loss in the prostate and highlighting the organ context specificity of tumorigenesis (**Reviewer 3, Figure 1C**).

We have updated the introduction of *Tmprss2-CreER^{T2}* specificity in the manuscript. We further provided the moribund phenotypes of our GEMM models (**Reviewer 3, Figure 1D**), from health check reports of the veterinarians in our vivarium. Moribund mice in *Tmprss2-CreER^{T2};Kmt2c^{ff};Pten^{ff}* and *Tmprss2-CreER^{T2};Kmt2d^{ff};Pten^{ff}* mice demonstrated bladder distension, hydronephrosis and ureter enlargement. The swollen abdomen was caused by distended bladder and/or distended kidney (Supplementary Table 2). Blood test of renal panel from *Tmprss2-CreER^{T2};Kmt2c^{ff};Pten^{ff}* mice were also provided as Supplementary Table 3. We observed profound increase of BUN and CREA in male mice (**Reviewer 3, Figure 1E**, shown in Supplementary Table 3),

indicating the development of renal failure. Mice in *Tmprss2-CreER^{T2};Kmt2c^{ff};Kmt2d^{ff};Pten^{ff}* group developed malignant gastric cancer, diarrhea, and hematochezia (Reviewer 3, Figure 1D, shown in Supplementary Table 2). But we did observe early tumor formation and increased Ki67+ cells in both ureter and bladder at 6-week timepoint post tamoxifen administration (Fig.6c-e).

It is important to note that while the expression of *Tmprss2* is not bladder specific, its expression within the bladder is specific to urothelial cells. Experimentally, we have not observed *Tmprss2-CreER^{T2}* mediated recombination in any non-epithelial cells, either assayed by IHC/IF, FACS, or single cell sequencing (Fig. 1b,g; Extended Data Fig. 1b-d). Thus, within the bladder, the Cre does not drive recombination in stromal cells that can affect the microenvironment.

To formally determine if urothelial only Cre is sufficient to cause tumorigenesis, we performed intravesical injections of 4OHT (4-hydroxytamoxifen), adeno-CMV-Cre (Cre driven by CMV promoter), adeno-K5-Cre (Cre driven by bovine keratin 5 promoter), and adeno-K8-Cre (Cre driven by mouse keratin 8 promoter) in *Tmprss2-CreER^{T2};Kmt2c^{ff};Kmt2d^{ff};Pten^{ff}* mice. Despite the diverse tumorigenic efficiency, muscle invasive urothelial cancer was observed in most tumors (Reviewer 3, Figure 1F, shown in Extended Data Fig. 9). As these are models of bladder specific target gene deletion, we did not see tumorigenesis in adjacent prostate tissue (Reviewer 3, Figure 1G) or distal stomach tissue (data not shown).

Consistent with GEMMs and allograft models, increased basal differentiation and decreased luminal differentiation were observed in these tumors (Reviewer 3, Figure 1F, shown in Extended Data Fig. 9).

Reviewer 3, Figure 1A: *Tmprss2* mRNA levels in mouse tissues (PMID: 27536883).

Reviewer 3, Figure 1B: HE staining of prostate from compound GEMMs 6 months post tamoxifen injection.

Reviewer 3, Figure 1C: BaseScope of *Kmt2c* and *Kmt2d* in prostate tissues stained on the same slide with bladder tissue. Note that a lot of cells are positive for *Kmt2c* and/or *Kmt2d* floxed alleles.

Reviewer 3, Figure 1D: Moribund phenotypes of *Tmprss2-CreER^{T2}* mediated GEMM models from health check reports of our veterinarians in the vivarium. (Shown in Supplementary Table 2)

	Mouse #	Months post tamoxifen administration	Gender	Phenotypes
Tmprss2-CreER^{T2};Kmt2c^{fl};Pten^{fl}	#1	7	Male	Hunched and lethargic with bladder distension and enlarged ureter
	#2	7.6	Male	Lethargic and unbalanced with bladder distension
	#3	4.4	Male	Lethargic with bladder distension
	#4	4.2	Male	Extremely lethargic with bladder distension
	#5	3	Male	Moribund with bladder distension
	#6	4.2	Male	Hunched and sluggish with bladder distension
	#7	3.5	Male	Moribund with bladder distension
	#8	4.7	Male	Extremely lethargic with bladder distension
	#9	7.2	Male	Extremely lethargic with bladder distension
	#10	6.5	Male	Hunched and sluggish with bladder distension
	#11	11	Female	Sluggish with hydronephrosis
	#12	6.7	Female	Hunched and sluggish with hydronephrosis and enlarged ureter
	#13	5.4	Female	Hunched and sluggish with hydronephrosis
	#14	7.6	Female	Distended abdomen with hydronephrosis
Tmprss2-CreER^{T2};Kmt2d^{fl};Pten^{fl}	#1	5.5	Male	Hunched and sluggish with bladder distension
	#2	6.3	Male	Distended abdomen with bladder distension, hydronephrosis and enlarged ureter
	#3	6.5	Male	Moribund with bladder distension, hydronephrosis and enlarged ureter
	#4	7	Male	Distended abdomen with hydronephrosis
	#5	6	Male	Hunched and sluggish with hydronephrosis and enlarged ureter
	#6	9.1	Male	Distended abdomen with hydronephrosis and enlarged ureter
Tmprss2-CreER^{T2};Kmt2c^{fl};Kmt2d^{fl};Pten^{fl}	#1	7 weeks	Male	Moribund with gastric cancer
	#2	7 weeks	Male	Moribund with gastric cancer
	#3	4.2 weeks	Male	Moribund with gastric cancer
	#4	4.2 weeks	Male	Moribund with gastric cancer
	#5	4 weeks	Male	Moribund with gastric cancer
	#6	6 weeks	Male	Moribund with gastric cancer
	#7	6.4 weeks	Male	Moribund with gastric cancer
	#8	6.2 weeks	Male	Moribund with gastric cancer and enlarged ureter
	#9	9 weeks	Male	Moribund with gastric cancer
	#10	12 weeks	Male	Moribund with gastric cancer and enlarged ureter
	#11	4 weeks	Female	Moribund with gastric cancer
	#12	7 weeks	Female	Moribund with gastric cancer and enlarged ureter
	#13	2 weeks	Female	Moribund with gastric cancer
	#14	3.5 weeks	Female	Moribund with gastric cancer
	#15	3.5 weeks	Female	Moribund with gastric cancer
	#16	4.2 weeks	Female	Moribund with gastric cancer
	#17	4.2 weeks	Female	Moribund with gastric cancer
	#18	7 weeks	Female	Moribund with gastric cancer

Reviewer 3, Figure 1E: Representative blood test of renal panel in male *Kmt2c^{fl};Pten^{fl}* mouse. (Shown in Supplementary Table 3)

Tmprss2-CreER^{T2};Kmt2c^{fl};Pten^{fl} , #1	Test ID: 1800479801	Result values	Reference range
Male, 7.1 months post tamoxifen	[BUN (mg/dL)]	363	5.0-28
	[CREA (mg/dL)]	4.12	0.2-0.5
	BUN/CREA Ratio	85.9	--
	[TP (g/dL)]	7.3	4.8-7.2
	[ALB (g/dL)]	3.2	2.4-4.3
	GLOB (g/dL)	4.1	1.7-2.2
	A/G Ratio	0.8	--
	[P (mg/dL)]	34.5	7.3-14.5
	[Ca (mg/dL)]	0.3	9.5-12.5
	[TCO2 (mEq/L)]	10	--
	[Na (mEq/L)]	133	145-181
	[K (mEq/L)]	29.3	7.3-11.1
	[CL (mEq/L)]	83	111-134
	[Na/K]	5	--
	[Anion Gap]	69	--

Reviewer 3, Figure 1F: Bladder specific deletions of *Kmt2c*/*Kmt2d*/*Pten* induces muscle invasive UC. (Shown in Extended Data Fig. 9)

Reviewer 3, Figure 1G, Representative H&E and gross morphology of prostate tissues with bladder tumors induced by intravesical adenoviral Cre administration.

Bearing in mind this important caveat (regarding the validity of their GEMM), what the authors show is that, on their own, KMT2C/KMT2D, are not sufficient to promote bladder tumorigenesis since GEMMs lacking KMT2C/KMT2D have only a subtle histological phenotype. In collaboration with other tumor drivers, namely loss of PTEN or the carcinogen BBN, the phenotype is modestly potentiated. In fact the PTEN compound mice die of an alternative cancer type. To this reviewer, these findings suggest that KMT2C/KMT2D are insufficient for bladder cancer; however, the authors conclude that KMT2C/KMT2D promote initiation. It is difficult to even understand how they could come to this conclusion based on the data provided. It might be possible to make this conclusion about initiation, if they temporally ordered the events (so for example initiated PTEN deletion and then KMT2C/KMT2D or vice versa). But the definitive experiments are not provided and instead we are left with a rather weak phenotype and an unsubstantiated conclusion. Another possibility is that the collaborating event has not been tested - for example, perhaps p53 would be more relevant than PTEN (since p53 is mutated in >50% of invasive bladder cancer). In fact, PTEN is rarely mutated in invasive bladder cancer and almost never together with KMT2C/KMT2D.

Response:

We agree that “initiation” is a poor choice of words since the mouse urothelial tissue is histologically normal, as in humans. We will change to “prime” instead. To address these questions, we first provided moribund phenotypes and blood test results of mice from *Tmprss2-CreER^{T2};Kmt2c^{ff};Pten^{ff}* and *Tmprss2-CreER^{T2};Kmt2d^{ff};Pten^{ff}* groups (Reviewer 3, Figure 1D, E). We have sent euthanized moribund *Tmprss2-CreER^{T2};Kmt2c^{ff};Pten^{ff}* mice for necropsy by MSK Comparative Department of Pathology and Laboratory Medicine and there was PIN in the prostate, hydronephrosis and urothelial neoplasia, and no other abnormalities. A representative whole section of *Tmprss2-CreER^{T2};Kmt2c^{ff};Pten^{ff}* mouse that include prostate anterior, ventral, and dorsolateral lobes and the prostatic urethra highlights the PIN in the prostate and

invasive urothelial cancer in the urethra (**Reviewer 3, Fig 2A**). Similarly, a representative section *Tmprss2-CreER^{T2};Kmt2c^{fl/fl};Pten^{fl/fl}* mouse prostate and separate section of the urethra showed similar findings (**Reviewer 3, Fig 2B**).

We observed early formation of urothelial cancer in *Tmprss2-CreER^{T2};Kmt2c^{fl/fl};Kmt2d^{fl/fl};Pten^{fl/fl}* group, but mice died of malabsorption, hematochezia and had gastric cancer. To study the bladder in isolation, we performed intravesical injections of 4OHT (4-hydroxytamoxifen), adeno-CMV-Cre (Cre driven by CMV promoter), adeno-K5-Cre (Cre driven by bovine keratin 5 promoter), and adeno-K8-Cre (Cre driven by mouse keratin 8 promoter) in *Tmprss2-CreER^{T2};Kmt2c^{fl/fl};Kmt2d^{fl/fl};Pten^{fl/fl}* mice. We observed muscle-invasive urothelial cancer in most tumors (**Reviewer 3, Figure 1F**, shown Extended Data Fig. 9).

We agree with the reviewer that *PTEN* aberrations is not as high as *PIK3CA* or *TP53* in urothelial cancer (**Reviewer 3, Figure 2C**). For *PTEN*, in addition to known drivers of homozygous deletion and detrimental mutations (6%), there is a substantial group with heterozygous loss (37%), at least some are focal (**Reviewer 3, Figure 2C**; shown in Extended Data Fig. 7).

In addition to *Pten* loss, we have tested the tumorigenic susceptibility to *KRAS G12V* mutation, *PIK3CA E545K* mutation, and *Trp53* loss in allograft assay (Shown in Fig. 7d-f). We further found that tumors from *Pten* loss and *PIK3CA-E545K* groups demonstrated similar histology (**Reviewer 3, Figure 2D**, shown in Fig. 7f).

Reviewer 3, Figure 2A: Representative HE staining of prostatic urethra with anterior prostate, ventral prostate, and dorsal-lateral prostate from *Tmprss2-CreER^{T2};Kmt2c^{fl/fl};Pten^{fl/fl}* mice.

Reviewer 3, Figure 2B: Representative HE staining from *Tmprss2-CreER^{T2};Kmt2d^{fl/fl};Pten^{fl/fl}* mice.

Kmt2d^{fl/fl};Pten^{fl/fl}

Anterior prostate

Prostatic urethra

Reviewer 3, Figure 2C: Oncoprint of TCGA Bladder Cancer of indicated genes. Middle, Oncoprint includes heterozygous loss of *PTEN* (cyan). Bottom panel is copy number landscape of Chr10 showing focal deletion of *PTEN*. (Shown in Extended Data Fig. 7)

Reviewer 3, Fig 2D; Representative HE and Krt5 IHC in grafted tumors. Note that no Upk2 staining was observed in these tumors. (Shown in Fig. 7f)

[This reviewer notes that the characterization of the GEMM phenotype is missing basic characterization of mouse cancer models, such as tumor (bladder) weights, proliferation rate, etc. Ki67 is shown for Pten in Fig. 6]

Response:

Due to distention of the bladder, we did not measure bladder weights in the GEMMs. We do have whole sections that show the size of distention and neoplasia (for example, Fig. 6c,d, Extended Data Fig. 9a, and Supplementary Fig. 10a).

To better describe the phenotypes in our GEMMs, we provided the health check reports from veterinarian of our vivarium. We also provided the morphology of bladder and ureter tissues upon the dissection (**Reviewer 3, Figure 1D**, shown in Supplementary Table 2). We further provided the blood chemistries of several mice showing terminal renal failure in moribund mice (**Reviewer 3, Figure 1E**, shown in Supplementary Table 3), from which we observed profound increase of phosphorus, BUN, and CREA in male mice (n=4 mice), suggesting the development of renal failure. In contrast, in the female mice (n=2 mice), we observed only slight upregulation of BUN, despite the development of hydronephrosis.

For histologic sections, we performed Ki67 quantification, IHC of H3K4me1, Pten, Krt5, and Upk2 in GEMMs (**Reviewer 3, Figure 3A**, shown in Fig. 6e, Extended Data Fig. 8d, and Supplementary Figs. 9c).

For new experiments using adeno-Cre injection, due to focal tumor formation, there was not distension from urinary outlet obstruction. We have provided the weight of bladder in mice with adeno-Cre injection (**Reviewer 3, Figure 3B**, shown in Extended Data Fig. 9c).

Reviewer 3, Figure 3A: IHC of H3K4me1, Pten, Krt5, and Upk2 in mouse ureter and bladder tissues. (Shown in Extended Data Fig. 8d, and Supplementary Figs. 9c).

Reviewer 3, Figure 3B: Weight of bladder post intravesical delivery of Adeno-Cre virus. (Shown in Extended Data Fig. 9c)

Following from the observations that KMT2C/KMT2D loss in GEMMs results in modest effects on bladder histopathology, the authors then embark on a series of studies to characterize their molecular phenotype. They show that these mice have more basal features and fewer luminal ones. This is an interesting observation but not evidence of a field effect. The fact that the normal adjacent bladder urothelium has similar molecular changes is correlative not causal evidence. Definitive experiments to show causality are not provided.

Response:

We apologize for the inadequate presentation of the histopathology in our initial submission. We have re-reviewed all the slides with expert bladder pathologist Hikmat A. Al-Ahmadie who has also reviewed several mouse urothelial tumor models (PMID: 37662238, 36761730, 34470779, 28082400, 27376116). Our animal pathology core has also reviewed some slides. We have selected lower magnification regions that more clearly show the histopathology of the bladder and ureter (**Reviewer 3, Fig. 4A**, shown in Fig. 6 and Extended Data Fig. 8b). In the ureter, no *Pten^{fl/fl}* mice but many *Kmt2c^{fl/fl};Pten^{fl/fl}* and *Kmt2d^{fl/fl};Pten^{fl/fl}* exhibited grossly invasive and microinvasive cancer. In the bladder, *Kmt2c^{fl/fl};Pten^{fl/fl}* and *Kmt2d^{fl/fl};Pten^{fl/fl}* mice developed thickened urothelium with dysplasia and carcinoma in situ. We have quantified the ureter based on invasive cancer and the bladder based on evidence of dysplasia and carcinoma in situ. In our view, the histopathology of the urothelium and tumor suppressive effect of *Kmt2c/Kmt2d* loss are quite striking. In addition, IHC of Ki67, Upk2, and Krt5 is also clearly different when *Kmt2c* and/or *Kmt2d* is lost (**Reviewer Fig. 3, Fig 4A**, shown in Fig. 6 and Extended Data Fig. 8d). We'll be happy to provide any additional whole slides for the reviewer to review.

Field effect consists of two criteria: 1) clonal selection and selective advantage, and 2) increased susceptibility to tumorigenesis with other drivers such as *PTEN* loss, *TP53* loss, *PIK3CA* and *KRAS* activating mutations.

To investigate whether *Kmt2c/d* loss urothelial cells may have clonal expansion advantage, we performed BaseScope with probes targeting floxed exons of *Kmt2c* (Exon3) and *Kmt2d* (Exon50-51). We treated the mice with two doses of 3mg tamoxifen (I.P.) and collected bladder tissues from *Control* (*Tmprss2-CreER^{T2}/R26-EYFP*, collected at 6-month) and *dKO* groups (*Tmprss2-CreER^{T2};Kmt2c^{ff};Kmt2d^{ff}* collected at 1-week, 6-week, and 6-month). We observed progressive decrease of *Kmt2c* and *Kmt2d* signal over time (**Reviewer 3, Figure 4B**, shown in Extended Data Fig. 2c), suggesting the growth advantage of *Kmt2c/d* loss urothelial cells in clonal expansion.

For criteria #2, Our data suggest that *Kmt2c/d* loss primes urothelial cells to tumorigenesis including GEMM of *Pten* deletion (Fig. 6). *Kmt2c/d* loss urothelium significantly accelerates bladder tumorigenesis by BBN treatment (**Reviewer 3, Figure 4C**, shown in Fig. 6g,h and Supplementary Fig. 10c). Notably, BBN does not commonly cause ureteral cancers and we observed frequent invasive ureteral cancers in *Kmt2c/d* loss mice. *Kmt2c/d* loss also cooperated with *Pten* deletion, *Trp53* deletion, *PIK3CA-E545K*, and *KRAS-G12V* in allograft experiments (**Reviewer 3, Figure 4D**, shown in Fig. 7d-f). The importance of *Kmt2c* and/or *Kmt2d* loss in the urothelium is highlighted by the inability of combined *Pten* loss and *KRAS-G12V* to form allografts (Shown in Fig. 7e).

Field effect implies that the sequence of events matter. To study the sequence of events, we performed the following experiment:

1. To model deletion of *Pten* first followed by *Kmt2c/d*, we generated bladder organoids from *Tmprss2-CreER^{T2};Kmt2c^{ff};Kmt2d^{ff}* mice and used CRISPR/Cas9 to knock out *Pten* in vitro. We then deleted *Kmt2c/Kmt2d* using 4-OHT in vitro or tamoxifen in vivo after grafting.
2. To model deletion of *Kmt2c/d* first followed by *Pten*, we generated bladder organoids from *Tmprss2-CreER^{T2};Pten^{ff}* mice and used CRISPR/Cas9 to knock out *Kmt2c/Kmt2d* in vitro. We then deleted *Pten* using 4-OHT in vitro or tamoxifen in vivo after grafting.

Compared to *Pten* loss at first, more robust tumorigenesis was observed when we first knocked out *Kmt2c/d* (**Reviewer 3, Figure 4E**, shown in Fig. 7g). Together, these results suggest that *Kmt2c/d* loss urothelial cells are primed with clonal expansion advantage and with enhanced tumorigenic susceptibility.

Reviewer 3, Figure 4A: Representative histopathology of ureter and bladder and quantification of neoplastic phenotypes. (Shown in Fig. 6 and Extended Data Fig. 8b)

Reviewer 3, Figure 4B: Quantification of *Kmt2c* and *Kmt2d* BaseScope signal post tamoxifen administration. (Shown in Extended Data Fig. 2c)

Reviewer 3, Figure 4C: Representative low-power bladder H&E of BBN treated WT and dKO mice (shown in Fig. 6h and Supplementary Fig. 10b,c)

Reviewer 3, Figure 4D: Allograft tumor formation efficiency (shown in Fig. 7e)

Tumorigenesis in Allografts

Genetic manipulations	Tumorigenesis
sgControl	0/10
sgKmt2c	0/10
sgKmt2d	0/10
sgdKO	0/10
sgControl+4OHT (Pten KO)	0/10
sgKmt2c+4OHT (Pten KO)	9/10
sgKmt2d+4OHT (Pten KO)	10/10
sgdKO+4OHT (Pten KO)	10/10
sgControl+PIK3CA (E545K)	0/10
sgdKO+PIK3CA (E545K)	10/10
sgControl+KRAS (G12V)	0/10
sgdKO+KRAS (G12V)	10/10
sgControl+sgTrp53	0/10
sgdKO+sgTrp53	10/10
sgControl+KRAS (G12V)+4OHT	0/10

Reviewer 3, Figure 4E: Sequence of gene deletion in UC initiation. (Shown in Fig. 7g)

Overall, this is a very disappointing study which is remarkably over interpreted while lacking key definitive experiments on causal relationships, and based on a GEMM that is essentially misrepresented.

Specific comments:

This reviewer notes that KMT2C and KMT2D actually differ in mice and humans (their names are actually inverted) - the authors do not note this (perhaps they are not aware of this).

Response:

Kmt2c in mice is the *KMT2C* homolog in humans (<https://www.informatics.jax.org/marker/MGI:2444959>) and *Kmt2d* in mice is *KMT2D* homolog in humans (<https://www.informatics.jax.org/marker/MGI:2682319>). The confusion may arise that both *KMT2B* and *KMT2D* have alternatively been referred to as MLL2 and MLL4. To avoid the confusion of terminology, we have provided gene IDs in the manuscript. *Kmt2c* (Gene ID: 231051) and *Kmt2d* (Gene ID: 381022). We avoid using the MLL terminology.

Figure 1: The phenotype of the GEMMs are not sufficiently well documented; the specificity of the Cre driver not provided or discussed; and the phenotype in other tissues in which the Cre is active is not discussed or described. The authors do not consider whether their findings are cell autonomous to the urothelium (or for example dependent on recombination of KMT2C/KMT2D in other tissues/cell types).

Response:

To address this concern, we added the introduction of *Tmprss2*-Cre activity in other tissues. We also added pathological descriptions of other tissues (e.g., prostate and stomach). It is important to note that while the expression of *Tmprss2* is not bladder specific, its expression within the bladder is specific to urothelial cells. Experimentally, we have not observed *Tmprss2*-CreER^{T2} mediated recombination in any non-epithelial cells, either assayed by IHC/IF or FACS (Fig. 1b, Extended Data Fig. 1). Further, H3K4me1 is notably preserved in the stroma cells and selectively lost in urothelium (Fig 1g).

Most importantly, we performed intravesical injections of 4OHT or Cre-expressing adenovirus to induce bladder specific deletions of *Kmt2c/Kmt2d/Pten* and observed muscle invasive urothelial cancer in 3 months (Shown in Extended Data Fig. 9).

Figure 2: The data in this figure show an increase in basal cells/features; these data are not sufficient to draw a conclusion that KMT2C/KMT2D alters "stem cell potential, basal differentiation or EMT". The results are modest and could be reflective of increased proliferation (which is not quantified in the figure).

Response:

As suggested, quantification of Ki67 was added to the result (**Reviewer 3, Figure 5A**, shown in Extended Data Fig. 2b). In the organoid formation assay (**Reviewer 3, Figure 5B**, shown in Fig 2i,j), we used freshly sorted urothelial cells and observed increased organoid formation efficiency, which is a commonly used method to evaluate stem cell potential in urothelial cells (PMID: 31562298). We further performed immunofluorescence staining of Krt14, a urothelial stem cell marker identified in prior report (PMID: 27320313). We identified significantly increased Krt14 positive cells in *Kmt2c/d dKO* group (**Reviewer 3, Figure 5B**, shown in Fig. 2f,g), consistent with high level of Krt14 expression in *dKO* cluster 1 (scRNA-seq, shown in Fig. 2d). To validate the decreased differentiation in *Kmt2c/d dKO* urothelium, we performed immunofluorescence staining of Upk2 and Krt20 in multiple mice. We observed no expression of Krt20 and thinner staining of Upk2 in *dKO* urothelium (**Reviewer 3, Figure 5B**, shown in Fig. 2f).

In scRNA-seq, we found increased expression of *Zeb2*, an EMT transcription factor in UC (PMID: 26328525), in *dKO* cluster 1 (**Reviewer 3, Figure 5C**, shown in Fig. 2d). Increased expression of *Zeb2* was also observed in cultured *dKO* urothelial organoid (**Reviewer 3, Figure 5D**, shown in Extended Data Fig. 3e). In transwell assay, we cultured *WT* and *dKO* urothelial cells on collagen-coated plates with organoid medium. Under this condition, we did not see significant difference of cell number after 4 days of culture (**Reviewer 3, Figure 5E**, shown in Fig. 8c). Moreover, we seeded the same number of *WT* and *dKO* cells on the top of transwell chamber and incubated for only 24 hours. These data suggest that the differential cell invasion was less likely caused by differential cell growth speed.

Reviewer 3, Figure 5A: Characterization of proliferation in WT and dKO urothelium. (Shown in Extended Data Fig. 2)

Reviewer 3, Figure 5B: Immunofluorescent staining of differentiation markers. (Shown in Figs. 2f-g and 2i-j)

Reviewer 3, Figure 5C: Heatmap of EMT markers in scRNA-seq.

Reviewer 3, Figure 5D: Quantitative PCR of EMT markers in urothelial organoid. (Shown in Extended Data Fig. 3e)

Reviewer 3, Figure 5E: Count of cells after 4 days of treatment on 2D culture. Comparisons were made between WT and dKO under each condition. (Shown in Fig. 8c)

Figure 3: It is very difficult to understand how these data support a "pre-tumorigenic" effect; the lack of cancerous phenotype (as is the case in the KMT2C/KMT2D GEMM) is not indicative of a "pre-tumor" phenotype it is actually better described as a non-tumor phenotype.

Response:

We agree that "pre-tumorigenic" is a poor description and we have changed it. This word has been historically used for premalignant polyps or PIN that has a neoplastic

phenotype but not invasive. Histologically normal human bladder with *KMT2C*, *KMT2D* or *KDM6A* mutations and our GEMM models are not truly pre-tumorigenic. We will change to use “primed” instead.

Figure 6: It seems unlikely from the phenotype that the mice are dying of bladder cancer, making panels b and c quite misleading.

Response:

While *Tmprss2-CreER^{T2}/Kmt2c^{ff}/Pten^{ff}* and *Tmprss2-CreER^{T2}/Kmt2d^{ff}/Pten^{ff}* mice do not die from urothelial tumor volume, they do die from renal failure that is a consequence of urothelial carcinoma obstructing the ureters and urethra. We will change the writing to clearly state the most proximal cause of death.

To better describe the phenotypes in our GEMM models, we provided the health check reports from veterinarian of our vivarium facility (**Reviewer 3, Figure 1D**, shown in Supplementary Table 2). We also provided the morphology of bladder and ureter tissues upon the dissection. Most moribund *Tmprss2-CreER^{T2};Kmt2c^{ff};Pten^{ff}* and *Tmprss2-CreER^{T2};Kmt2d^{ff};Pten^{ff}* mice demonstrated bladder distension, hydronephrosis and ureter enlargement. The swollen abdomen was exclusively caused by distended bladder and/or distended kidney. Blood test of renal panel from *Tmprss2-CreER^{T2};Kmt2c^{ff};Pten^{ff}* mice were provided as Supplementary Table 3 (**Reviewer 3, Figure 1E**), from which we observed profound increase of phosphorus, BUN, and CREA in male mice (n=4 mice), suggesting the development of renal failure.

In *Tmprss2-CreER^{T2};Kmt2c^{ff};Kmt2d^{ff};Pten^{ff}* group, moribund mice had diarrhea, hematochezia, and stomach cancer. But we have solved this issue by performing intravesical injections of 4OHT or Cre-expressing adenovirus. We observed muscle-invasive urothelial cancer in these models 3 months post injection (Shown in Extended Data Fig. 9).

Figure 7: These data seem out of place since they actually address the cooperatively of *KMT2C/KMT2D* with various other relevant drivers of bladder cancer. However, it is not possible to say that *KMT2C/KMT2D* prime the cancers since the temporal order of deletion is not controlled for. These data are also not fully developed and seem to be added after the fact, when in fact if properly developed might support the conclusions they would like to make. [These data would need to be much more fully developed to support their conclusions.]

Response:

We agree this model is designed to compare the tumorigenic sensitization in *WT* and *Kmt2c/d* *dKO* urothelial cells, and it is insufficient to prove field cancerization.

In BBN models, we treated the mice with BBN two weeks post tamoxifen administration to delete *Kmt2c* and *Kmt2d*. Prior whole exome sequencing experiments of BBN tumors

that develop in a longer time frame without *Kmt2c/d* deletion showed *Kmt2c* and *Kmt2d* were the most mutated genes (PMID: 29367767).

Assessment of the sequence of genetic events cannot be done well in our GEMM model and would best be assessed using different recombinases and different alleles (e.g., Cre-loxP and Flp/FRT). Due to the time required for these experiments, we chose an alternative approach of introducing the first mutation using CRISPR/Cas9 in vitro, grafting the urothelial cells and use tamoxifen to induce the second mutation. The data, as described above, suggest that *Kmt2c/d* deletion first is more tumorigenic than *Pten* deletion first (**Reviewer 3, Figure 4E**).

Figure 8: It is hard to understand how the authors could make a conclusion about EGFR inhibition KMT2C/KMT2D contests when they have not provided a strong case for the role of these genes in bladder cancer. They are also lacking the extremely obvious definitive connection to the molecular studies that they provided in the previous figures.

Response:

The cellular response to EGF is typically analyzed by observing phosphorylation of MAPKs and subsequent transcriptional activation of immediate-early genes (IEGs, PMID: 17322878). In scRNA-seq studies, gene sets of IEGs and EGF-stimulated genes were positively enriched with *Kmt2c/d* knockout (**Reviewer 3, Figure 6**, shown in Fig. 3a,b). Increased transcription of IEGs to EGF stimulation were further validated in *dKO* cells (**Reviewer 3, Figure 6**, shown in Fig. 3c,d), suggesting more robust response to EGF stimulation. We also demonstrated that IEGs such as *Egr1*, *Egr3*, *Nr4a1*, *Nr4a2*, and *Nr4a3* are characterized by higher CpG content at TSS and by elevated *Kmt2a* binding with *Kmt2c/d* KO (**Reviewer 3, Figure 6**, shown in Supplementary Fig. 7d). Based on these observations, we wanted to test whether *Kmt2c/d* loss may lead to dependence on EGF signaling, then we treated *WT* and *dKO* urothelial cells with EGFR inhibitors.

Reviewer 3, Figure 6:

Other specific comments:

Line 50 — “licence a field effect...” egregiously overstated and overintepreted.

Response:

We have provided more data in the revision to better support this statement.

Line 55 — These sentences do not connect

Response:

As suggested, we have changed the expression.

Line 67 — Needs a reference

Response:

Reference has been added.

Line 80-81 — Unclear what Kabuki syndrome and Kleefstra syndrome have to do with bladder cancer or any of the results presented herein.

Response:

We have deleted this from introduction.

Line 94-97 — Conclusions not supported by the data.

Response:

We have performed new experiment to better support the conclusions.

Line 104-105 — Since the data do not demonstrate a role for KMT2C/KMT2D in bladder cancer, hard to understand how they can nominate a druggable target.

Response:

In human bladder cancer and normal human bladder *KMT2C* and *KMT2D* loss of function mutations are strongly implicated as early genetic events through multiple studies. This forms a strong rationale to identify targets.

Line 112-113 — Alarmingly misleading since there is no mention that their Cre driver results in gene recombination in many tissues.

Response:

To address this concern, we performed intravesical injections of 4OHT or Cre-expressing adenovirus to induce bladder specific deletions of *Kmt2c/Kmt2d/Pten* and observed muscle invasive urothelial cancer in 3 months. (**Reviewer 3, Figure 1F**, shown in Extended Data Fig. 9)

Line 130-136 — Why look only at the catalytic function of KMT2C/KMT2D? Why not measure the deletion of KMT2C/KMT2D directly in the cells - this seems worrisome that these direct data are not shown. [It is often the case that the cells in which the genes are actually deleted can be eliminated several months later.]

Response:

The lack of suitable antibodies for IHC had previously limited our ability to directly detect loss of *Kmt2c* and *Kmt2d*. To validate the in situ deletions of *Kmt2c* and *Kmt2d* floxed alleles at the indicated timepoint, we performed BaseScope with probes targeting floxed exons (*Kmt2c* Exon3 or *Kmt2d* Exon50-51) in mouse ureter and bladder tissues (**Reviewer 3, Figure 7A**, shown in Extended Data Fig. 1c-d, and Supplementary Fig. 9a,b). Six months post tamoxifen administration, we observed efficient deletions of *Kmt2c/d* in both bladder and ureter urothelial cells. In urothelial organoids, we performed genotyping with primers amplifying floxed and excised alleles (**Reviewer 3,**

Figure 7B, shown in Extended Data Fig. 4c). Due to the shortage of good anti-Kmt2c antibody, we performed western blot of Kmt2d (**Reviewer 3, Figure 7B**, shown in Extended Data Fig. 4b).

Reviewer 3, Figure 7A: BaseScope of *Kmt2c* and *Kmt2d* floxed exons in bladder and ureter tissues (Shown in Extended Data Fig. 1c-d, and Supplementary Fig. 9a,b).

Continued on next page.

Reviewer 3, Figure 7B: Validation of *Kmt2c/d* deletion in urothelial organoids. (Shown in Extended Data Fig. 4a-c)

Link Redacted

If you wish to submit a suitably revised manuscript we would hope to receive it within 6 months. If you cannot send it within this time, please let us know. We will be happy to consider your revision so long as nothing similar has been accepted for publication at Nature Genetics or published elsewhere. Should your manuscript be substantially delayed without notifying us in advance and your article is eventually published, the received date would be that of the revised, not the original, version.

Nature Genetics is committed to improving transparency in authorship. As part of our efforts in this direction, we are now requesting that all authors identified as 'corresponding author' on published papers create and link their Open Researcher and Contributor Identifier (ORCID) with their account on the Manuscript Tracking System (MTS), prior to acceptance. ORCID helps the scientific community achieve unambiguous attribution of all scholarly contributions. You can create and link your ORCID from the home page of the MTS by clicking on 'Modify my Springer Nature account'. For more information please visit please visit www.springernature.com/orcid.

Thank you for the opportunity to review your work.

Sincerely,

Safia Danovi
Editor
Nature Genetics

Referee expertise:

Referee #1: bladder cancer models

Referee #2: field cancerisation

Referee #3: bladder cancer

Reviewers' Comments:

Reviewer #1:

Remarks to the Author:

Wang et al. report on the role of Kmt2c and Kmt2d in the urothelium using a variety of tools including genetic mouse models, chemical carcinogenesis, and genomics. The work is novel, original, and comprehensive. Their integrative epigenetic analysis linking Kmt2c/d to chromatin landscapes and gene regulation is a real tour de force and provides novel clues as to the mechanisms through which these two genes contribute to urothelial cancer. Specifically, they show that Menin plays a crucial role through its relocation to bivalent promoters associated with transcriptional de-repression. The GEMM experiments are also very strong but there is scarce evidence that the deletions are uniform and involve both alleles: only indirect measures of the effect of gene deletion is provided.

General comments

While the work is highly relevant, I have a few general concerns.

1. Most of the epigenetic analysis use an n=1 (for most of their ChIPseq experiments). I don't think that this is adequate for a paper submitted to Nat Genet. First, it does not allow assessing the reproducibility of the findings; second, it does not allow to perform adequate statistical analyses. This is particularly important considering that multiple alleles are target for Cre but there is not definite evidence that the alleles are consistently and uniformly deleted in all cells. For all critical experiments, they should perform ideally 3 biological replicates; otherwise, all critical data should be supported by targeted analyses (ChIP-Seq or else, as required). They even perform the studies on bulk RNA-Seq only in duplicates, which is not acceptable.
2. The Cre driver used in this work is not urothelial-specific; therefore, it is not possible rule out effects from deletion of target genes in other tissues. This should be addressed in the Discussion and the precise tissue activity of the Cre should be indicated.
3. The authors do not provide direct evidence of loss of protein expression in the experiments, not even in the critical ones. This is a severe limitation of the work.
4. The role of Menin is an important finding proposed in this work; however, there is a dearth of mechanistic insight about how this takes place. This should be addressed (or at least discussed) by the authors.

5. The Results are described in detail, but this section should be distilled to provide more specific messages to the reader. Need be, they can add some explanations in the Supplementary Data.

6. The authors fail to place their findings in the context of bladder cancer in humans. Inactivation of KMT2C and KMT2D occurs frequently in non muscle-invasive bladder cancer in patients, which means that it is associated with tumors showing urothelial - rather than basal - differentiation. Therefore, the authors should discuss how their model contributes to an understanding of the role of these mutations in human cancer. In addition, they should discuss how KMD6A/UTX and/or PTEN mutations cooperate with KMT2C/KMT2D mutations in humans. The relevance of the mouse findings to human cancer is insufficiently developed.

Specific comments

1. Page 3, lines 109-125: show protein expression in control urothelia as well as upon TMX-induced recombination.
2. Supplementary Fig. 2: cluster 7 is enriched in transcripts coding for ribosomal proteins- artifact? is this related to quality of these cells?.
3. Page 5, line 166: I'm surprised that, in general, there are no big expression differences amongst the clusters for the 5 basal marker genes in the WT populations..Are there significant differences?
4. Page 5, line 174 figure 2F: please, provide quantification of Krt14+ cells in dKO vs WT needed
5. Page 5, line 175 figure 2G: authors should clarify in the figure legend or method section the number of n's in each group of the comparisons. Furthermore, authors should test expression of the human orthologues of the genes in figure 2c-e for the comparisons in figure 2G.
6. Page 5, line 179: How was Mki67 positivity defined based on expression? Which expression threshold was used for saying that the cell is positive?
7. Page 6, lines 195-210: the analysis of differentiation is very superficial, I suggest to be more cautious regarding these statements - just remove the term "umbrella cells". Same applies to line 221 of the text.
8. Page 6, line 205 figure S3D: Authors should also evaluate Krt20 expression as this is uniquely expressed in umbrella cells and not in intermediate cells.
9. Page 6, line 225 figure 3A: Authors should also include the molecular subtype gene sets from Figure 2G or consensus subtype gene sets from Kamoun et al., 2020 in this analysis. Where do the luminal, basal, and squamous gene sets fall in this graph?
10. Page 7, line 233: Why are DKO cells more responsive to EGF? Which components of the EGFR signaling pathway is upregulated in DKOs? Are EGFRs upregulated?
11. Page 7, line 234 figure 3C,D: Is this non-organoid 2D cell experiments? How were they performed (freshly sorted urothelial cells, based of which markers, or cell lines established from the GEMMs)? the same applies to the experiments in page 7, line 240, Figure 3F.
12. Page 7, line 245: Is loss of urothelial differentiation and basalization also observed in DKOs in the pooled scRNA analysis ?
13. Page 7, line 258-260: are the findings in NAT samples due to KMT2C/D mutations or are they a general feature of NAT? The NAT samples in Aran et al., 2017 are from TCGA and the healthy are from Gtex and both should have information on somatic mutations for KMT2C/D. The authors should test if there is an enrichment KMT2C/D mutations in the NAT vs Healthy samples.
14. Line 279-285: testing the chromatin marks is the appropriate way to confirm deletions. As indicated above, the authors should test for protein expression with specific antibodies.
15. Page 8, line 311: Figure S6d shows, Cxxc1 and Set1a signals in Kmt2d bound enhancers. Is there high concordance of Cxxc1 and Set1a peak intensity in all promoter-distal sites or just in Kmt2d bound enhancers? The text should be modified accordingly.
16. Page 9, line 325-326: yet all except for Menin decrease significantly at Kmt2d negative enhancers. Authors should explain how this can be.
17. Page 9, line 328-329: why was this type of normalization done? Has this been published before? Authors should explain.
18. Page 9, line 338 figure 4H: Figure 4H and S6H represent two single experiments but for the same ChIPs (except for H3K27me3 which only was made once). When experimental replicates are performed, the data (reads, etc. should be evaluated together not separately).
19. Page 9, line 349: ChIP-qPCR for K4me1 & K27ac at enhancer and promoter regions for Fgfr3 and Upk3bl as well as RT-qPCR for these genes should be performed in WT and DKO to confirm the seq data.
20. Page 10, line 387: authors should also test how urothelial differentiation genes behave in GSEA.
21. Page 11, line 411 figure 5H: on what basis was the threshold set for defining bivalency and non-bivalency? In figure legend it says "Here, we defined TSS with H3K4me3 \geq 4 and H3K27me3 $<$ 4.5 as H3K4me3 only, while TSS with H3K4me3 \geq 4 and H3K27me3 \geq 4.5 as 1434 bivalent". Authors should explain why they didn't use the called peaks to define these states.
22. Page 11, line 443 figure S11 A-E: ChIP-qPCR for Menin, K4me1 & K27ac at promoter regions for Krt16, Zeb2, Nr4a1, B2m, and H2-K1 should be performed as well as RT-qPCR for these genes in WT and DKO to confirm the seq data.
23. Page 12, line 472-486 figure S13a-d: non-tamoxifen treatment conditions should also be included in the IHCs.
24. Fig. 6b,c: regarding the differences between males and females, it is not clear whether this also reflects extra-urothelial effects. In humans, females have more advanced tumors and worse outcome. This is important considering the fact that the Cre driver is not urothelium-specific and is active in the prostate.
25. Lines 477-480: describe better phenotype of mice in urether and bladder. Fig. 6d-f. Cause of bladder distension? tumor-related?

Minor comments

- Page 3, line 72: reference 4 should be 5, instead.
- Page 6, line 190: Santos et al., 2019 does not describe Krt8 as an umbrella cell marker but an intermediate/supra-basal

marker, as has been shown by other studies. This should be corrected here.

- Page 8, lines 303-306: information on distribution should be reported in a more discriminant manner: promoters, intragenic, intergenic.
- Page 8, line 303: should be rephrased as not all components of the complexes are being ChIP'ed (e.g. Kmt2c, Kdm6a, Kmt2a, Kmt2b, and Set1b are not being ChIPed).
- Page 8, line 308: callout to Fig S6c should come later in line 310 as it shows prior data.
- Page 9, line 356: again, should be rephrased since not all components of the complexes are being ChIPed (e.g., Kmt2c, Kdm6a, Kmt2a, Kmt2b, and Set1b are not being ChIPed)..
- Page 10, line 377: state 3 in Fig5d doesn't seem to have more down genes than state 1.
- Line 489: do the authors really mean "urethral urothelial cancer"? urethral cancers tend to be squamous and unrelated to bladder urothelial tumors.
- Discuss better differences between the role of Kmt2c and Kmt2d, which appears obvious in Fig. S13.

Reviewer #2:

Remarks to the Author:

The manuscript by Wang et al., seeks to determine how urothelial field cancers drive urothelial cancers and in a sense, "license" downstream oncogenic transformation. Overall, the manuscript is highly impactful as it provides a possible epigenetic mechanism for impairing differentiation in the urothelium, by KMT2C/D loss, and priming a basal cell state at risk for subsequent tumor initiation. The studies are considered largely original and the methodology for these studies is an innovative blend of single cell genomics, mouse modeling, and epigenetic analyses. The overall conclusion that KMT2c/d loss elicits a field effect that provides a "license" for transformation is however, not entirely substantiated by the current data. There are, several weaknesses, that if addressed experimentally would support this conclusion. These weaknesses are bulleted below (major and minor).

Major Concerns:

Modeling Field effects in the Urothelium – A major weakness is that it is unclear if there is strong field effect in the urothelium due to the high dose tamoxifen (2x IP) and long lineage tracing (6 months). Whether or not, the Kmt2C/D deleted urothelium clonally expands and competes alongside adjacent normal urothelium is unclear. Low dose tamoxifen followed by a robust chase /timecourse should be considered (i.e. low dose tamoxifen to infrequently label Kmt2c/d clones and then sacrifice 1 week, 2 weeks, 4 weeks, etc.). As it stands now, the Kmt2c/d experiments are fairly static and are a simple description of Kmt2c/d loss in the entire epithelium. In the best case scenario, the EYFP reporter allele could be included in the Kmt2c/d f/f experiments and compared to Kmt2c/d WT in order to compare and contrast field effects across the alleles.

Quantifying Field effects in the Urothelium – The authors provide clear histology demonstrating normal like structure of the urothelium in the tamoxifen induced Kmt2c/d urothelium. However, A major weakness is that field effects are implied in the histologically normal with a correlative downstream assessment of H3K4me1. I find these stains difficult to interpret and not altogether convincing. Although, qPCR is provided indicating robust KO, in situ confirmation is necessary. One easy alternative, would be to perform RNAscope for Kmt2c/d (2 color) plus markers of the urothelium.

Quantifying Field effects as a driver of "primed" or a "license" for tumorigenesis – Figure 6 cooperation experiments are considered important but do again do not unequivocally show mutual loss of PTEN/Kmt2c/Kmt2d in the epithelium. Additional in situ analysis of each allele and important differentiation markers are necessary. Figure 7 is arguably one of the most important figures presented to support the conclusion that field effects prime / license the epithelium for tumorigenesis. The organoid experiments are appropriately conceived but the analyses are lacking from the perspective of field effects and tumors. In the tissues collected could sections be used for field assessment (i.e. how does the Kmt2c/d field size impact tumor growth)? Similarly, these studies are difficult to assess in the autochthonous model (Figure 7hi) due to the H3k4me1 staining. Additional mRNA FISH for the dKO alleles should be considered. Moreover, the BBN treatment really assesses cooperativity and not field driven effects. A mosaic experiment (i.e. low dose tamoxifen) and careful assessment of tumors for field origin (i.e. Kmt2c/d loss) would be highly impactful and support the overall conclusion.

Single cell genomics analyses – Comparing multiple datasets in scRNASeq can be challenging, and many techniques have been developed (including various data integration techniques developed by R. Satija). The current methods do not adequately describe which technique is being used. In addition, alternative non integrative techniques should also be considered and cell typing using available workflows (i.e. SCType) should be considered. The current datasets only include N = 4 (WT) and N = 3 (dKO) so drawing conclusions regarding cell state changes is difficult. Additional biological replicates are necessary – and could be performed using alternative techniques that are more cost effective (i.e. RNAscope, IF of multiple samples) + markers of each cell lineage.

Impaired basal cell differentiation – This conclusion requires additional experiments that include appropriate Cre drivers in the basal compartment. The authors remark that the Tmress2-CreER is present in all layers of the urothelium. So, it is unclear if the basal state found in the dKO is due to impaired differentiation or induction of cellular plasticity from differentiated progeny (i.e. intermediate cells and umbrella cells).

Epigenetic analyses – The epigenetic analysis appears to be quite comprehensive, but as written it is extremely dense and challenging to follow.

Therapeutic vulnerability to EGFP inhibitors – The EGFR inhibition studies are considered somewhat underdeveloped and

overstated. For instance, Afatinib and Gefitinib both seem to equally potent on WT and dKO cells - multiple independent experiments and statistical assessment of EC50 would be necessary to prove otherwise. In addition, in the natural state a mixture of WT and dKO cells would likely be present in the field. How does mixing affect responses and could clonal competition / vulnerability be assessed? The experiment in 8c is still very early in the tumor growth phase. It is unclear if treatment will reduce tumor growth rates if the tumors are allowed to grow to tumor endpoint (2000 mm³). The representative stains in 8d are also of low quality.

Minor Concerns:

Image analysis – several opportunities for image analysis are missed, and if included, would greatly bolster their conclusions. This includes panels like Figure 2f.

Organoids and cell lines – neither model is well described in the text. In addition, confirmation of dKO status is not clear. This is especially important when considering the bottlenecks that occur during selection for organoid growth and cell line growth. The invasion potential seems to be underdeveloped, and contrary to the idea that the dKO are histologically normal but show increased invasion potential *ex vivo*? The increased EGF responsiveness is certainly interesting, but it is unclear how this changes the phenotype of the cells in their *in vitro* assays.

Figure layouts – some of the figures are challenging to follow. Consider Figure 3 as an example, the panel layout is out of order. Others, including Figure 7 are challenging to follow in the current layout.

Model in Figure 8e – the proposed model in Figure 8 is very challenging to follow and not well described in the text / legend.

Reviewer #3:

Remarks to the Author:

The study by Wang et al investigates the role of KMT2C and KMT2D in the urothelium primarily using a genetically engineered mouse model. The mouse model is based on a Cre driver that the authors had previously identified for its use in prostate cancer, however, the Cre driver is not restricted to prostate but rather achieves gene recombination in a variety of other tissues, one of which being the bladder urothelium, which is the focus of the current study. It is important to note at the outset that the authors do not explicitly discuss the distribution of Cre recombination in these mice and that this reviewer had to look up several publications to piece this together. Not only do the authors not specifically address this point, the data presented in the manuscript including in the supplementary figures (S1 and S2) do not explicitly look at other tissues, only bladder. This raises serious concerns that the authors' intentions are not to be forthcoming about this very important feature of their mouse model. Additionally, the fact that the Cre driver is not limited to bladder also raises the issue about the cell autonomy of the phenotype; this is partially but fully addressed by the organoid models.

Bearing in mind this important caveat (regarding the validity of their GEMM), what the authors show is that, on their own, KMT2C/KMT2D, are not sufficient to promote bladder tumorigenesis since GEMMs lacking KMT2C/KMT2D have only a subtle histological phenotype. In collaboration with other tumor drivers, namely loss of PTEN or the carcinogen BBN, the phenotype is modestly potentiated. In fact the PTEN compound mice die of an alternative cancer type. To this reviewer, these findings suggest that KMT2C/KMT2D are insufficient for bladder cancer; however, the authors conclude that KMT2C/KMT2D promote initiation. It is difficult to even understand how they could come to this conclusion based on the data provided. It might be possible to make this conclusion about initiation, if they temporally ordered the events (so for example initiated PTEN deletion and then KMT2C/KMT2D or vice versa). But the definitive experiments are not provided and instead we are left with a rather weak phenotype and an unsubstantiated conclusion. Another possibility is that the collaborating event has not been tested - for example, perhaps p53 would be more relevant than PTEN (since p53 is mutated in >50% of invasive bladder cancer). In fact, PTEN is rarely mutated in invasive bladder cancer and almost never together with KMT2C/KMT2D.

[This reviewer notes that the characterization of the GEMM phenotype is missing basic characterization of mouse cancer models, such as tumor (bladder) weights, proliferation rate, etc. Ki67 is shown for Pten in Fig. 6]

Following from the observations that KMT2C/KMT2D loss in GEMMs results in modest effects on bladder histopathology, the authors then embark on a series of studies to characterize their molecular phenotype. They show that these mice have more basal features and fewer luminal ones. This is an interesting observation but not evidence of a field effect. The fact that the normal adjacent bladder urothelium has similar molecular changes is correlative not causal evidence. Definitive experiments to show causality are not provided.

Overall, this is a very disappointing study which is remarkably over interpreted while lacking key definitive experiments on causal relationships, and based on a GEMM that is essentially misrepresented.

Specific comments:

This reviewer notes that KMT2C and KMT2D actually differ in mice and humans (their names are actually inverted) - the authors do not note this (perhaps they are not aware of this).

Figure 1: The phenotype of the GEMMs are not sufficiently well documented; the specificity of the Cre driver not provided or discussed; and the phenotype in other tissues in which the Cre is active is not discussed or described. The authors do not consider whether their findings are cell autonomous to the urothelium (or for example dependent on recombination of

KMT2C/KMT2D in other tissues/cell types).

Figure 2: The data in this figure show an increase in basal cells/features; these data are not sufficient to draw a conclusion that KMT2C/KMT2D alters "stem cell potential, basal differentiation or EMT". The results are modest and could be reflective of increased proliferation (which is not quantified in the figure).

Figure 3: It is very difficult to understand how these data support a "pre-tumorigenic" effect; the lack of cancerous phenotype (as is the case in the KMT2C/KMT2D GEMM) is not indicative of a "pre-tumor" phenotype it is actually better described as a non-tumor phenotype.

Figure 6: It seems unlikely from the phenotype that the mice are dying of bladder cancer, making panels b and c quite misleading.

Figure 7: These data seem out of place since they actually address the cooperativity of KMT2C/KMT2D with various other relevant drivers of bladder cancer. However, it is not possible to say that KMT2C/KMT2D prime the cancers since the temporal order of deletion is not controlled for. These data are also not fully developed and seem to be added after the fact, when in fact if properly developed might support the conclusions they would like to make. [These data would need to be much more fully developed to support their conclusions.]

Figure 8: It is hard to understand how the authors could make a conclusion about EGFR inhibition KMT2C/KMT2D contests when they have not provided a strong case for the role of these genes in bladder cancer. They are also lacking the extremely obvious definitive connection to the molecular studies that they provided in the previous figures.

Other specific comments:

Line 50 — "licence a field effect..." egregiously overstated and overintepreted.

Line 55 — These sentences do not connect

Line 67 — Needs a reference

Line 80-81 — Unclear what Kabuki syndrome and Kleeftstra syndrome have to do with bladder cancer or any of the results presented herein.

Line 94-97 — Conclusions not supported by the data.

Line 104-105 — Since the data do not demonstrate a role for KMT2C/KMT2D in bladder cancer, hard to understand how they can nominate a druggable target.

Line 112-113 — Alarmingly misleading since there is no mention that their Cre driver results in gene recombination in many tissues.

Line 130-136 — Why look only at the catalytic function of KMT2C/KMT2D? Why not measure the deletion of KMT2C/KMT2D directly in the cells - this seems worrisome that these direct data are not shown. [It is often the case that the cells in which the genes are actually deleted can be eliminated several months later.]

Version 1:

Decision Letter:

Our ref: NG-A62824R

18th Jul 2024

Dear Dr Chen,

Thank you for submitting your revised manuscript "Kmt2c/d loss primes urothelium for tumorigenesis and redistributes Kmt2a to bivalent promoters" (NG-A62824R). It has now been seen by the original referees and their comments are below. The reviewers find that the paper has improved in revision, and therefore we'll be happy in principle to publish it in Nature Genetics, pending minor revisions to comply with our editorial and formatting guidelines.

Sincerely,

Safia Danovi, PhD
Senior Editor, Nature Genetics
ORCID: 0009-0007-7822-5479

Reviewer #1 (Remarks to the Author):

In this paper, Wang et al assess how loss-of-function of Kmt2c/Kmt2d contribute to urothelial bladder cancer using a wide variety of approaches and experimental models. As I already indicated in my prior review of this manuscript, the work represents a real tour de force towards our understanding of how these genes involved in epigenetic regulation contribute to tumorigenesis. They show that LOF primes urothelial cells for proliferation/transformation in vitro and in vivo. They then show that their LOF drives redistribution of the Kmt2a/Menin complex to enhancers and bivalent promoters with resulting gene expression. They also show cooperation of these LOF events with Pten towards transformation in the urothelium. Accompanying these findings, they note that a basalization occurs in the urothelium. They finally show that Kmt2c/d LOF confers sensitivity to EGFR inhibitors. The paper is data-dense but is well written, flows well and it is an important contribution to the field.

The authors have largely responded to my comments and have performed additional experiments. While there are minor points where I would think that further work would be required to substantiate the author's conclusions, I do not think that it would fair to delay publication any further.

Congratulations to the authors for this excellent piece o work.

Reviewer #2 (Remarks to the Author):

My concerns have been adequately addressed.

Version 2:

Decision Letter:

In reply please quote: NG-A62824R1 Chen

23rd Oct 2024

Dear Dr Chen,

I am delighted to say that your manuscript "Kmt2c/d loss primes urothelium for tumorigenesis and redistributes KMT2A to bivalent promoters" has been accepted for publication in an upcoming issue of Nature Genetics.

Your paper will be published online after we receive your corrections and will appear in print in the next available issue. You can find out your date of online publication by contacting the Nature Press Office (press@nature.com) after sending your e-proof corrections.

Before your paper is published online, we shall be distributing a press release to news organizations worldwide, which may very well include details of your work. We are happy for your institution or funding agency to prepare its own press release, but it must mention the embargo date and Nature Genetics. Our Press Office may contact you closer to the time of

publication, but if you or your Press Office have any enquiries in the meantime, please contact press@nature.com.

Please note that *Nature Genetics* is a Transformative Journal (TJ). Authors may publish their research with us through the traditional subscription access route or make their paper immediately open access through payment of an article-processing charge (APC). Authors will not be required to make a final decision about access to their article until it has been accepted. [Find out more about Transformative Journals](https://www.springernature.com/gp/open-research/transformative-journals)

Authors may need to take specific actions to achieve [compliance](https://www.springernature.com/gp/open-research/funding/policy-compliance-faqs) with funder and institutional open access mandates. If your research is supported by a funder that requires immediate open access (e.g. according to [Plan S principles](https://www.springernature.com/gp/open-research/plan-s-compliance)) then you should select the gold OA route, and we will direct you to the compliant route where possible. For authors selecting the subscription publication route, the journal's standard licensing terms will need to be accepted, including [those licensing terms](https://www.nature.com/nature-portfolio/editorial-policies/self-archiving-and-license-to-publish) will supersede any other terms that the author or any third party may assert apply to any version of the manuscript.

If you have not already done so, we strongly recommend that you upload the step-by-step protocols used in this manuscript to protocols.io. protocols.io is an open online resource that allows researchers to share their detailed experimental know-how. All uploaded protocols are made freely available and are assigned DOIs for ease of citation. Protocols can be linked to any publications in which they are used and will be linked to from your article. You can also establish a dedicated workspace to collect all your lab Protocols. By uploading your Protocols to protocols.io, you are enabling researchers to more readily reproduce or adapt the methodology you use, as well as increasing the visibility of your protocols and papers. Upload your Protocols at <https://protocols.io>. Further information can be found at <https://www.protocols.io/help/publish-articles>.

Sincerely,

Safia Danovi, PhD
Senior Editor, Nature Genetics
ORCID: 0009-0007-7822-5479

Click here if you would like to recommend Nature Genetics to your librarian
<http://www.nature.com/subscriptions/recommend.html#forms>

** Visit the Springer Nature Editorial and Publishing website at [www.springernature.com/editorial-and-publishing-jobs](http://editorial-jobs.springernature.com?utm_source=ejp_NGen_email&utm_medium=ejp_NGen_email&utm_campaign=ejp_NGen) for more information about our career opportunities. If you have any questions please click [here](mailto:editorial.publishing.jobs@springernature.com). **
